## Mid-Holocene Climate Change over China: Model-Data Discrepancy

Yating Lin [1,2,4], Gilles Ramstein [2], Haibin Wu [1,3,4], Raj Rani [2], Pascale Braconnot [2],

Masa Kageyama [2], Qin Li [1,3], Yunli Luo [5], Ran Zhang [6] and Zhengtang Guo [1,3,4]

1. Key Laboratory of Cenozoic Geology and Environment, Institute of Geology and Geophysics, Chinese Academy of Sciences, Beijing 100029, China

2. Laboratoire des Sciences du Climat et de l'Environnement, LSCE/IPSL, CEA-CNRS-UVSQ, Université Paris-Saclay, Gif-sur-Yvette 91191, France

3. CAS Center for Excellence in Life and Paleoenvironment, Beijing, 100044, China

4. University of Chinese Academy of Sciences, Beijing 100049, China

5. Institute of Botany, Chinese Academy of Sciences, Beijing 100093, China

6. Institute of Atmospheric Physics, Chinese Academy of Sciences, Beijing 100029, China

**Abstract:**

The mid-Holocene period (MH) has long been an ideal target for the validation of Global Circulation Model (GCM) results against reconstructions gathered in global datasets. These studies aim to test the GCM sensitivity mainly to the seasonal changes induced by the orbital parameters (longitude of the perihelion). Despite widespread agreement between model results and data on the MH climate, some important differences still exist. There is no consensus on the continental size (the area of the temperature anomaly) of the MH thermal climate response, which makes regional quantitative reconstruction critical to obtain a comprehensive understanding of the MH climate patterns. Here, we compare the annual and seasonal outputs from the most recent Paleoclimate Modelling Intercomparison Project Phase 3 (PMIP3) models with an updated synthesis of climate reconstruction over China, including, for the first time, a seasonal cycle of temperature and precipitation. Our results indicate that the main discrepancies between model and data for the MH climate are the annual and winter mean temperature. A warmer-than-present climate condition is derived from pollen data for both annual mean

temperature (~0.7 K on average) and winter mean temperature (~1 K on average), while most of the models provide both colder-than-present annual and winter mean temperature and a relatively warmer summer, showing linear response driven by the seasonal forcing. By conducting simulations in BIOME4 and CESM, we show that the surface processes are the key factors drawing the uncertainties between models and data. These results pinpoint the crucial importance of including the non-linear responses of the surface water and energy balance to vegetation changes.

*Keywords:* PMIP3    Pollen data    Inverse Vegetation Model    Seasonal climate change

## 1. Introduction

Much attention of paleoclimate studies has been focused on the current interglacial (the Holocene), especially the mid-Holocene (MH, 6±0.5 ka). The major difference in the experimental configuration between the MH and pre-Industrial (PI) arises from the orbital parameters which brings about an increase in the amplitude of the seasonal cycle of insolation of the Northern Hemisphere and a decrease in the Southern Hemisphere (Berger, 1978). Thus, the MH provides an excellent case study on which to base an evaluation of the climate response to changes in the distribution of insolation. Great efforts have been devoted by the modeling community to design of MH common experiments using similar boundary conditions (Joussaume and Taylor., 1995; Harrison et al., 2002; Braconnot et al., 2007a, b). In addition, much work has been done to reconstruct the paleoclimate change based on different proxies at global and continental scale (Guiot et al., 1993; Kohfeld and Harrison, 2000; Prentice et al.,

2000; Bartlein et al., 2011). The greatest progress in understanding the MH climate change and
variability has consistently been made by comparing large-scale analyses of data with
simulations from global climate models (Joussaume et al., 1999; Liu et al., 2004; Harrison et
al., 2014).
However, the source of discrepancies between model results and data is still an open and
stimulating question. Two types of inconsistencies have been identified: 1) where the model
and data show opposite signs, for instance, paleoclimate evidence from data-records indicates
an increase of about 0.5 K in global annual mean temperature during the MH compared with PI
(Shakun et al.. 2012; Marcott et al., 2013), while there is a cooling trend in model simulations
(Liu et al., 2014). 2) where the same trend is displayed by both model and data but with different
magnitudes. Previous studies have shown that while climate models can successfully reproduce
the direction and large-scale patterns of past climate changes, they tend to consistently
underestimate the magnitude of change in the monsoons of the Northern Hemisphere as well as
the amount of the MH precipitation over northern Africa (Braconnot et al., 2012; Harrison et
al., 2015). Moreover, significant spatial variability has been noted in both observations and
simulations (Peyron et al., 2000; Davis et al., 2003; Braconnot et al., 2007a; Wu et al., 2007;
Bartlein et al., 2011). For instance, Marcott et al. (2013) reconstructed a cooling trend of global
temperature during Holocene, mainly from the marine records (~80%). While based on 642
sub-fossil pollen data, Marsicek et al. (2018) shows a long-term warming defined the Holocene
until around 2000 years ago for Europe and North America continents. The different trends of
pollen- and marine-based reconstruction indicate the spatial variability of annual temperature
change during MH over the globe, which has already been investigated by Bartlein et al. (2010).
That makes regional quantitative reconstruction (Davis et al., 2003; Mauri et al., 2015) essential
to obtain a comprehensive understanding of the MH climate patterns, and to act as a benchmark
to evaluate climate models (Fischer and Jungclaus, 2011; Harrison et al., 2014;).
China offers two advantages with respect to these issues. The sheer expanse of the country
means that the continental response to insolation changes over a large region can be investigated.
Moreover, the quantitative reconstruction of seasonal climate changes during the MH, based on
the new pollen dataset, provides a unique opportunity to compare the seasonal cycles for models
and data. Previous studies indicate that warmer and wetter than present conditions prevailed
over China during the MH and that the magnitude of the annual temperature increases varied
from 2.4-5.8 K spatially, with an annual precipitation increase in the range of 34-267 mm (e.g.,
Sun et al., 1996; Jiang et al., 2010; Lu et al., 2012; Chen et al., 2015). However, Jiang et al.
(2012) clearly show a mismatch between multi-proxy reconstructions and model simulations.
In terms of climate anomalies (MH-PI), besides the ~1 K increase in summer temperature, 35
out of 36 Paleoclimate Modelling Intercomparison Projects (PMIP) models reproduce annual
(~0.4 K) and winter temperatures (~1.4 K) that are colder than the baseline, and a drier-than-
baseline climate in some western and middle regions over China is depicted in models (Jiang
et al., 2013). Jiang et al. (2012) point out the model-data discrepancy over China during the
MH, but the lack of seasonal reconstructions in their study limits comparisons with simulations.
An important issue raised by Liu et al. (2014) is that the discrepancy at the annual level could
be due to incorrect reconstructions of the seasonal cycle, a key objective in our paper. Moreover,
it has been suggested that the vegetation change can strengthen the temperature response in
high latitudes (O'Ishi et al., 2009; Otto et al., 2009), as well as alter the hydrological conditions
in the tropics (Z. Liu et al., 2007). However, compared to the substantial land cover changes in
the MH derived from pollen datasets (Ni et al., 2010; Yu et al., 2000), the changes in vegetation
have not yet been fully quantified and discussed in PMIP3 (Taylor et al., 2012).
In this study, we first present new reconstructions. We use a quantitative biomization method
to reconstruct vegetation types during the MH, based on a new synthesis of pollen datasets, and
then use an Inverse Vegetation Model (Guiot et al., 2000; Wu et al., 2007) to obtain the mean
annual temperature, the mean temperature of the warmest month (MTWA), the mean
temperature of the coldest month (MTCO), the mean annual precipitation, July precipitation
and January precipitation over China for the MH. Furthermore, we present a comprehensive
evaluation of the PMIP3 simulations performed with state-of-art climate models, based on our
reconstructions of temperature and precipitation. This is the first time that such a progress
towards a quantitative seasonal climate comparison for the MH over China has been made. This
point is crucial because the MH PMIP3 experiment is essentially one that looks at the response
of the models to changes in the seasonality of insolation, and the attempt to derive
reconstructions of both summer and winter climate to compare with the simulations will thus
enable us to answer the question posed by Liu et al. (2014) on the importance of seasonal
reconstructions.
**2. Data and Methodology**
**2.1 Data**
In this study, we collected 159 pollen records, covering most of China, for the MH period
(6000±500 $^{14}$C yr BP) (Fig. 1). Notably, according to IntCal13 (Reimer et al., 2013), the MH
time slice 6000±500 $^{14}$C yr BP is about 6800 Cal BP (the average value), which is not totally
consistent with the "mid-Holocene" used in CMIP5/PMIP3 experiment (6000 Cal BP). But for
a better comparison with BIOME6000 (in which the MH is defined as 6000±500 14C yr BP),
we decided to choose the pollen data at 6000±500 14C yr BP in our study. In the 159 records,
were from the China Quaternary Pollen Database (CQPD, 2000), three were original datasets
obtained for our study, and the others were digitized from pollen diagrams in published papers
with a recalculation of pollen percentages based on the total number of terrestrial pollen types.
These digitized 91 pollen records were selected according to three criteria: (1) clearly readable
pollen diagrams with a reliable chronology with the minimum of three independent age control
points since the LGM; (2) including the pollen taxa during $6000 \pm 500$ $^{14}$C yr BP period with a
minimum sampling resolution of 1000 years per sample; (3) abandon the pollen records if the
published paper mentions the influence of human activity. For the age control, different dating
methods are utilized in the collected pollen records, we applied CalPal 2007 (Weninger et al.,
2007) to correct $^{14}$C age into calendar age so that they can be contrasted with each other. For
lacustrine records, if the specific carbon pool age is mentioned in the literature, the calendar
age is corrected after deducting the carbon pool. Otherwise, the influence of carbon pool is not
considered. The age-depth model for the pollen records was estimated by linear interpolation
between adjacent available dates or by regression. Using ranking schemes from the Cooperative
Holocene Mapping Project, the quality of dating control for the mid-Holocene was assessed by
assigning a rank from 1 to 7. 70% of the records used in our study fell into the first and second
classes (see Table 1 for detailed information) according to the Webb 1-7 standards (Webb, III
T., 1985). Vegetation type was quantitatively reconstructed using biomization (Prentice et al.,
1996), following the classification of plant functional types (PFTs) and biome assignment in
China by the Members of China Quaternary Pollen Data (CQPD, 2000), which has been widely
tested in surface sediment. The new sites (91 digitized data and three original data) added to
our database improved the spatial coverage of pollen records, especially in the northwest, the
Tibetan Plateau, the Loess Plateau and southern regions, where the data in the previous
databases are very limited.
Modern monthly mean climate variables investigated in this study, including temperature,
precipitation and cloudiness (total cloud fraction), have been collected for each modern pollen
site based on the datasets (1951-2001) from 657 meteorological observation stations over
China (China Climate Bureau, China Ground Meteorological Record Monthly Report, 1951-
2001). The MH soil properties and characteristics used in the inverse vegetation model were
kept the same with PI conditions, which are derived from the digital world soil map produced
by the Food and Agricultural organization (FAO) (FAO, 1995). Atmospheric $CO_2$
concentration for the MH was taken from ice core records (EPICA community members
2004), and was set at 270 ppmv.

A 3-layer back-propagation (BP) artificial neural network technique (ANN) was used for

interpolation on each pollen site (Caudill and Butler, 1992). Five input variables (latitude,
longitude, elevation, annual precipitation, annual temperature) and one output variable (biome
scores) have been chosen in ANN for the modern vegetation. The ANN has been calibrated
on the training set, and its performance has been evaluated on the verification set (20%,
randomly extracted from the total sets). After a series of training run, the lowest verification
error is obtained with 5 neurons in the hidden layer after 10000 iterations. In our study, at
each pollen site, we firstly applied the biomization method to get the biome scores for both
present-day and MH. The anomalies between past (6 ka) and modern vegetation indices
(biome scores) was then interpolated to the 0.2×0.2° grid resolution by applying the ANN.
After that, the modern grid values are added to the values of the grid of palaeo-anomalies to
provide gridded paleo-biome indices. Finally, the biome with the highest index is attributed to
each grid point. This ANN method is more efficient than many other techniques on the
condition that the results are validated by independent data sets, and therefore, it has been
widely applied in paleoclimatology (Guiot et al., 1996; Peyron et al., 1998). The schematic
diagram of ANN (Figure S1) can be found in Supplementary Information.
**2.2 Climate models**

PMIP, a long-standing initiative, is a climate-model evaluation project which provides an

efficient mechanism for using global climate models to simulate climate anomalies for past
periods and to understand the role of climate feedbacks. In its third phase (PMIP3, Braconnot
et al., 2011), the models were identical to those used in the Climate Modelling Intercomparison
Project 5 (CMIP5) experiments. The experimental set-up for the mid-Holocene simulations in
PMIP3    followed    the    PMIP    protocol    (Taylor    et    al.,    2012,
https://wiki.lsce.ipsl.fr/pmip3/doku.php/pmip3:design:6k:final). The main forcing between the
MH and PI in PMIP3 is the change in the orbital configuration. More precisely, the orbital
configuration in the MH climate has an increased summer insolation and a decreased winter
insolation in the Northern Hemisphere compared to the PI climate (Berger, 1978). In addition,
the $CH_4$ concentration is prescribed at 650 ppbv in the MH, while it is set at 760 ppbv in PI
(Table 2).
All 13 models (Table 3) from PMIP3 that have the MH simulation have been included in
our study, including eight atmosphere-ocean (AO) models and five atmosphere-ocean-
vegetation (AOV) models. Means for the last 30 years were calculated from the archived time-
series data on individual model grids for climate variables. Then the near surface temperature
and precipitation flux, were bi-linearly interpolated to a uniform 2.5° grid, in order to compute
the bioclimatic variables (e.g. MAT, MAP, MTWM, MTCO, July precipitation) onto a common
grid for comparison with the reconstruction results.
**2.3 Vegetation model**
The vegetation model, BIOME4 is a coupled bio-geography and biogeochemistry model
developed by Kaplan et al. (2003). Monthly mean temperature, precipitation, sunshine
percentage (an inverse measure of cloud area fraction), absolute minimum temperature,
atmospheric $CO_2$ concentration and subsidiary information about the soil's physical properties
like water retention capacity and percolation rates are the main input variables. It represents 13
plant functional types (PFTs), which have different bioclimatic limits.   The PFTs are based on
physiological attributes and bioclimatic tolerance limits such as heat, moisture and chilling
requirements and resistance of plants to cold. These limits determine the areas where the PFTs
can grow in a given climate. A viable combination of these PFTs defines a particular biome
among 28 potential options. These 28 biomes can be further classified into 8 megabiomes
(Table S1). BIOME4 has been widely utilized to analyze the past, present and potential future
vegetation patterns (e.g. Bigelow et al., 2003; Diffenbaugh et al., 2003; Song et al., 2005). In
this study, we conducted 13 PI and associated MH biome simulations using PMIP3 climate
fields (temperature, precipitation and sunshine) as inputs. The climate fields, obtained from
PMIP3, are the monthly mean data of the last 30 model years.

**2.4 Statistics and interpolation for vegetation distribution**

To quantify the differences between simulated (based on BIOME4 forced by the climate
model output) and reconstructed (from pollen) megabiomes, a map-based statistic (point-to-
point comparison with observations) called $\Delta V$ (Sykes et al., 1999; Ni et al., 2000) was applied
to our study. $\Delta V$ is based on the relative abundance of different plant life forms (e.g. trees, grass,
bare ground) and a series of attributes (e. g. evergreen, needle-leaf, tropical, boreal) for each
vegetation class. The definitions and attributes of each plant form follow naturally from the
BIOME4 structure and the vegetation attribute values in the $\Delta V$ computation were defined for
BIOME4 in the same way as for BIOME1 (Sykes et al., 1999). The abundance and attribute
values are given in Table 4 and Table 5, which describe the typical floristic composition of the
biomes. Weighting the attributes is subjective because there is no obvious theoretical basis for
assigning relative significance. Transitions between highly dissimilar megabiomes have a
weighting of close to 1, whereas transitions between less dissimilar megabiomes are assigned
smaller values. The overall dissimilarity between model and data megabiome maps was
calculated by averaging the $\Delta V$ for the grids with pollen data, while the value was set at 0 for
any grid without data. $\Delta V$ values $< 0.15$ can be considered to point to very good agreement
between simulated and actual distributions, 0.15-0.30 is good, 0.30-0.45 is fair, 0.45-0.60 is
poor, and $> 0.80$ is very poor (adjusted from Zhang et al., 2010).

**2.5 Inverse vegetation model**

The Inverse Vegetation Model (Guiot et al., 2000; Wu et al., 2007), highly dependent on the BIOME4 model, is applied to our reconstruction. The key concept of this model can be summarized in two points: firstly, a set of transfer functions able to transform the model output into values directly comparable with pollen data is defined. There is not full compatibility between the biome typology of BIOME4 and the biome typology of pollen data. A transfer matrix (Table S2) was defined in our study where each BIOME4 vegetation type is assigned a vector of values, one of each pollen vegetation type, ranging from 0 (representing an incompatibility between BIOME4 type and pollen biome type) to 15 (corresponding to a maximum compatibility). Secondly, using an iterative approach, a representative set of climate scenarios compatible with the vegetation records is identified within the climate space, constructed by systematically perturbing the input variables (e.g. $\Delta T$, $\Delta P$) of the model (Table S3).

The Inverse Vegetation Model (IVM) provides a possibility, for the first time, to reconstruct both annual and seasonal climates for the MH over China. Moreover, it offers a way to consider the impact of $CO_2$ concentration on competition between PFTs as well as on the relative abundance of taxa, and thus make the reconstructions from pollen records more reliable. More detailed information about IVM can be found in Wu et al. (2007).

We applied the inverse model to modern pollen samples to validate the approach by reconstructing the modern climate at each site and comparing it with the observed values. The high correlation coefficients (R=0.75–0.95), intercepts close to 0 (except for the mean temperature of the warmest month), and slopes close to 1 (except for the July precipitation) demonstrated that the inversion method worked well for most variables in China (see Table 6).

## 2.6 Sensitivity test for vegetation feedback

To quantify the vegetation feedback on climate change during mid-Holocene over China, we performed a sensitivity test using CESM version 1.0.5. This version, developed at the National Center for Atmospheric Research, is a widely used coupled model with dynamic atmosphere (CAM4), land (CLM4), ocean (POP2), and sea-ice (CICE4) components (Gent et al., 2011). Here, we use ~2° resolution for CAM4(~1.9° for latitudes × 2.5° for longitudes) in the horizontal direction and 26 layers in the vertical direction. The POP2 adopts a finer grid, with a nominal 1° horizontal resolution and 60 layers in the vertical direction. The land and sea-ice components have the same horizontal grids as the atmosphere and ocean components, respectively.

Two experiments were conducted, including a mid-Holocene (MH) experiment (6 ka) with original vegetation setting (prescribed as PI vegetation for MH) and a MH experiment with reconstructed vegetation (6 ka_VEG). In detail, experiment 6 ka used the MH orbital parameters (Eccentricity=0.018682; Obliquity=24.105°; Longitude of the perihelion =0.87°) and modern vegetation (Salzmann et al., 2008). Compared to experiment 6 ka, experiment 6 ka_VEG used our reconstructed vegetation in China. Except for the changed vegetation, all other boundary conditions were kept unchanged in these two experiments, including the solar constant (1365 W m$^{-2}$), modern topography and ice sheet, and pre-industrial greenhouse gases ($CO_2$ = 280 ppmv; $CH_4$ = 760 ppbv; $N_2O$ = 270 ppbv). Experiment 6 ka was initiated from the default pre-industrial simulation and run for 500 model years. Experiment 6 ka_VEG was initiated from model year 301 of experiment 6 ka and run for another 200 model years. We analyzed the computed climatological means of the last 50 model years from each experiment here.

 **3. Results**

 **3.1 Comparison of annual and seasonal climate changes at the MH**

    In this study, we collected 159 pollen records, broadly covering the whole of China (Fig. 1).

 To check the reliability of the collected data, we first categorized our pollen records into

 megabiomes in line with the standard tables developed for the BIOME6000 (Table S1), and

 compared them with the BIOME6000 dataset (Fig.2). The match between collected data and

 the BIOME6000 is more than 90% (145 out of 159 sites) for both the MH and PI.

    Based on pollen records, the spatial pattern of climate changes over China during the MH,

 deduced from IVM, are presented in Fig. 3 (left panel, points), alongside the results from PMIP3

 models (shaded in Fig. 3). For temperature, a warmer-than-present annual climate condition

 (~0.7 K on average) is derived from pollen data (the points in Fig. 3a), with the largest increase

 occurring in the northeast (3-5 K) and a decrease in the northwest and on Tibetan Plateau. On

 the other hand, the results from a multi-model ensemble (MME) indicate a colder annual

 temperature generally (~-0.4 K on average), with significant cooling in the south and slight

 warming in the northeast (shaded in Fig. 3a). Of the 13 models, 11 simulate a cooler annual

 temperature compared with PI as MME. However, two models (HadeGEM2-ES and CNRM-

 CM5) present the same warmer condition as was found in the reconstruction (Fig. 3d).

 Compared to the reconstruction, the annual mean temperature during the MH is largely

 underestimated by most PMIP3 models, which depict an anomaly ranging from ~-1 to ~0.5 K.

 Concerning seasonal change, during the MH, MTWA from the data is ~0.5 K higher than PI,

 with the largest increase in the northeast and a decrease in the northwest. From model outputs,

 an average increase of ~1.2 K is reproduced by MME, with a more pronounced warming at

 high latitudes which is consistent with the insolation change (Berger, 1978). Fig. 3e shows that

 all 13 models reproduce the same warmer summer temperatures as the data, and that

HadGEM2-ES and CNRM-CM5, reproduce the largest increases among the models. Although
models simulate warmer MTWA, which is consistent with reconstructions, there is a
discrepancy between them on MTCO. In Fig. 3c, the data show an overall increase of ~1 K,
with the largest increase occurring in the northeast and a decrease of opposite magnitude on the
Tibetan Plateau. Inversely, MME reproduces a decreased MTCO with an average amplitude of
~-1.3 K, the areas with strongest cooling being the southeast, the Loess Plateau and the
northwest. Similarly to the MME, all 13 models simulate a colder-than-present climate with
amplitudes ranging from ~-2.0 K (CCSM4 and FGOALS-g2) to ~-0.7 K (HadGEM2-ES and
CNRM-CM5).
Concerning the annual change in precipitation, the reconstruction shows wetter conditions
during the MH across almost the whole of China with the exception of part of the northwest.
The southeast presents the largest increase in annual precipitation. All but 2 models (MIROC-
ESM and FGOALS-g2) depict wetter conditions with an amplitude of ~10 mm to ~50 mm. The
reconstruction and MME results also indicate an increased annual precipitation during MH
(Fig.4a), with a much larger magnitude visible in the reconstruction (~30 mm, ~230 mm
respectively). The main discrepancy in annual precipitation between simulations and
reconstruction occurs in the northeast, which is depicted as drier by the models and wetter by
the data. With regard to seasonal change, the reconstruction shows an overall increase in July
rainfall (~50 mm on average), with a decrease in the northwestern regions and East Monsoon
region at Yangtze River valley. In line with the reconstruction, the MME also shows an overall
increase in rainfall (~7 mm on average), with a decrease in the northwest for July (Fig.4b).
Notably, a much larger increase is simulated for the south and the Tibetan Plateau by the models,
while the opposite pattern emerges along the eastern margin from both models and data. For
January precipitation, the reconstruction shows an overall increase in most region (~15 mm),
except for the northwestern region, while MME indicates a slight decrease (~3 mm on average).
More detailed information about the geographic distribution of simulated temperature and
precipitation for each model can be found in Fig. S2-S7.
Table S4 provides the biome score from IVM for pollen data collected from published papers.
The reconstructed climate change derived from IVM at each pollen site can be found in Table
S5, in which the columns show the median and the 90% interval (5th to 95th percentage) for
feasible climate values produced with the IVM approach. The simulated values for each of the
climate variables as shown in the boxplots (Figure 3 and Figure 4) are given in the Table S6
and Table S7. All the values mentioned above are the mean of the values at 159 pollen sites.
**3.2 Comparison of vegetation change at the MH**
The use of the PMIP3 database is clearly limited by the different vegetation inputs among
the models for the MH period (Table S8). Only HadGEM2-ES and HadGEM2-CC use a
dynamic vegetation for the MH, and the vegetation of the other 11 models are prescribed to PI
with or without interactive LAI, which can introduce a bias to the role of vegetation-atmosphere
interaction in the MH climates. To evaluate the model results against the reconstruction for the
MH vegetation, we conducted 13 biome simulations with BIOME4 using the PMIP3 climate
fields, and the megabiome distribution for each model during the MH is displayed in Fig. 5 (see
Fig. S8 for a comparison of PI biomes). To quantify the model-data dissimilarity between
megabiomes, a map-based statistic called $\Delta V$(Sykes et al., 1999; Ni et al., 2000) was applied
here (cf. Section 2.4).
Fig. S9 shows the dissimilarity between simulations and observations for megabiomes during
the MH, with the overall values for $\Delta V$ ranging from 0.43 (HadGEM2-ES) to 0.55 (IPSL-
CM5A-LR). According to the classification of $\Delta V$ (cf. Section 2.4) for the 13 models, 12 (all
except HadGEM2-ES) showed poor agreement with the observed vegetation distribution. Most
models poorly simulate the desert, grassland and tropical forest areas for both periods, but
perform better for warm mixed forest, tundra and temperate forest. However, this statistics is
based on a point-to-point comparison and so the ΔV calculated here cannot represent an
estimation of full vegetation simulation due to the uneven distribution of pollen data and the
potentially huge difference in area of each megabiome. For instance, tundra in our data for PI
is represented by only 4 points, which counts for a small contribution to the ΔV since we
averaged it over a total of 159 points, but this calculation could induce a significant bias if these
4 points are representative of a large area of China.
So, we used the biome scores based on the artificial neural network technique as described
by Guiot et al. (1996) for interpolation (the plots in red rectangle in Fig. 5), and compared the
simulated vegetation distribution from BIOME4 for each model with the interpolated pattern.
During the MH, most models are able to capture the tundra on the Tibetan Plateau as well as
the combination of warm mixed forest and temperate forest in the southeast. However, all
models fail to simulate or underestimate the desert area in the northwest compared to
reconstructed data. The main model-data inconsistency in the MH vegetation distribution
occurs in the northeast, where data show a mix of grassland and temperate forest, and the
models show a mix of grassland and boreal forest.
The area statistics carried out for simulated vegetation changes (Fig. 6) reveals that the main
difference during the MH, compared with PI, is that grassland replaced boreal forest in large
tracts of the northeast (Fig. 5, Fig. S8). No other significant difference in vegetation distribution
between the two periods was derived from models. Unlike in models, three main changes in
megabiomes during the MH are depicted by the data. Firstly, the megabiomes converted from
grassland to temperate forest in the northeast. Secondly, a large area of temperate forest was
replaced in the southeast by a northward expansion of warm mixed forest. Thirdly, in the
northwest and at the northern margin of the Tibetan Plateau, part of the desert area changed into
grassland. However, none of the models succeed in capturing these features, especially the
transition from grassland into forest in the northeast during the MH. Therefore, this failure to
capture vegetation changes between the two periods will lead to a cumulating inconsistency in
the model-data comparison for climate anomalies because if these computed vegetation were
used as boundary conditions in MH climate simulations.
**4.  Discussion**
**4.1 Validation and uncertainties of the reconstructions**
To investigate the discrepancy between model and reconstructions for the MH climate change
over China, the reliability of our reconstruction should be first considered. We therefore
compared our reconstruction with previous studies concerning the MH climate change over
China based on multiple proxies (including pollen, lake core, palaeosol, ice core, peat and
stalagmite), the related references and detailed information are listed in Supplementary
Information (Table S9 and Table S10). In comparison with PI conditions, most reconstructions
reproduced warmer and wetter annual condition during the MH (Fig. 7), as in our study. In
other words, this model-data discrepancy for climate change over China during the MH is
common and robust w.r.t. reconstructions derived from different proxies. Our study reinforces
the picture given by the discrepancies between PMIP simulations and pollen data based on a
synthesis of the literature.
However, there could still be some biases due to the reconstruction method. Estimated
climates for the present day from IVM were compared with observed climates (Table 6). The
slopes and intercepts show a slight bias for annual and January precipitation, while there is a
larger bias between IVM reconstruction and observation for temperature and July precipitation.
IVM relies heavily on BIOME4, and since BIOME4 is a global vegetation model, it is possible
that the spatial robustness of regional reconstruction could be less than that of global
reconstruction due to the failure in simulating local features (Bartlein et al., 2011). Moreover,
the output of the model cannot be directly compared to the pollen data, the conversion of
BIOME4 biomes to pollen biomes by the transfer matrix may add the source of uncertainty in
reconstruction. All these biases in reconstruction should be considered in the evaluation of the
discrepancy between model and data for climate change during the MH over China.
**4.2 Uncertainties for simulations**

The discrepancies between model and data for MH climate change can also result from

uncertainties in simulation and/or model characteristics. First, the coarse spatial resolution of
models can be a factor of discrepancy: previous studies show that the GCMs from PMIP3 are
reliable to simulate the geographical distribution of temperature and precipitation over China
for present day. However, compared with observation, most models have topography-related
cold biases (Jiang et al., 2016). The climate fields, directly used from the model output without
downscaling, will not contain the spatial variability of modern climate that in topographically
complex areas. Thus, it is necessary to check to which degree the model-data mismatch is
related to rough topography used in the climate models. In PMIP3, MRI-CGCM3 has the
highest resolution (Atmosphere: 320*160*L48; Ocean: 364*368*L51), while IPSL-CM5A-LR
has the lowest one (Atmosphere: 96*96*L39; Ocean: 182*149*L31). In Fig. 8, we give the
actual modern topography and the interpolated topography used in MRI-CGCM3 and IPSL-
CM5A-LR. For MRI-CGCM3, the topography is very close to the observation, so for this model,
the model-data discrepancy during MH over China is not related to the resolution. However,
for the model with coarse resolution (IPSL-CM5A-LR), it is true that the coarse version of
model will lead to bias in topography when the regional diversity is discussed. The spatial
variations in topography could influence the vegetation and hence the simulated climate. To
quantify this impact, we compare the topography and PI climate results of IPSL-CM5A-LR and
IPSL-CM5A-MR. Fig. 9 shows that the difference in topography caused by model resolution
does have an impact on small scales (e.g. south region of the Tibetan Plateau), not on the overall

pattern. But those small or regional-scale variations in climate can have a large impact on vegetation and hence reconstructed climate. For a better comparison, in the future work, downscaled climate variables should be considered.

Secondly, besides the qualitative consistency among models, caused by the protocol of PMIP3 experiments (Table 2), a variability in the magnitude of anomalies between models is clearly illustrated by the boxplots (Fig.3 and Fig.4), especially for the temperature anomaly. Fig. S10 demonstrates that there is no clear relationship between PI temperature and temperature anomaly (MH-PI). In other words, these disparities in value or even pattern among models do not related to the difference in PI simulations in a simple manner, instead, they reflect the obvious differences in the response by the climate models to the MH forcing, which raises on the question of the magnitude of feedbacks among models.

As positive feedbacks between climate and vegetation are important to explain regional climate changes, the failure of the models to represent the amplitude and pattern of the observed vegetation differences (see Section 3.2) could amplify and partly account for the model-data disparities in climate change, mainly due to variations in the albedo. Because the HadGEM2-ES and HadGEM2-CC are the only two models in PMIP3 with dynamic vegetation simulation for the MH, we focused on these models to examine the variations in vegetation fraction in the simulations. The main vegetation changes during the MH demonstrated by HadGEM2-ES are increased tree coverage (~15%) and a decreased bare soil fraction (~6%), while HadGEM2-CC depicts a ~3% decrease in tree fraction and a ~1% increase in bare soil (Fig. S11). We made a rough calculation of albedo variance caused solely by vegetation change for both two models and for our reconstruction, based on the area fraction and albedo value of each vegetation type (Betts, 2000; Bonfils et al., 2001; Oguntunde et al., 2006; Bonan, 2008). The overall albedo change from the vegetation reconstruction during the MH shows a ~1.8% decrease when snow-free, with a much larger impact (~4.2% decrease) when snow-covered. The results from

HadGEM2-ES are highly consistent with the albedo changes from the reconstruction, featuring
a ~1.4% (~6.5%) decrease without (with) snow, while HadGEM2-CC produces an increased
albedo value during the MH (~0.22% for snow-free, ~1.9% with snow-cover), depending on its
vegetation simulation. Two ideas could be inferred from this calculation, 1) HadGEM2-ES is
much better in simulating the MH vegetation changes than HadGEM2-CC. 2) the failure by
models to capture these vegetation changes will result in a much larger impact on winter albedo
(with snow) than summer albedo (without snow). In conclusion, there is an obvious advantage
of using AOVGCM instead of AOGCM when we discussing about the MH climate, but the
premise is that the AOVGCM can simulate accurate vegetation distribution.
These surface albedo changes due to vegetation changes could have a cumulative effect on
the regional climate by modifying the radiative fluxes. For instance, the spread of trees into the
grassland biome in the northeast during the MH, revealed by the reconstruction in our study,
should act as a positive feedback to climate warming by increasing the surface net shortwave
radiation associated with reductions in albedo due to taller and darker canopies (Chapin et al.,
2005). Previous studies show that cloud and surface albedo feedbacks on radiation are major
drivers of differences between model outputs for past climates. Moreover, the land surface
feedback shows large disparities among models (Braconnot and Kageyama, 2015).
We used a simplified approach (Taylor et al., 2007) to quantify the feedbacks and to compare
model behavior for the MH, thus justifying the focus on surface albedo and atmospheric
scattering (mainly accounting for cloud change). Surface albedo and cloud change are
calculated using the simulated incoming and outgoing radiative fluxes at the Earth's surface
and at the top of atmosphere (TOA), based on data for the last 30 years averaged from all models.
Using this framework, we quantified the effect of changes in albedo on the net shortwave flux
at TOA (Braconnot and Kageyama, 2015), and further investigated the relationship between
these changes and temperature change. Fig.10 shows that most models produced a negative
cloud cover and surface albedo feedback on the annual mean shortwave radiative forcing.
Concerning seasonal change, the shortwave cloud and surface feedback in most models tend to
counteract the insolation forcing during the boreal summer, while they enhance the solar forcing
during winter. A strong positive correlation between albedo feedback and temperature change
is depicted, with a large spread in the models owing to the difference in albedo in the 13 models.
In particular, CNRM-CM5 and HadGEM2-ES capture higher values of cloud and surface
albedo feedback, which could be the reason for the reversal of the decreased annual temperature
seen in other models (Fig. 3d).
However, the vegetation patterns produced by BIOME4 in Figure 5 are not used in PMIP3
experiment setup. They are onlydetermined by the input variables from models. Therefore,
the disagreements of MH vegetation pattern are possibly inherited from the PI. To better
quantify the vegetation-climate feedback, two experiments were conducted in CESM version
1.0.5, including a mid-Holocene (MH) experiment (6 ka) with original vegetation setting
(prescribed as PI vegetation for the MH) and a MH experiment with reconstructed vegetation
(6 ka_VEG). Fig. 11 shows the climate anomalies (6 ka_VEG minus 6 ka) between two
simulations, for both annual and seasonal scale. For temperature, it is clear that the 6 ka_VEG
simulation reproduces a warmer annual mean climate (~0.3 K on average) as well as an
obviously warmer winter (~0.6 K on average). For precipitation, the reconstructed vegetation
leads to more annual and seasonal precipitation, which can also reconcile the model-data
discrepancy of increase amplitude for precipitation during the MH (data reproduced larger
amplitude than model, revealed by our study). So the mismatch between model and data in
MH vegetation could partly account for the discrepancy of climate due to the interaction
between vegetation and climate through radiative and hydrological forcing with albedo. These
results highlight the value of building a new generation of models able to capture not only the
atmosphere and ocean response, but also the non-linear responses of vegetation and
hydrology to climate change.

## 5.    Conclusion

In this study, we compare the annual and seasonal outputs from the PMIP3 models with an
updated synthesis of climate reconstructions over China, including, for the first time, the
seasonal cycle of temperature and precipitation. In response to the seasonal insolation change
prescribed in PMIP3 for the MH, all models produce similar large-scale patterns for seasonal
temperature and precipitation (higher than present July precipitation and MTWA, lower than
present MTCO). The main discrepancy emerging from the model-data comparison occurs for
the mean annual temperature and MTCO, where data show an increased value and most
models simulate the opposite except CNRM-CM5 and HadGEM2-ES that reproduced the
higher-than-present MH annual temperature. By conducting simulations with BIOME4 and
CESM, we show that surface processes are the key factors explaining the discrepancies
between models and data. These results show the importance of including the non-linear
responses of the surface water and energy balance related to vegetation changes. However, it
should also be noted that prescribing the vegetation with reconstructed biomes would reduce
the power of the biome-based climate reconstruction, owing to the potential circularity
(prescribe the vegetation to get the vegetation). Moreover, besides the vegetation influence, to
the impact of rough topography, soil type and other possible factors on model-data
discrepancy remains to be investigated in future work.

**Data availability**

The PMIP3 output is publicly available on the PMIP website (http://pmip3.lsce.ipsl.fr/) . The
65 pollen biomization results are provided by Members of China Quaternary Pollen Data Base
(CQPD), Table 1 shows the information (including references) of the 91 collected pollen
records and 3 original ones in our study, the biome scores of these 94 pollen records derived
from IVM are listed in Table S4, and the digitized data of pollen can be requested to Qin Li
(liqin@mail.iggcas.ac.cn). All the reconstructed climate values at each pollen site from IVM
are provided in Table S5. For the data from CQPD, the basic information (location, data
supporter, age control and biome type of each site) can be found in CQPD (2000), while the
original data are not publicly available yet. These data can be requested to Yunli Luo
(lyl@ibcas.ac.cn, Institute of Botany, Chinese Academy of Sciences, Beijing, 100093, China),
a core member and academic secretary of the CQPD.
**Author contribution**
Yating Lin carried out the model-data analysis and prepared the first manuscript, Gilles
Ramstein contributed a lot to the paper's structure and content, Haibin Wu provided the
reconstruction results from IVM and contributed the paper's structure and content. Raj Rani-
Singh conducted the BIOME4 simulations. Ran Zhang carried out the simulation in CESM.
Pascale Braconnot, Masa Kegeyama and Zhengtang Guo contributed great ideas on model-data
comparison work. Qin Li and Yunli Luo provided pollen data. All co-authors helped to improve
the paper.
**Competing interest**
The authors declare no competing interests.
**Acknowledgements**
We acknowledge the Paleoclimate Modeling Intercomparison Project and World Climate
Research Program's Working Group on Coupled Modelling, which is responsible for
PMIP/CMIP, and we thank the climate modelling groups for producing and making available
their model output. We are grateful to Marie-France Loutre, Patrick Bartlein and three

anonymous reviewers for constructive comments. This research was funded by the National

Basic Research Program of China (Grant no. 2016YFA0600504), the National Natural Science

Foundation of China (Grant nos. 41572165, 41690114, and 41125011), the Sino-French

Caiyuanpei Program, the Bairen Programs of the Chinese Academy of Sciences, and the JPI-

Belmont PACMEDY project (Grant no. ANR-15-JCLI-0003-01). We also acknowledge Labex

L-IPSL, funded by the French Agence Nationale de la Recherche (Grant #ANR-10-LABX-

0018) for its support to the biome modelling based on the PMIP database.

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

 **Table 1. Basic information of the pollen dataset used in this study**

| Site | Lat | Lon | Alt | Webb 1-7 | Source |
|------|-----|-----|-----|----------|--------|
| **Sujiawan** | 35.54 | 104.52 | 1700 | 2 | original data (Zou et al., 2009) |
| **Xiaogou** | 36.10 | 104.90 | 1750 | 2 | original data (Wu et al., 2009) |
| **Dadiwan** | 35.01 | 105.91 | 1400 | 1 | original data (Zou et al., 2009) |
| **Sanjiaocheng** | 39.01 | 103.34 | 1320 | 1 | Chen et al., 2006 |
| **Chadianpo** | 36.10 | 114.40 | 65 | 2 | Z. Zhang et al., 2007 |
| **Qindeli** | 48.08 | 133.25 | 60 | 2 | Yang and Wang, 2003 |
| **Fuyuanchuangye** | 47.35 | 133.03 | 56 | 3 | Xia, 1988 |
| **Jingbo Lake** | 43.83 | 128.50 | 350 | 2 | C. Li et al., 2011 |
| **Hani Lake** | 42.22 | 126.52 | 900 | 1 | Cui et al., 2006 |
| **Jinchuan** | 42.37 | 126.43 | 662 | 5 | Jiang et al., 2008 |
| **Maar Lake** | 42.30 | 126.37 | 724 | 1 | Liu et al., 2009 |
| **Maar Lake** | 42.30 | 126.37 | 724 | 1 | Liu et al., 2008 |
| **Xie Lake SO4** | 37.38 | 122.52 | 0 | 1 | Zhou et al., 2008 |
| **Nanhuiheming Core** | 31.05 | 121.58 | 7 | 2 | Jia and Zhang, 2006 |
| **Toushe** | 23.82 | 120.88 | 650 | 1 | Liu et al., 2006 |
| **Dongyuan Lake** | 22.17 | 120.83 | 415 | 2 | Lee et al., 2010 |
| **Yonglong CY** | 31.78 | 120.44 | 5 | 3 | Zhang et al., 2004 |
| **Hangzhou HZ3** | 30.30 | 120.33 | 6 | 4 | J. Liu et al., 2007 |
| **Xinhua XH1** | 32.93 | 119.83 | 2 | 3 | Shu et al., 2008 |
| **ZK01** | 31.77 | 119.80 | 6 | 2 | Shu et al., 2007 |
| **Chifeng** | 43.97 | 119.37 | 503 | 2 | Xu et al., 2002 |
| **SZK1** | 26.08 | 119.31 | 9 | 1 | Zheng et al., 2002 |
| **Gucheng** | 31.28 | 118.90 | 6 | 4 | Yang et al., 1996 |
| **Lulong** | 39.87 | 118.87 | 23 | 2 | Kong et al., 2000 |
| **Hulun Lake** | 48.92 | 117.42 | 545 | 1 | Wen et al., 2010 |
| **CH-1** | 31.56 | 117.39 | 5 | 2 | Wang et al., 2008 |
| **Sanyi profile** | 43.62 | 117.38 | 1598 | 4 | Wang et al., 2005 |
| **Xiaoniuchang** | 42.62 | 116.82 | 1411 | 1 | Liu et al., 2002 |
| **Haoluku** | 42.87 | 116.76 | 1333 | 2 | Liu et al., 2002 |
| **Liuzhouwan** | 42.71 | 116.68 | 1410 | 7 | Liu et al., 2002 |
| **Poyang Lake 103B** | 28.87 | 116.25 | 16 | 4 | Jiang and Piperno, 1999 |
| **Baiyangdian** | 38.92 | 115.84 | 8 | 2 | Xu et al., 1988 |
| **Bayanchagan** | 42.08 | 115.35 | 1355 | 1 | Jiang et al., 2006 |
| **Huangjiapu** | 40.57 | 115.15 | 614 | 7 | Sun et al., 2001 |
| **Dingnan** | 24.68 | 115.00 | 250 | 2 | Xiao et al., 2007 |
| **Guang1** | 36.02 | 114.53 | 56 | 1 | Z. Zhang et al., 2007 |
| **Angulinao** | 41.33 | 114.35 | 1315 | 1 | H. Wang et al., 2010 |
| **Yangyuanxipu** | 40.12 | 114.22 | 921 | 6 | Wang et al., 2003 |

| | | | | | |
|---|---|---|---|---|---|
| **Shenzhen Sx07** | 22.75 | 113.78 | 2 | 2 | Zhang and Yu, 1999 |
| **GZ-2** | 22.71 | 113.51 | 1 | 7 | X. Wang et al., 2010 |
| **Daihai99a** | 40.55 | 112.66 | 1221 | 2 | Xiao et al., 2004 |
| **Daihai** | 40.55 | 112.66 | 1221 | 2 | Sun et al., 2006 |
| **Sihenan profile** | 34.80 | 112.40 | 251 | 1 | Sun and Xia, 2005 |
| **Diaojiaohaizi** | 41.30 | 112.35 | 2015 | 1 | Yang et al., 2001 |
| **Ganhaizi** | 39.00 | 112.30 | 1854 | 3 | Meng et al., 2007 |
| **Jiangling profile** | 30.35 | 112.18 | 37 | 1 | Xie et al., 2006 |
| **Helingeer** | 40.38 | 111.82 | 1162 | 3 | X. Li et al., 2011 |
| **Shennongjia2** | 31.75 | 110.67 | 1700 | 1 | Liu et al., 2001 |
| **Huguangyan Maar Lake** | 21.15 | 110.28 | 59 | 2 | Wang et al., 2007 |
| **Yaoxian** | 35.93 | 110.17 | 1556 | 2 | Li et al., 2003a |
| **Jixian** | 36.00 | 110.06 | 1005 | 6 | Xia et al., 2002 |
| **Shennongjia Dajiu Lake** | 31.49 | 110.00 | 1760 | 2 | Zhu et al., 2006 |
| **Qigai nuur** | 39.50 | 109.85 | 1300 | 1 | Sun and Feng, 2013 |
| **Beizhuangcun** | 34.35 | 109.53 | 519 | 1 | Xue et al., 2010 |
| **Lantian** | 34.15 | 109.33 | 523 | 1 | Li and Sun, 2005 |
| **Bahanniao** | 39.32 | 109.27 | 1278 | 1 | Guo et al., 2007 |
| **Midiwan** | 37.65 | 108.62 | 1400 | 1 | Li et al., 2003b |
| **Jinbian** | 37.50 | 108.33 | 1688 | 2 | Cheng, 2011 |
| **Xindian** | 34.38 | 107.80 | 608 | 1 | Xue et al., 2010 |
| **Nanguanzhuang** | 34.43 | 107.75 | 702 | 1 | Zhao et al., 2003 |
| **Xifeng** | 35.65 | 107.68 | 1400 | 3 | Xu, 2006 |
| **Jiyuan** | 37.13 | 107.40 | 1765 | 3 | X. Li et al., 2011 |
| **Jiacunyuan** | 34.27 | 106.97 | 1497 | 2 | Gong, 2006 |
| **Dadiwan** | 35.01 | 105.91 | 1400 | 1 | Zou et al., 2009 |
| **Maying** | 35.34 | 104.99 | 1800 | 1 | Tang and An, 2007 |
| **Huiningxiaogou** | 36.10 | 104.90 | 1750 | 2 | Wu et al., 2009 |
| **Sujiawan** | 35.54 | 104.52 | 1700 | 2 | Zou et al., 2009 |
| **QTH02** | 39.07 | 103.61 | 1302 | 1 | Yu et al., 2009 |
| **Laotanfang** | 26.10 | 103.20 | 3579 | 2 | W. Zhang et al., 2007 |
| **Hongshui River2** | 38.17 | 102.76 | 1511 | 1 | Ma et al., 2003, |
| **Ruoergai** | 33.77 | 102.55 | 3480 | 1 | Cai, 2008 |
| **Hongyuan** | 32.78 | 102.52 | 3500 | 2 | Wang et al., 2006 |
| **Dahaizi** | 27.50 | 102.33 | 3660 | 1 | Li et al., 1988 |
| **Shayema Lake** | 28.58 | 102.22 | 2453 | 1 | Tang and Shen, 1996 |
| **Luanhaizi** | 37.59 | 101.35 | 3200 | 5 | Herzschuh et al., 2006 |
| **Lugu Lake** | 27.68 | 100.80 | 2692 | 1 | Zheng et al., 2014 |
| **Qinghai Lake** | 36.93 | 100.73 | 3200 | 2 | Shen et al., 2004 |
| **Dalianhai** | 36.25 | 100.41 | 2850 | 3 | Cheng et al., 2010 |
| **Erhai ES Core** | 25.78 | 100.19 | 1974 | 1 | Shen et al., 2006 |
| **Xianmachi profile** | 25.97 | 99.87 | 3820 | 7 | Yang et al., 2004 |
| **TCK1** | 26.63 | 99.72 | 3898 | 1 | Xiao et al., 2014 |

| | | | | |
|---|---|---|---|---|
| **Yidun Lake** | 30.30 | 99.55 | 4470 | 4 | Shen et al., 2006 |
| **Kuhai lake** | 35.30 | 99.20 | 4150 | 1 | Wischnewski et al., 2011 |
| **Koucha lake** | 34.00 | 97.20 | 4540 | 2 | Herzschuh et al., 2009 |
| **Hurleg** | 37.28 | 96.90 | 2817 | 2 | Zhao et al., 2007 |
| **Basu** | 30.72 | 96.67 | 4450 | 3 | Tang et al., 1998 |
| **Tuolekule** | 43.34 | 94.21 | 1890 | 1 | An et al., 2011 |
| **Balikun** | 43.62 | 92.77 | 1575 | 1 | Tao et al., 2010 |
| **Cuona** | 31.47 | 91.51 | 4515 | 3 | Tang et al., 2009 |
| **Dongdaohaizi2** | 44.64 | 87.58 | 402 | 1 | Li et al., 2001 |
| **Bositeng Lake** | 41.96 | 87.21 | 1050 | 1 | Xu, 1998 |
| **Cuoqin** | 31.00 | 85.00 | 4648 | 4 | Luo, 2008 |
| **Yili** | 43.86 | 81.97 | 928 | 2 | X. Li et al., 2011 |
| **Bangong Lake** | 33.75 | 78.67 | 4241 | 1 | Huang et al., 1996 |
| **Shengli** | 47.53 | 133.87 | 52 | 2 | CQPD, 2000 |
| **Qingdeli** | 48.05 | 133.17 | 52 | 1 | CQPD, 2000 |
| **Changbaishan** | 42.22 | 126.00 | 500 | 2 | CQPD, 2000 |
| **Liuhe** | 42.90 | 125.75 | 910 | 7 | CQPD, 2000 |
| **Shuangyang** | 43.27 | 125.75 | 215 | 1 | CQPD, 2000 |
| **Xiaonan** | 43.33 | 125.33 | 209 | 1 | CQPD, 2000 |
| **Tailai** | 46.40 | 123.43 | 146 | 5 | CQPD, 2000 |
| **Sheli** | 45.23 | 123.31 | 150 | 4 | CQPD, 2000 |
| **Tongtu** | 45.23 | 123.30 | 150 | 7 | CQPD, 2000 |
| **Yueyawan** | 37.98 | 120.71 | 5 | 1 | CQPD, 2000 |
| **Beiwangxu** | 37.75 | 120.61 | 6 | 1 | CQPD, 2000 |
| **East Tai Lake1** | 31.30 | 120.60 | 3 | 1 | CQPD, 2000 |
| **Suzhou** | 31.30 | 120.60 | 2 | 7 | CQPD, 2000 |
| **Sun-Moon Lake** | 23.51 | 120.54 | 726 | 2 | CQPD, 2000 |
| **West Tai Lake** | 31.30 | 119.80 | 1 | 1 | CQPD, 2000 |
| **Changzhou** | 31.43 | 119.41 | 5 | 1 | CQPD, 2000 |
| **Dazeyin** | 39.50 | 119.17 | 50 | 7 | CQPD, 2000 |
| **Hailaer** | 49.17 | 119.00 | 760 | 2 | CQPD, 2000 |
| **Cangumiao** | 39.97 | 118.60 | 70 | 1 | CQPD, 2000 |
| **Qianhuzhuang** | 40.00 | 118.58 | 80 | 6 | CQPD, 2000 |
| **Reshuitang** | 43.75 | 117.65 | 1200 | 1 | CQPD, 2000 |
| **Yangerzhuang** | 38.20 | 117.30 | 5 | 7 | CQPD, 2000 |
| **Mengcun** | 38.00 | 117.06 | 7 | 5 | CQPD, 2000 |
| **Hanjiang-CH2** | 23.48 | 116.80 | 5 | 2 | CQPD, 2000 |
| **Hanjiang-SH6** | 23.42 | 116.68 | 3 | 7 | CQPD, 2000 |
| **Hanjiang-SH5** | 23.45 | 116.67 | 8 | 2 | CQPD, 2000 |
| **Hulun Lake** | 48.90 | 116.50 | 650 | 1 | CQPD, 2000 |
| **Heitutang** | 40.38 | 113.74 | 1060 | 1 | CQPD, 2000 |
| **Zhujiang delta PK16** | 22.73 | 113.72 | 15 | 7 | CQPD, 2000 |
| **Angulitun** | 41.30 | 113.70 | 1400 | 7 | CQPD, 2000 |

| | | | | | |
|---|---|---|---|---|---|
| **Bataigou** | 40.92 | 113.63 | 1357 | 1 | CQPD, 2000 |
| **Dahewan** | 40.87 | 113.57 | 1298 | 2 | CQPD, 2000 |
| **Yutubao** | 40.75 | 112.67 | 1254 | 7 | CQPD, 2000 |
| **Zhujiang delta K5** | 22.78 | 112.63 | 12 | 1 | CQPD, 2000 |
| **Da-7** | 40.52 | 112.62 | 1200 | 3 | CQPD, 2000 |
| **Hahai-1** | 40.17 | 112.50 | 1200 | 5 | CQPD, 2000 |
| **Wajianggou** | 40.50 | 112.50 | 1476 | 4 | CQPD, 2000 |
| **Shuidong Core A1** | 21.75 | 111.07 | -8 | 2 | CQPD, 2000 |
| **Dajahu** | 31.50 | 110.33 | 1700 | 2 | CQPD, 2000 |
| **Tianshuigou** | 34.87 | 109.73 | 360 | 7 | CQPD, 2000 |
| **Mengjiawan** | 38.60 | 109.67 | 1190 | 7 | CQPD, 2000 |
| **Fuping BK13** | 34.70 | 109.25 | 422 | 7 | CQPD, 2000 |
| **Yaocun** | 34.70 | 109.22 | 405 | 2 | CQPD, 2000 |
| **Jinbian** | 37.80 | 108.60 | 1400 | 4 | CQPD, 2000 |
| **Dishaogou** | 37.83 | 108.45 | 1200 | 2 | CQPD, 2000 |
| **Shuidonggou** | 38.20 | 106.57 | 1200 | 5 | CQPD, 2000 |
| **Jiuzhoutai** | 35.90 | 104.80 | 2136 | 7 | CQPD, 2000 |
| **Luojishan** | 27.50 | 102.40 | 3800 | 1 | CQPD, 2000 |
| **RM-F** | 33.08 | 102.35 | 3400 | 2 | CQPD, 2000 |
| **Hongyuan** | 33.25 | 101.57 | 3492 | 1 | CQPD, 2000 |
| **Wasong** | 33.20 | 101.52 | 3490 | 1 | CQPD, 2000 |
| **Guhu Core 28** | 27.67 | 100.83 | 2780 | 7 | CQPD, 2000 |
| **Napahai Core 34** | 27.80 | 99.60 | 3260 | 2 | CQPD, 2000 |
| **Lop Nur** | 40.50 | 90.25 | 780 | 7 | CQPD, 2000 |
| **Chaiwobao1** | 43.55 | 87.78 | 1100 | 2 | CQPD, 2000 |
| **Chaiwobao2** | 43.33 | 87.47 | 1114 | 1 | CQPD, 2000 |
| **Manasi** | 45.97 | 84.83 | 257 | 2 | CQPD, 2000 |
| **Wuqia** | 43.20 | 83.50 | 1000 | 7 | CQPD, 2000 |
| **Madagou** | 37.00 | 80.70 | 1370 | 2 | CQPD, 2000 |
| **Tongyu** | 44.83 | 123.10 | 148 | 5 | CQPD, 2000 |
| **Nanjing** | 32.15 | 119.05 | 10 | 2 | CQPD, 2000 |
| **Banpo** | 34.27 | 109.03 | 395 | 1 | CQPD, 2000 |
| **QL-1** | 34.00 | 107.58 | 2200 | 7 | CQPD, 2000 |
| **Dalainu** | 43.20 | 116.60 | 1290 | 7 | CQPD, 2000 |
| **Qinghai** | 36.55 | 99.60 | 3196 | 2 | CQPD, 2000 |






**1135 Table 2. Earth's orbital parameters and trace gases as recommended by the PMIP3**
**1136 project**

| Simulation | Orbital parameters | | Trace gases | | | |
|---|---|---|---|---|---|---|
| | Eccentricity | Obliquity(°) | Longitude of the perihelion(°) | $CO_2$(ppmv) | $CH_4$(ppbv) | $N_2O$(ppbv) |
| **PI** | 0,016724 | 23,446 | 102,04 | 280 | 760 | 270 |
| **MH** | 0,018682 | 24,105 | 0,87 | 280 | 650 | 270 |



**1139 Table 3. PMIP3 model characteristics and references**

| Model Name | Modelling centre | Type | Grid | Reference |
|---|---|---|---|---|
| *BCC-CSM-1-1* | BCC-CMA (China) | AOVGCM | Atm: 128×64×L26; Ocean: 360×232×L40 | Xin et al. (2013) |
| *CCSM4* | NCAR (USA) | AOGCM | Atm: 288 × 192×L26; Ocean: 320×384×L60 | Gent et al. (2011) |
| *CNRM-CM5* | CNRM&CERFACS (France) | AOGCM | Atm: 256 × 128×L31; Ocean: 362×292×L42 | Voldoire et al. (2012) |
| *CSIRO-Mk3-6-0* | QCCCE, Australia | AOGCM | Atm: 192 × 96×L18; Ocean: 192×192×L31 | Jeffrey et al. (2013) |
| *FGOALS-g2* | LASG-IAP (China) | AOVGCM | Atm: 128 × 60×L26; Ocean: 360×180×L30 | Li et al. (2013) |
| *FGOALS-s2* | LASG-IAP (China) | AOVGCM | Atm: 128 × 108×L26; Ocean: 360×180×L30 | Bao et al. (2013) |
| *GISS-E2-R* | GISS (USA) | AOGCM | Atm: 144 × 90×L40; Ocean: 288×180×L32 | Schmidt et al. (2014a,b) |
| *HadGEM2-CC* | Hadley Centre (UK) | AOVGCM | Atm: 192 × 145×L60; Ocean: 360×216×L40 | Collins et al. (2011) |
| *HadGEM2-ES* | Hadley Centre (UK) | AOVGCM | Atm: 192 × 145×L38; Ocean: 360×216×L40 | Collins et al. (2011) |
| *IPSL-CM5A-LR* | IPSL (France) | AOVGCM | Atm: 96 × 96×L39; Ocean: 182×149×L31 | Dufresne et al. (2013) |
| *MIROC-ESM* | Utokyo&NIES (Japan) | AOVGCM | Atm: 128×64×L80; Ocean: 256×192×L44 | Watanabe et al. (2011) |
| *MPI-ESM-P* | MPI (Germany) | AOGCM | Atm: 196×98×L47; Ocean: 256×220×L40 | Giorgetta et al. (2013) |
| *MRI-CGCM3* | MRI (Japan) | AOGCM | Atm: 320 × 160×L48; Ocean: 364×368×L51 | Yukimoto et al. (2012) |




**Table 4. Important values for each plant life form used in the ΔV statistical calculation as assigned to the megabiomes**

| Megabiomes | Life form | | |
|---|---|---|---|
| | **Trees** | **Grass/grass** | **Bare ground** |
| *Tropical forest* | 1 | | |
| *Warm mixed forest* | 1 | | |
| *Temperate forest* | 1 | | |
| *Boreal forest* | 1 | | |
| *Grassland and dry shrubland* | 0.25 | 0.75 | |
| *Savanna and dry woodland* | 0.5 | 0.5 | |
| *Desert* | | 0.25 | 0.75 |
| *Tundra* | | 0.75 | 0.25 |


**Table 5. Attribute values and the weights for plant life forms used by the ΔV statistic**

| Life form | Attribute | | | |
|---|---|---|---|---|
| ***Trees*** | Evergreen | Needle-leaf | Tropical | Boreal |
| *Tropical forest* | 1 | 0 | 1 | 0 |
| *Warm mixed forest* | 0.75 | 0.25 | 0 | 0 |
| *Temperate forest* | 0.5 | 0.5 | 0 | 0.5 |
| *Boreal forest* | 0.25 | 0.75 | 0 | 1 |
| *Grassland and dry shrubland* | 0.75 | 0.25 | 0.75 | 0 |
| *Savanna and dry woodland* | 0.25 | 0.75 | 0 | 0.5 |
| *weights* | 0.2 | 0.2 | 0.3 | 0.3 |
| ***Grass/Shrub*** | Warm | Arctic/alpine | | |
| *Grassland and dry shrubland* | 1 | 0 | | |
| *Savanna and dry woodland* | 0.75 | 0 | | |
| *Desert* | 1 | 0 | | |
| *Tundra* | 0 | 1 | | |
| *weights* | 0.5 | 0.5 | | |
| ***Bare Ground*** | Arctic/alpine | | | |
| *Desert* | 0 | | | |
| *Tundra* | 1 | | | |
| *weight* | 1 | | | |

**Table 6. Regression coefficients between the reconstructed climates by inverse vegetation models and observed meteorogical values**

| Climate parameter | Slope | Intercept | R | ME | RMSE |
|---|---|---|---|---|---|
| **MAT** | 0..82±0..02 | 0.92±0.18 | 0.89 | 0.16 | 3.25 |
| **MTCO** | 0.81±0.01 | -1.79±0.18 | 0.95 | -0.17 | 3.19 |
| **MTWA** | 0.75±0.03 | 4.57±0.60 | 0.75 | -0.19 | 4.02 |
| **MAP** | 1.15±0.02 | 32.90±18.41 | 0.94 | 138.01 | 263.88 |
| **Pjan** | 1.01±0.02 | 0.32±0.47 | 0.94 | 0.52 | 8.89 |
| **Pjul** | 1.30±0.03 | -21.67±4.52 | 0.89 | 16.45 | 52.9 |

The climatic parameters used for regression are the actual values (data source: China Climate Bureau, China Ground Meteorological Record Monthly Report, 1951-2001). MAT annual mean temperature, MTCO mean temperature of the coldest month, MTWA mean temperature of the warmest month, MAP annual precipitation, RMSE the root-mean-square error of the residuals, ME mean error of the residuals, Pjan: precipitation of January, Pjul: precipitation of July, R is the correlation coefficient, ± stand error

**Figure 1.** Distribution of pollen sites during mid-Holocene period in China. Black circle is the original China Quaternary Pollen Database, red circles are digitized ones from published papers, green circles represent the three original pollen data used in this study. The area with green color represents the Tibetan Plateau, yellow color for the Loess Plateau.
























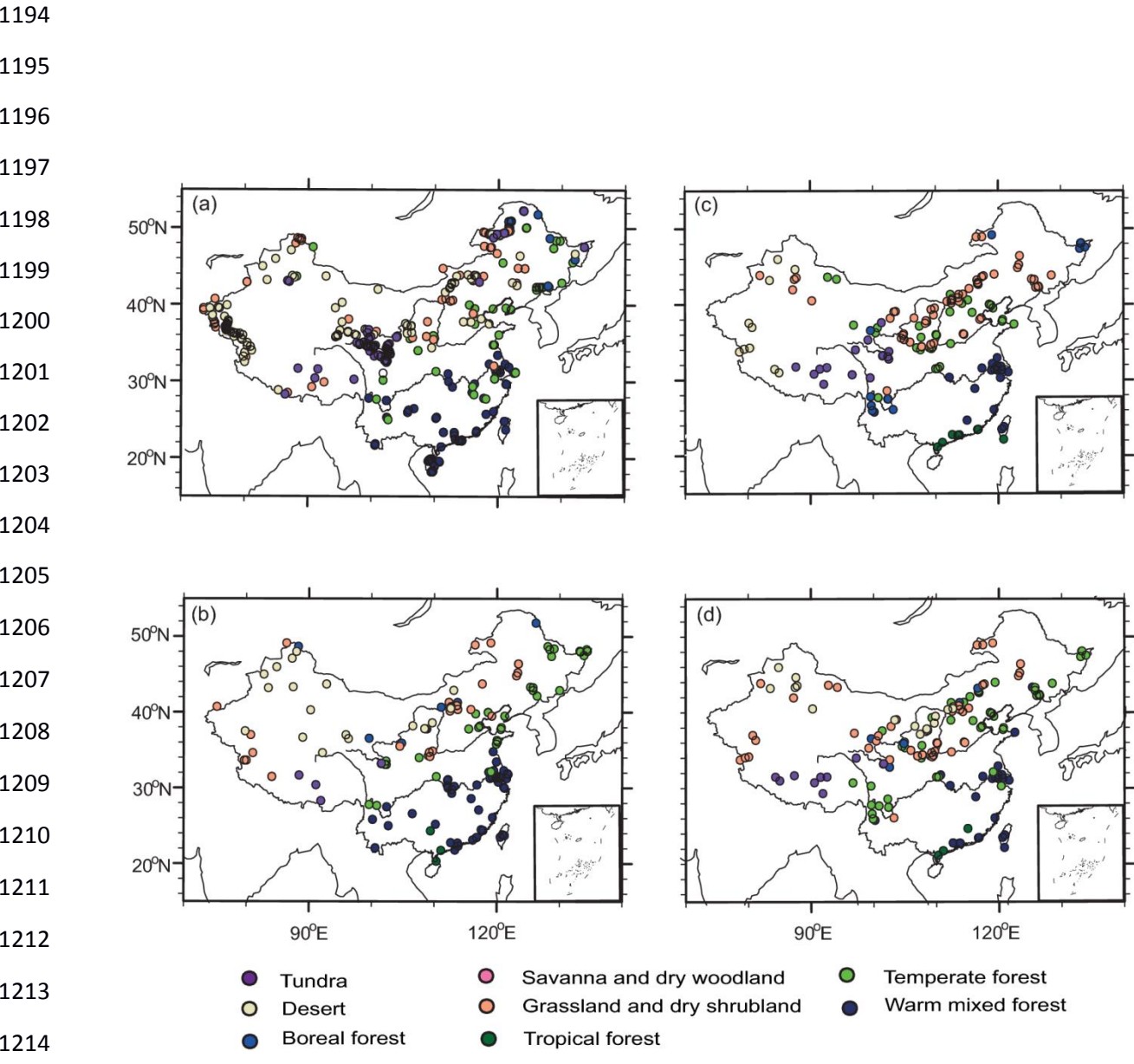

**Figure 2.** Comparison of megabiomes for PI (first row) and the MH (second row): (a,b)
BIOME6000, (c,d) pollen data collected in this study.







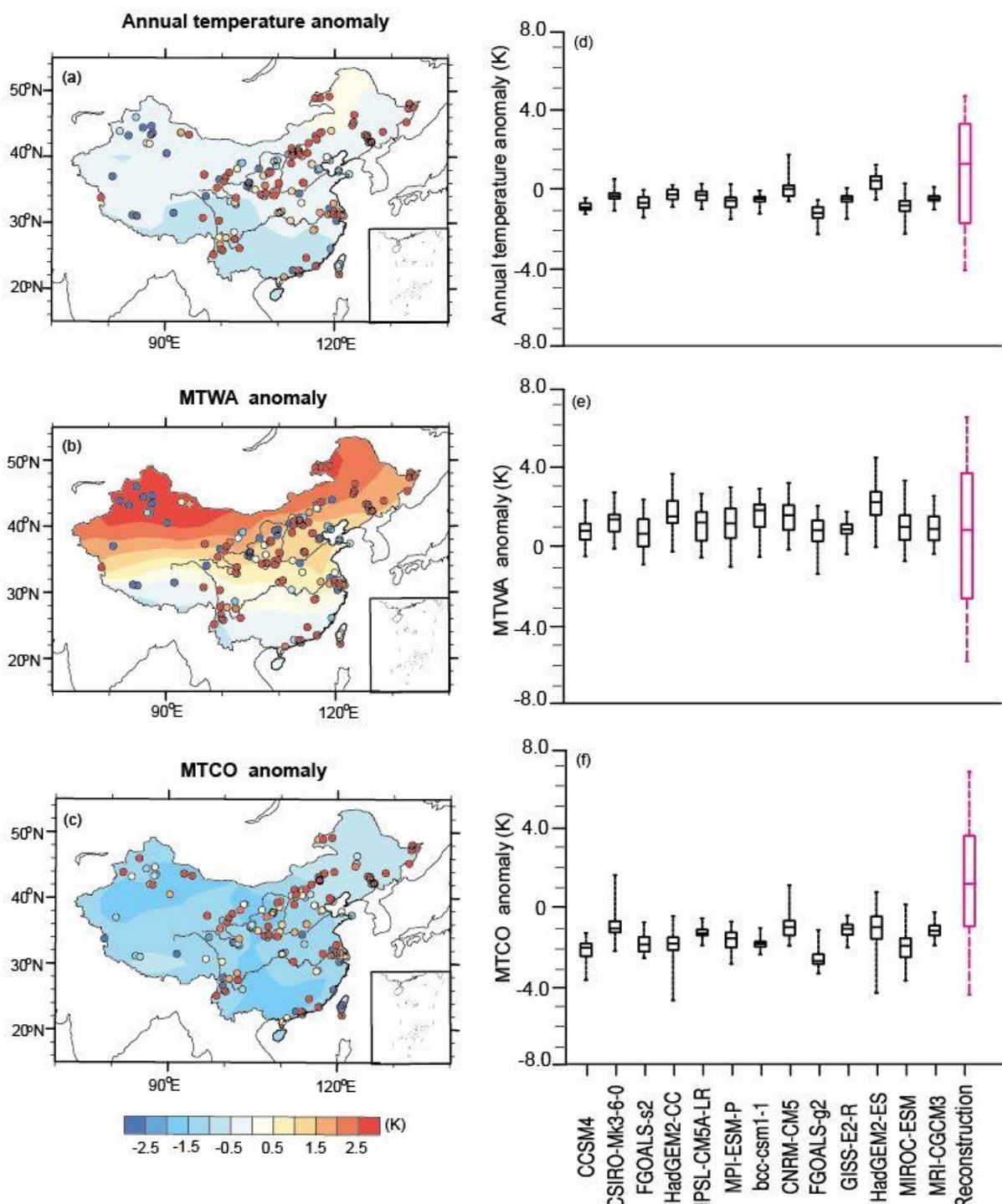


**Figure 3.** Model-data comparison for annual and seasonal (MTWA and MTCO) temperature
(K). For the left panel (a-c), points represent the reconstruction from IVM, shades show the
last 30-year means simulation results of multi-model ensemble (MME) for 13 PMIP3 models.
The box-and-whisker plots (d-f) show the changes as shown by each PMIP3 model and the
reconstruction. (d) considers changes in annual temperature, (e) indicates changes in MTWA,
and (f) shows changes in MTCO. The lines in each box shows the median value from each set
of measurements, the box shows the 25%-75% range, and the whiskers show the 90% interval
(5th to 95th percentile).

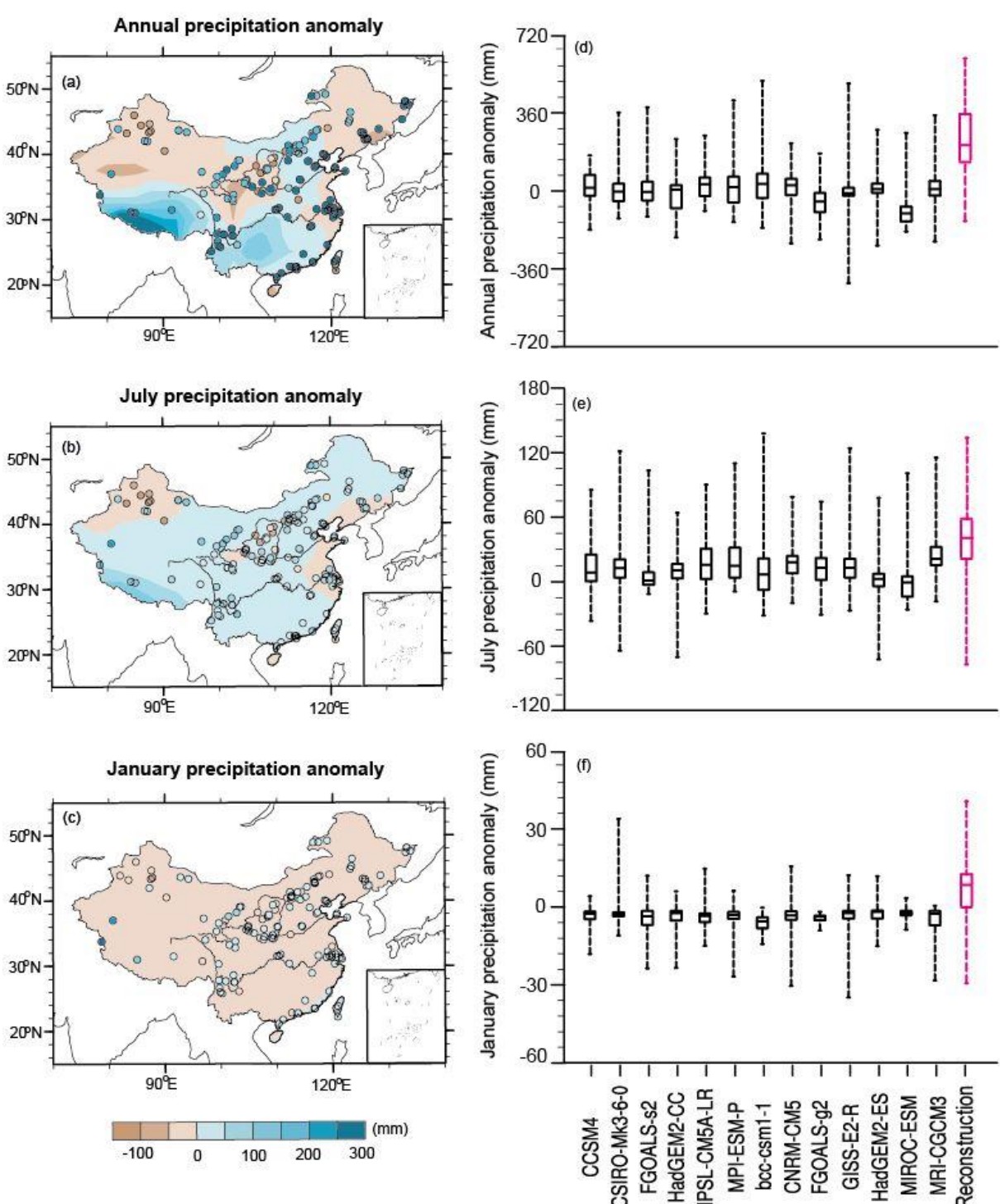


**Figure 4.** Model-data comparison for annual, July and January precipitation (mm). For the
left panel (a,b), points represent the reconstruction from IVM, shades show the last 30-year
means simulation results of multi-model ensemble (MME) for 13 PMIP3 models. The box-
and-whisker plots (d-f) show the changes as shown by each PMIP3 model and the
reconstruction. (d) considers changes in annual precipitation, (e) indicates changes in July
precipitation, and (f) shows changes in January precipitation. The lines in each box shows the
median value from each set of measurements, the box shows the 25%-75% range, and the
whiskers show the 90% interval (5th to 95th percentile).

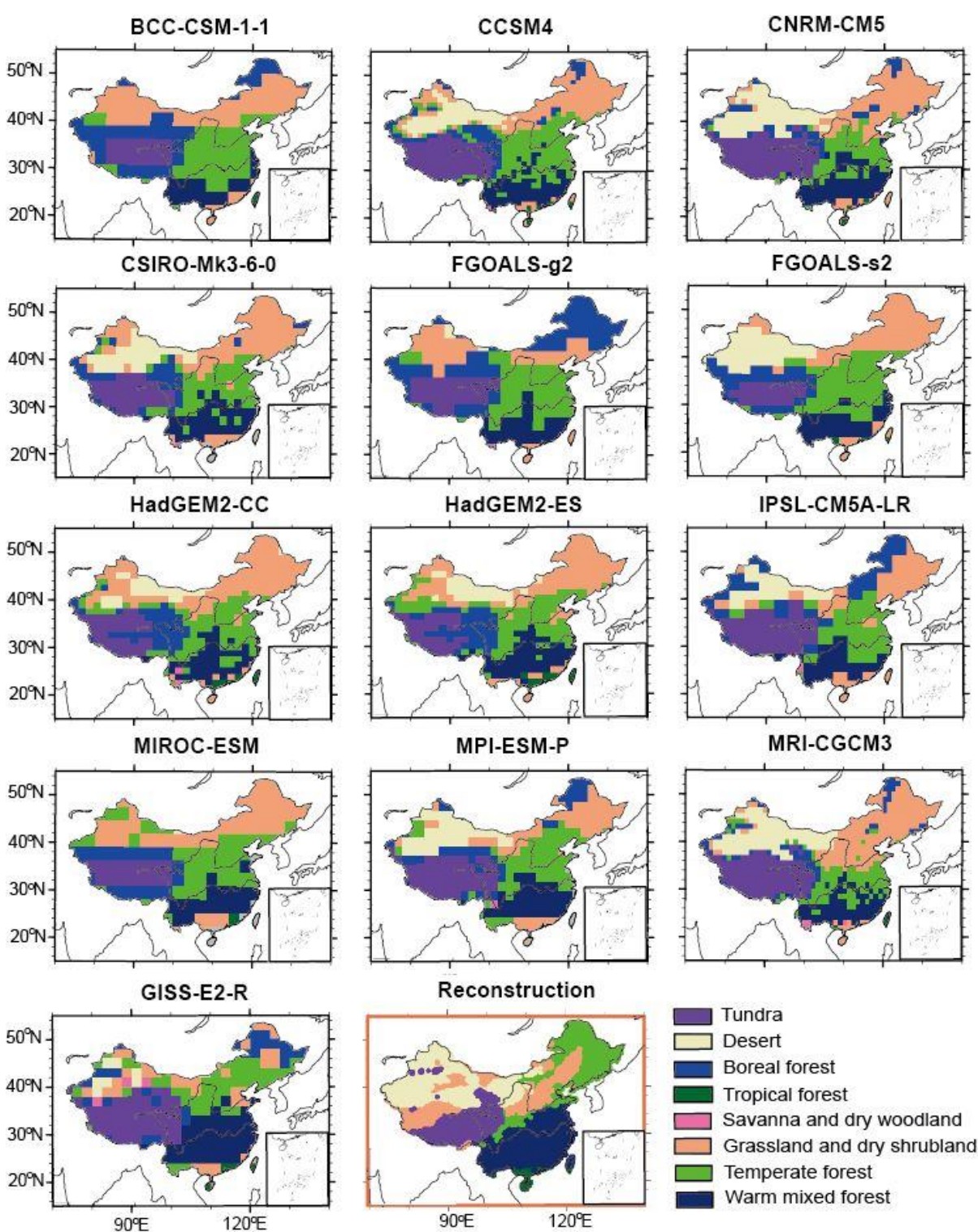


**Figure 5.** Comparison of interpolated megabiomes distribution (plot in red rectangle) with the simulated spatial pattern from BIOME4 for each model during mid-Holocene.






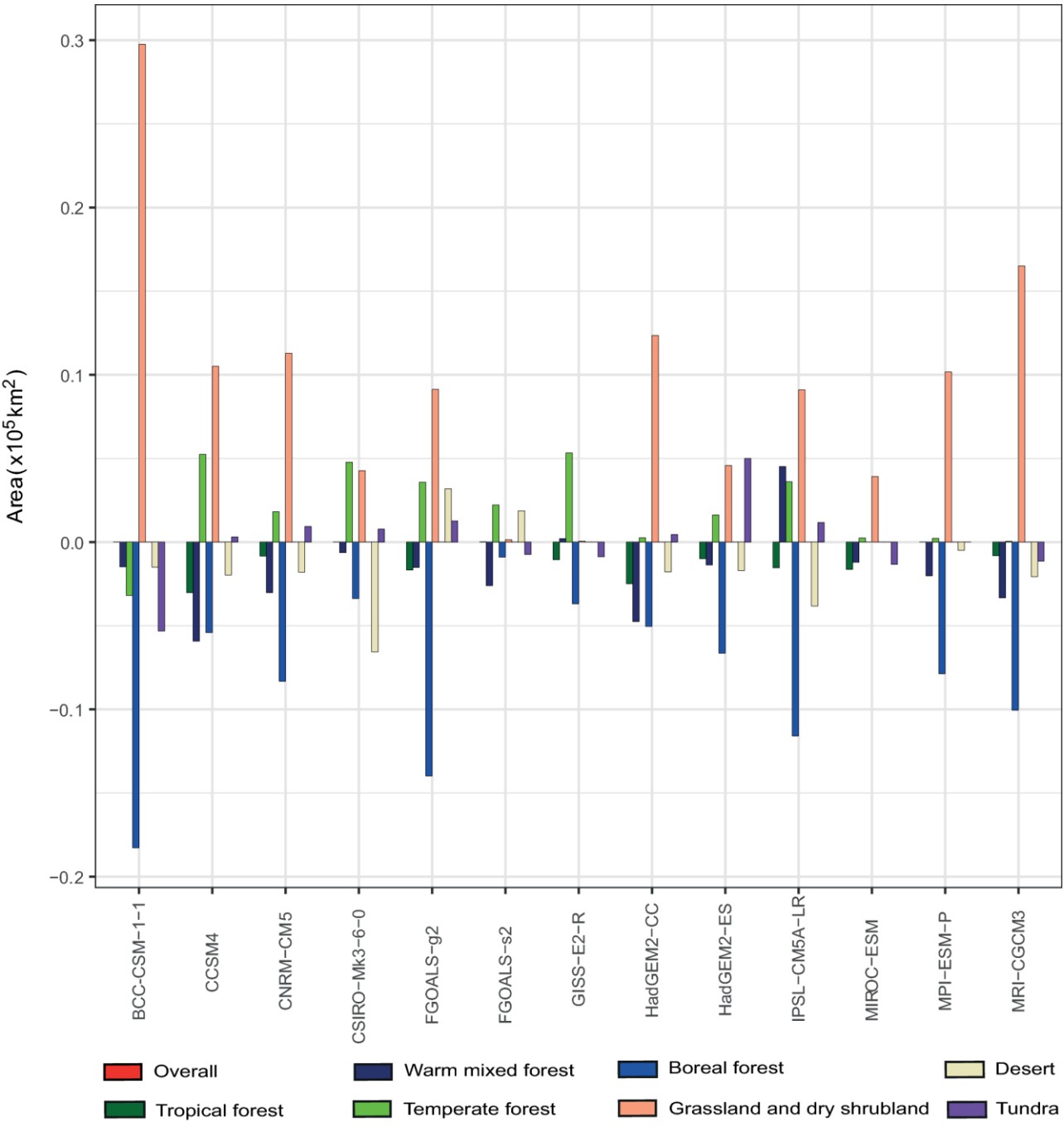

**Figure 6.** Changes in the extent of each megabiome as a consequence of simulated climate changes for each model, both expressed as change relative to the PI extent of same megabiome.

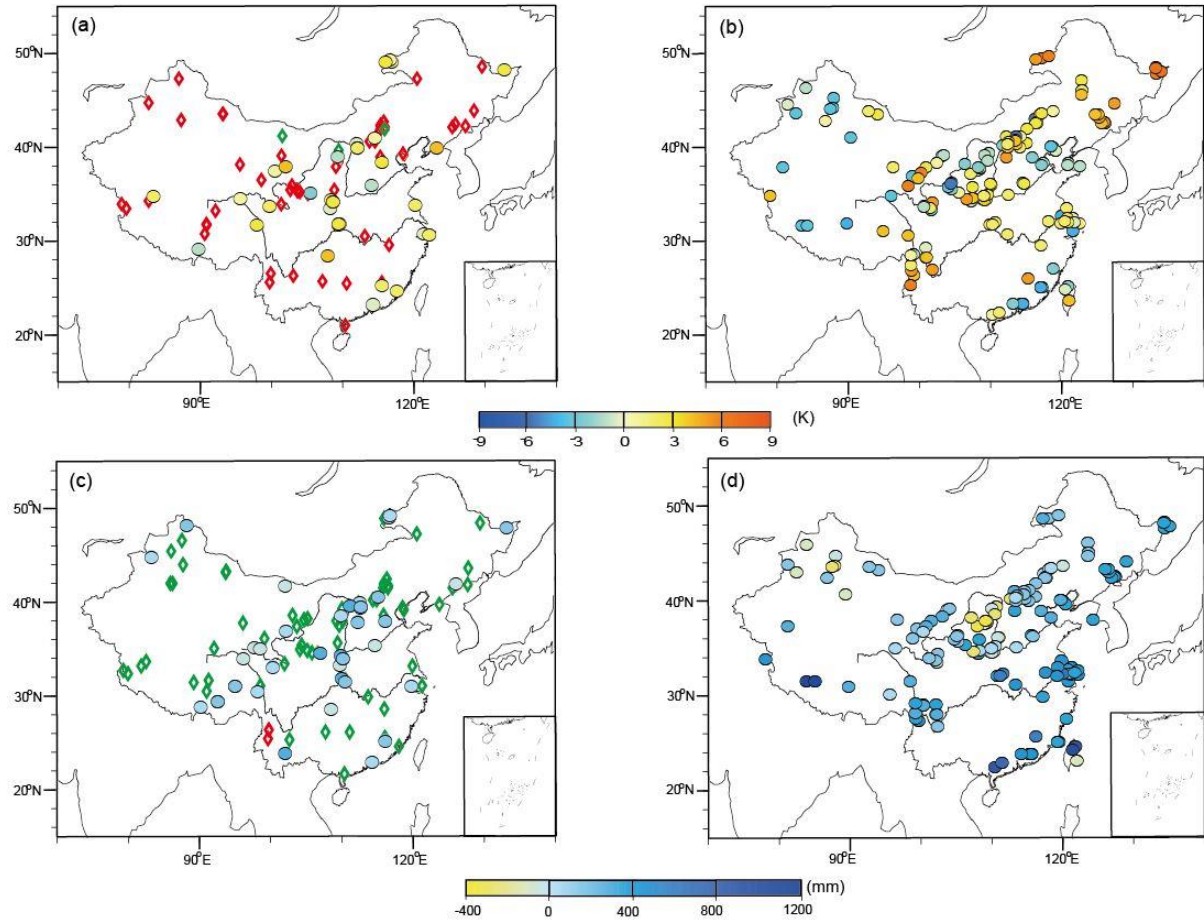

**Figure 7**. Comparison between the climate reconstruction and previous reconstruction over China. (a) Previous temperature results. Diamond is the qualitative reconstruction, red is the temperature increase and green is the temperature decrease; Circle is quantitative reconstruction; (b) Mean annual temperature reconstruction in this study; (c) Previous precipitation results, diamond is the qualitative reconstruction, red is the precipitation decrease and green is the precipitation increase; Circle is the quantitative reconstruction; (d) Mean annual precipitation reconstruction in this study.




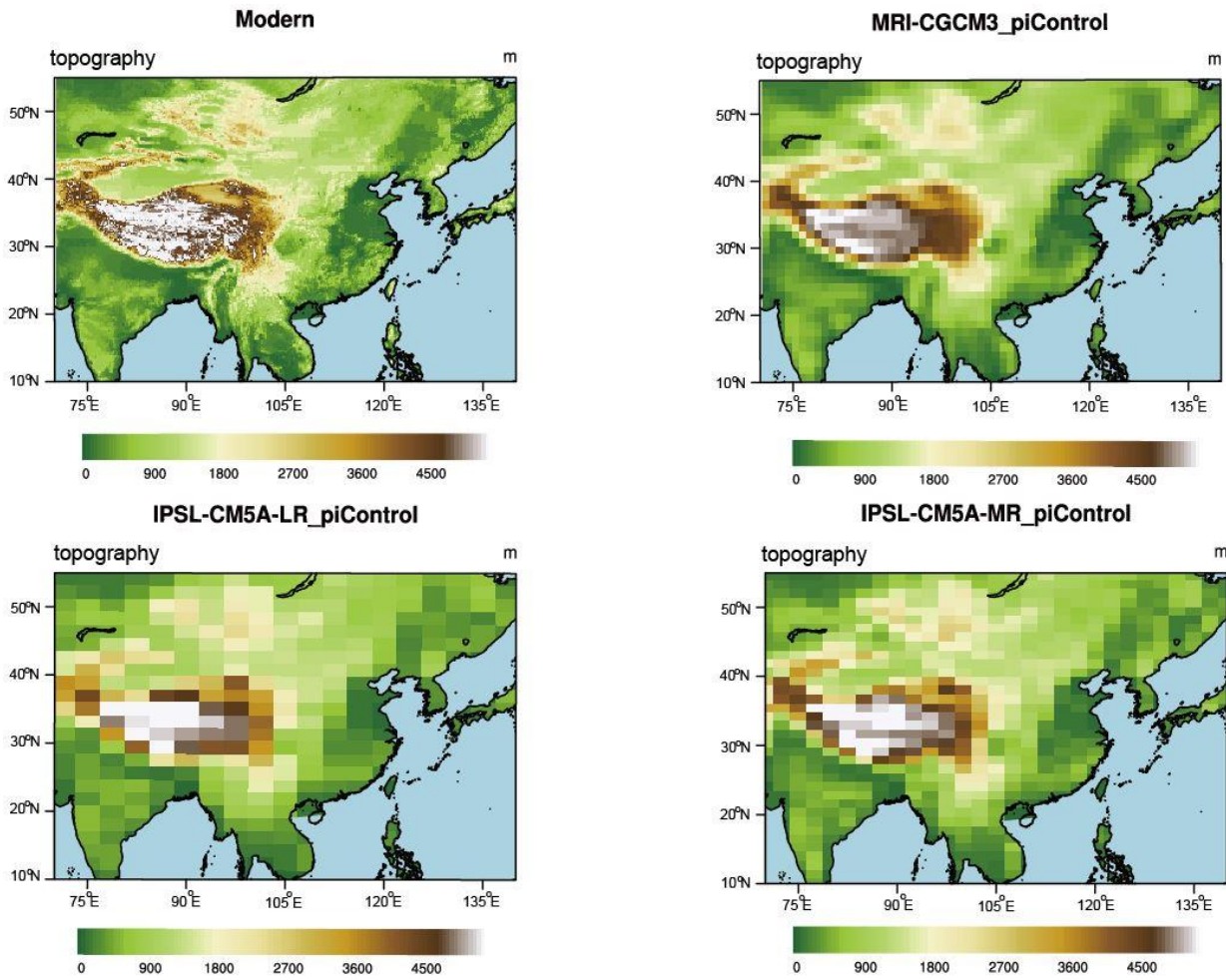



**Figure 8**. The topography comparison between models and observation.









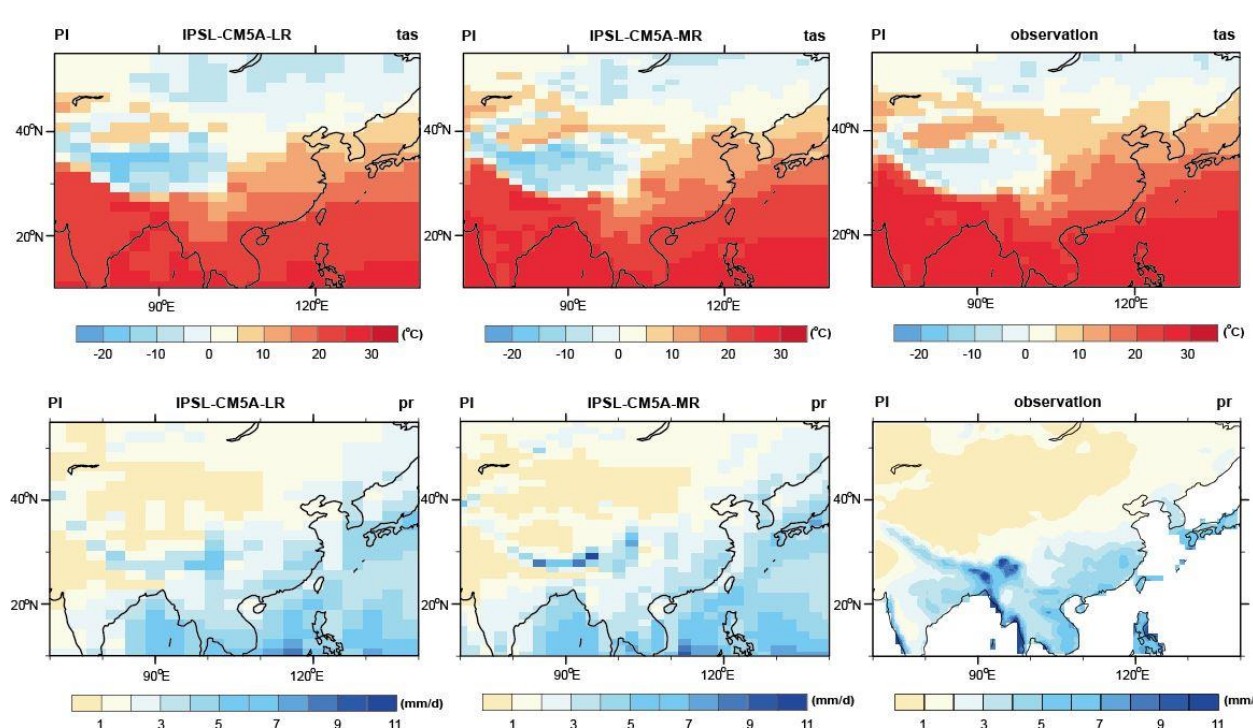

**Figure 9**. The preindustrial climate comparison between simulation and observation. Tas
means temperature above 2m surface, pr means precipitation.

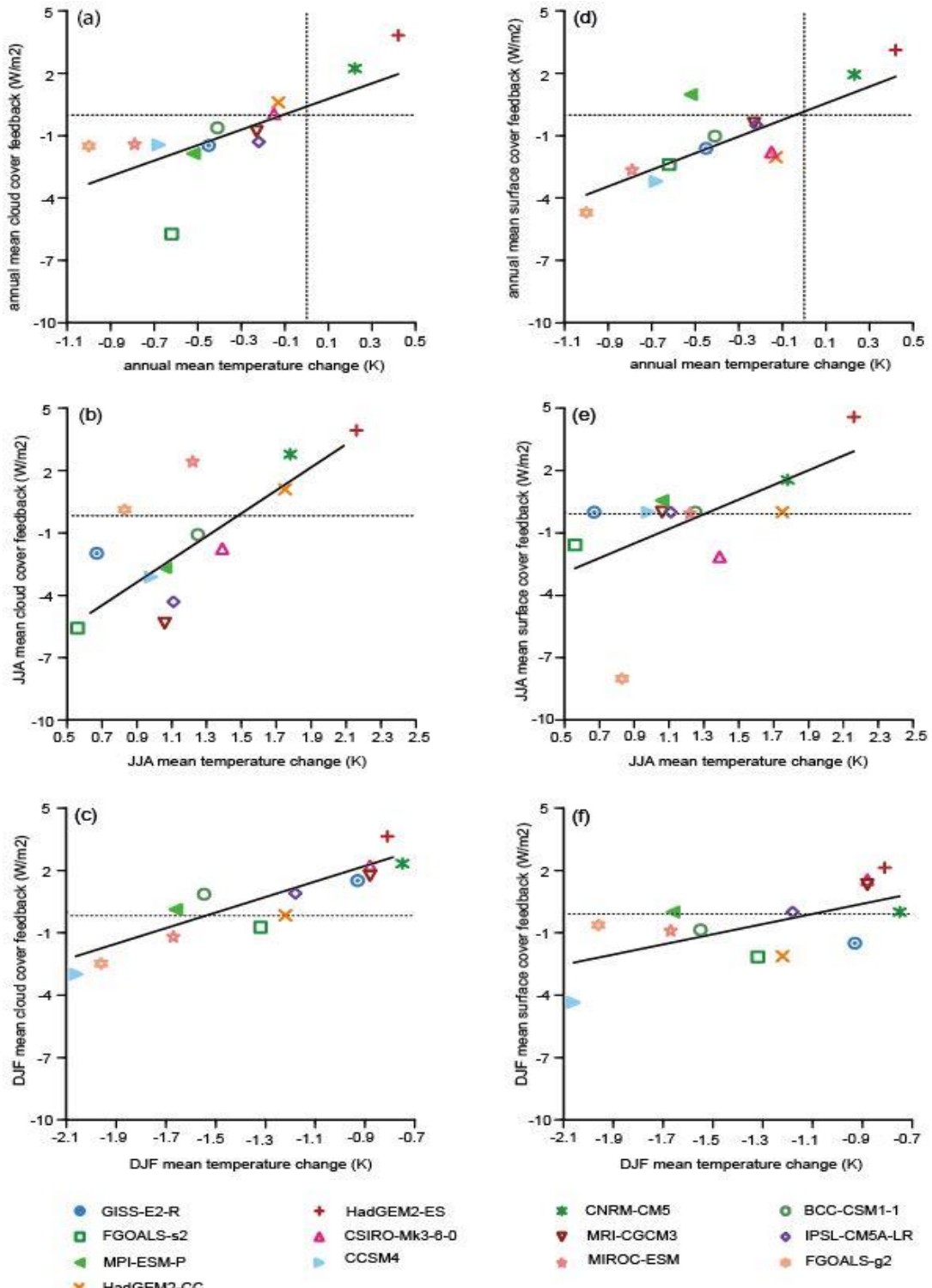


**Figure 10.** Scatter plots showing temperature, cloud cover feedback and surface albedo
feedback changes during the MH. The values shown are the simulated 30-year mean anomaly
(MH-PI) for the 13 models. **a**, annual mean temperature relative to the annual mean cloud
cover feedback and d, annual surface albedo feedback. b, Summer (JJA) mean temperature
relative to the summer mean cloud cover feedback and **e**, Summer surface albedo feedback.
**c**, Winter (DJF) mean temperature relative to the summer mean cloud cover feedback and **f**,
Winter surface albedo feedback. The horizontal and vertical lines in plots represent the value
of 0.

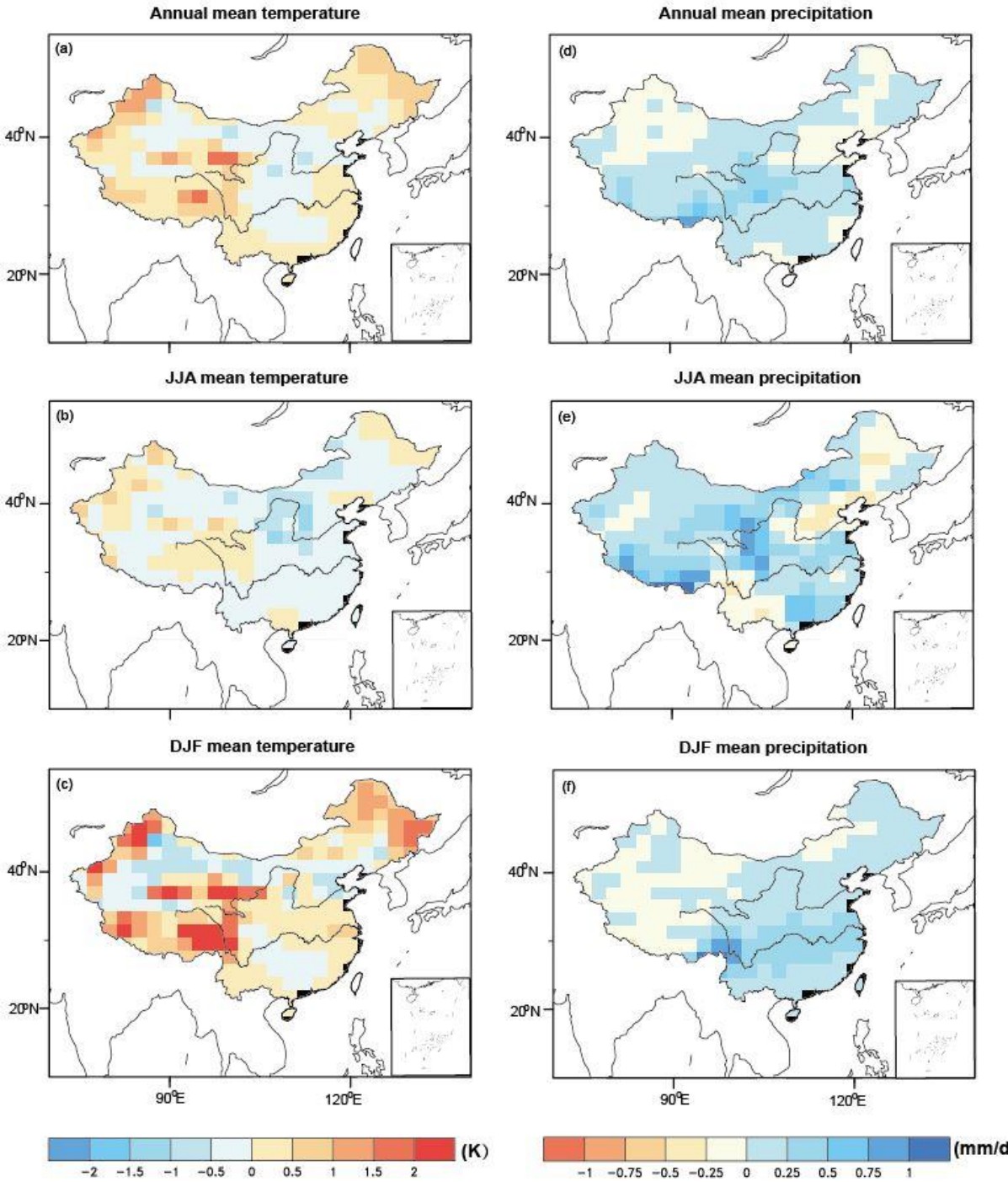


**Figure 11.** Climate anomalies between the two experiments (6 ka and 6 ka_VEG) conducted
in CESM version 1.0.5. The anomalies (6 ka_VEG-6 ka) of temperature and precipitation at
both annual and seasonal scale are presented, and all these climate variables are calculated as
the last 50-year means from two simulations.


