# Peer review of "Mid-Holocene Climate Change over China: Model-Data Discrepancy"

_Climate of the Past, 2018_

## Editor Comment (EC1) · M.F. Loutre (Editor) · 20 Nov 2018

Climate of the Past, as well as all the Copernicus journals recognize the essential importance of data sets, model code, samples, etc. as output of the research. They also insisted in FAIR (findable, accessible, interoperable, and reusable) data. Along that line we invite the authors to provide essential additions to their paper.

1. Archiving data

Input data –

\* Ideally, the pollen assemblages for each of the 161 sites, at least for the 6 ka should be made available (e.g. in trusted archives). The biographic reference must be included for each of the sites. Many citations in Table 1 are not listed in the reference list.

[Figure]

\* All the original data must be archived in a vetted repository and a citation should be linked to them.

\* climate simulations: the citations in Table 3 are not listed in the reference list. This should be amended.

Output data – the data used to draw figures 2-4 should be archived. In particular

\* all reconstructed climate variables and the biomes for each of the 161 sites at 6 ka (the values for the colored dots in Figs 2-4)

\* the simulated values for each of the climate variables (the colored bars in Figs. 2-4)

\* basic metadata for each site (the information listed in Table 1)

2. The "data availability" section

It should include the doi/url that links to the output data from this study and the reiterates the location of the input data (above). Alternatively, location of the input data can be provided in the table.

3. Of course, the authors might also want to make the digital data for their maps available, or anything that they think might be useful for future studies.

A special attention to the reference list is required. As it stands many references are missing (red in the attached file). The authors should also check lines 483, 637 and 694 of their reference list. Is CQPD 2000 or 2001 or both?

Please also note the supplement to this comment:
https://www.clim-past-discuss.net/cp-2018-145/cp-2018-145-EC1-supplement.pdf

**Supplement:**

[Figure]

717    **Table 1. Basic information of the pollen dataset used in this study**

[revised manuscript text omitted]

---

## Referee Comment (RC1) · Anonymous Referee #1 · 30 Nov 2018

Review of CP-2018-145 entitled "Mid-Holocene climate change over C 1 hina: model-data discrepancy" by Lin et al. The model-data comparison of climate change during mid-Holocene (MH) is an important issue to validate the results from Global Circulation Model (GCM) against the proxies gathered from dataset. Based on the new pollen dataset and Inverse Vegetation Model (IVM),This study provideda quantitative reconstruction of climate variables during MH over China was provided and compared to the simulation results from 13 models in PMIP3. A large discrepancy on the temperature anomaly between model-data at both annual and seasonal scale was depicted, mainly due to the failure of capturing vegetation change during MH by models, which is very helpful for better understanding the climatic changes during MH, and also pinpoints the possible way to reconcile model and data by accurately simulating the non-linear

responses of vegetation and hydrology in GCMs. The manuscript can be accepted for publication after minor revision. A few basic comments and some issues to deal with as follow: 1. Since it's a quantitative model-data comparison based on pollen dataset, in which 91 records were digitized from published papers. More detailed information about the data should be provided, like the age control, pollen assemblages from around 6 ka at each site. 2. As mentioned in the manuscript, there is a difference in vegetation inputs for the MH period among models in PMIP3, a table for detail information should be given. 3. The disparity of temperature anomaly during MH among models could be resulted from the difference in pre-industrial (PI) simulation. Authors should prove that there is no any clear relationship between PI temperature and temperature change (MH-PI). 4. Some references are missing in the reference list. such as the citations in Table 3.

---

## Referee Comment (RC2) · Anonymous Referee #2 · 10 Dec 2018

Data-model comparison is often problematic especially at regional scale, for which there are many reasons. This paper presents an interesting analysis to investigate the possible impact of poor representation of vegetation in climate models on the model-data discrepancy over China. The authors compare the PMIP3 results with their "reconstruction" and propose that lack of vegetation dynamics is the main reason of model-data discrepancy in seasonal climate over China. The results are clearly explained and the paper is well written. Especially, a large amount of data are presented and would be a remarkable contribution to the Holocene study. I would recommend its publication after the following comments are considered:

1. At least to my knowledge, regional diversity exists inside China regarding the timing of the mid-Holocene thermal maximum. However, the insolation of 6 ka BP is used in

the PMIP3 simulations. In which degree might the model-data discrepancy be related to the forcing used in the climate models? Have the authors compared their reconstructions to simulations of other periods like 9 ka and 12 ka or to transient simulations to see whether the data-model comparison can be improved if different forcing is considered?

2. One way to test the proposal of the authors would be prescribing the reconstructed vegetation in a climate model to see how the model results would be altered and whether the model would reproduce more realistic results when compare to other proxy data.

3. The spatial resolution of all the GCMS is very coarse when regional diversity within China is considered. The regional details of topography are not necessarily well represented. I wonder in which degree the model-data mismatch is related to rough topography used in the climate models.

4. Line 372: the authors consider the poor capacity of vegetation modelling in climate models to be the major reason for model-data discrepancy. Before the author test for other reasons like those related to topography, soil types, selected climate forcings, I am not very convinced that vegetation is the major reason.

5. Two models of 13 use dynamical vegetation model. According to your analyses, is there obvious advantage of using AOV instead of AO?

6. Line 36: do you mean " an increase in the seasonal cycle of insolation"

7. I wonder how the pollen data of PI was collected. Were they collected from the surface? Is there any influence from human activities?

8. In Figure 3: is the anomaly relative to PI? How was the grid mean value calculated?

9. Line 320: PMIP3

---

## Referee Comment (RC3) · Anonymous Referee #3 · 19 Dec 2018

The manuscript entitled "Mid-Holocene climate change over China: model-data discrepancy" by Lin et al. presented a study on model-data comparison by using the pollen data collection in China and PMIP3 mid-Holocene simulations. From the large discrepancy showed in model-data comparison, both in annual mean, warmest month and coldest month, they conclude that the major reason that PMIP3 simulations do not agree with data is because the vegetation distribution is not properly represented in climate models, where most models do not include dynamical vegetation and the prescribed MH vegetation map is the same as preindustrial. The MH vegetation issues have been recognised in recent years and many efforts are made to reconstruct a better MH land cover map, this includes the PAGES working group on Landcover6k. Therefore a good vegetation map from China would be expected to contribute to an

eventual global land-cover map during the mid-holocene and benefit the paleoclimate community. However, the current work has a somewhat mislead focus and I have the following major concerns.

General comments:

1. The reconstructed mid-Holocene climate in their study is largely depend on the pollen data collection. I am not an expert on pollen data, but I am wondering if all the published data use the same standard on data process. Can they be synthesised by Webb1-7 standard and put together for comparison? I hope a reviewer from pollen community may have some insights on the data process. There are no discussion on the potential uncertainties on collected data, at least one comparison with other proxy data can provide the cross-proxy verification. The authors emphasised three original data but no detailed information, which are important if they are not published. When the significant differences are found in model-data comparisons, the uncertainties from the data should be discussed as well. One can not regard reconstruction is the truth. We need to know how reliable is the reconstructed climate from pollen data, given that the IVF method used to reconstruct the climate is a crude estimate. Otherwise it is dangerous if this paper is published and people take for granted that this is the climate (and vegetation map) in China during mid-Holocene.

2. The BIOME4 produced vegetation pattern in fig5 is determined by the input climate variables from the model, given the supplementary figures s1-s6 and previous studies by Jiang et al. (2012) have already show different climate patterns produced by different models, therefore the mismatch in vegetation pattern and reconstructed map in Fig5 is expected. I don't think this mismatch can be used to argue that the modelled MH climate is not good because they did not use a correct vegetation map and include the vegetation-climate interaction. Those vegetation patterns produced by BI-MOE4 are not used in PMIP experiment setup, it would make more sense if authors compare the reconstruction and PMIP prescribed landcover map, or compare BIOME4 produced vegetation map with the ones produced by those climate models (for example HadGEM2-ES) that have dynamical vegetation to gain some understanding on vegetation-climate feedback.

Specific comments:

1. The abstract need to provide more information from this work, now only contains motivation and conclusion. And the conclusion in abstract actually is a speculation, did not come from the results of this work.

2. Take line 49 as an example, 0.5K should be write as 0.5 K, follow SI standard, there is a space between number and unit. May correct throughout the manuscript.

3. Line 116, "The new sites", if it is new, the data information should be described, otherwise they are unknown.

4. LIne 120, what is cloudiness, how are they measured? Because this is not a common variable, should be described.

5. Line 129, how do you determine the anomalies for biome scores? What is the purpose of this paragraph L120-L139, to produce reconstruction in Fig5?

6. Line 143 to Line 147, on description of PMIP is a bit strange, what do you mean "in which the PI experiment was denied". "The main variability between MH and PI" should be "The main forcing between MH and PI".

7. Line 156, "interpolated to a common 2.5 grid", why do you think 2.5 is a common grid, given the pollen data are very local, 2.5 degree grid is too coarse.

8. Line 161-162, How do you obtain the sunshine data from observation and model? Should be described more specific.

9. Line 184, "Weighting the attributes is subjective", will it cause uncertainties?

10. Line 191, from Zhang et al., 2010, the reference can not be found in reference list.

11. I am wondering if the warmest month and coldest month changes between MH

and PI (and between the models), or always July and January? Give there is a change in seasonality in MH, authors should mention this.

12. Line 261, "with a decrease in the northeastern regions", also decrease in east monsoon region at Yangzi river valley.

13. Line 310-312, "this failure to capture ..", see above general comment 2.

14. Line 320, "triggered" is a weird word.

15. Fig 7 on feedback discussion, how do you determine the feedbacks from the cloud cover or surface cover? In Line 356 the authors mentioned the "surface albedo and cloud change are calculated . . .", I don't understand why the changes in forcing can be regarded as feedback, physically it is a climate response to forcing.

16. Line 733, "Importance" should be "Important".

17. Table 6, should give more information for meteorological data, how long, and give which month is the warmest coldest month, in line 742, should be "warmest month".

18. Line 744, "stand error" means "standard deviation"?

19. Figure 1, should you mark your three original data in this map separately?

20. Figure 4, the huge annual precipitation anomaly in reconstruction, how reliable is it? I highly suspect it. The unit for precipitation is mm, does it mean annual 240 mm equal 20 mm/month? I suggest you use mm/month to avoid confusion.

21. Figure 6, Line 848 to 850, why do you give the abbreviation, they are not in the figure.

---

## Referee Comment (RC4) · Bartlein (Referee) · 24 Dec 2018

General comments:

This paper presents an ambitious attempt at comparing simulations from the CMIP5/PMIP3 "midHolocene" archive with a new synthesis of fossil-pollen data for China. The pollen data are used in two ways: 1) to develop a set of quantitative reconstructions of several climate variables using an inverse-modeling approach (to compare with the climate-model output), and 2) to develop a map of "megabiomes" for present and 6 ka (for direct comparison with vegetation simulated by BIOME4 using climate-model output). The paper shows that there is a considerable mismatch between the reconstructed and simulated climates and vegetation. The authors attribute

this mismatch to experimental-design issues, in which vegetation and land-cover data in the climate models were fixed at present-day values, thereby limiting the ability of the climate models to correctly represent the potential impact of vegetation (and surface water- and energy-balance) feedback in the paleo simulations.

I have two reservations about the results and conclusions: First, there is insufficient information on the protocols adopted for generating the both present-day and paleo vegetation, as well as the paleo reconstructions. As for vegetation, Fig. 5 shows that there are large mismatches between the observed and simulated modern vegetation. These would naturally arise if the climate-model output were used directly to simulate the vegetation. We know that at their current resolutions, there is still considerable bias in present-day (or PI) climate simulations, and there is no reason to believe that those bias are the same in paleo simulations (or that they somehow go away). My impression of Fig. 5 and S7 is that those biases in simulated climate are indeed large, possibly swamping the real vegetation change, and so we're not really getting much insight into the nature of the mid-Holocene climate simulations, but instead learning about modern-day bias.

As for reconstructed climate, despite the author's assertion otherwise, there is also considerable bias in the inverse-model reconstruction approach (Table 6)—only for Pjan does the regression between observed and fitted values not differ from one with a slope of 1.0 and an intercept of 0.0. It is not immediately clear how that bias might affect the reconstructed climate, but it reinforces the necessity of looking at the uncertainties in the reconstructions. Again, we may be learning more about the inverse approach than about model-data mismatches.

Second, the attribution of the mismatches to the experimental design of the CMIP5/PMIP3 simulations, while plausible, is not really supported by any direct hypothesis tests, or by the consideration and dismissal of alternative hypotheses. The correlation between the temperature responses and cloud-cover feedback (Figure 7) implicates at least one: inadequate simulation of atmospheric circulation as it may

influence moisture flux or precipitation-generating mechanisms.

I think that if the questions related to the protocol for data-model comparisons are answered, and due consideration given to other possible mechanisms for the mismatch, then the paper will ultimately be publishable.

Specific comments:

Abstract: The abstract fails to disclose conclusions of paper.

Line 14: "proxy reconstructions" Aren't the reconstructions used here "real" reconstructions? I understand the notion of paleoclimatic evidence that can be used as a "proxy" for climate or other phenomena (like land cover). But the reconstructions here are actual reconstructions, not a stand-in or substitute for reconstructions.

Line 18: "continental size" Are you referring to the area of the temperature anomaly or to terrestrial as opposed to marine responses?

Line 20: New definition for PMIP?

Line 22: "a seasonal cycle..."

Line 25: "access surface processes" I don't know what this means.

Line 27: "non-linear process associated with vegetation changes in hydrology and radiative forcing" Does this mean "non-linear responses in hydrology and radiative forcing to vegetation changes"? "Radiative forcing" in the context of the midHolocene experiment is usually reserved for describing the insolation forcing, so an alternative expression might be "non-linear response of the surface water and energy balance to vegetation changes" (which is what I think the paper is arguing for).

Line 34: This definition of the age of the mid-Holocene is inconsistent with what is actually used in the paper (line 101). It might be good to distinguish between the mid-Holocene time slice, and the "midHolocene" CMIP5/PMIP3 experiment, throughout the paper.

Line 36: "an increase in insolation in the seasonal cycle" Replace with "an increase in the amplitude of the seasonal cycle of insolation..."

Line 38: "climate response to changes in the seasonal distribution" It's not the response to the seasonal variations of insolation that you're looking at here, but instead the response to changes in the distribution.

Line 42: "consistency of the dataset incorporating different proxies" I don't know what that means.

Line 45: Again, the data are real, not proxy.

Line 47: "the source of discrepancies..."

Lines 50-51: But see Marsicek et al. (2018, Nature)—the "Holocene conundrum" apparently arose from comparing apples and oranges. A different example might be more convincing.

Line 62: The sheer expanse of the country... Why should the synthesis of paleoclimatic data or simulations necessarily be restricted to political subdivisions? Extending the area of the comparison deeper into the interior of Eurasia would generate a bit more "leverage" in comparing the data and models, but I understand the logic of restricting the analysis to China.

Line 64 (and elsewhere). The article "the" is required before "MH" in this context (i.e. when "MH" is being used as a noun). Elsewhere, as in line 55, where "MH" is used as a modifier of another word ("precipitation" in this context), the article is not used.

Line 66: "warmer and wetter than present..."

Line 73: "colder than the baseline" What baseline? Present-day or preindustrial?

Line 75: "This study" Which study? Reword as "That study..." or more explicitly "Jiang et al. (2013) were the first to point out the model-data discrepancy over China during the MH, but the lack of seasonal reconstructions in their study limits comparisons with

simulations."?

Line 83: Bartlein et al. didn't synthesize land-cover changes.

Lines 86-91: The terminology here needs to be sorted out. The "process-based bio-geographic model" alluded to here is BIOME4, and it is employed in making inferences about past climates using an "inverse modeling through iterative forward modeling" (IMIFM) approach (Guiot et al. 2000; Wu et al., 2007, 2009). (See Izumi and Bartlein, 2016, GRL for further discussion.) So BIOME4 is the vegetation model, while the over-all approach (which employs that model) is "IMIFM" (or after that is all explained, simply "the inverse approach").

Line 91: "In the case of models. . ." Which models? Is it the case that you're evaluating the PMIP3 simulations made with state-of-the-art climate models using reconstructions of temperature and precipitation?

Line 94: "thanks to the seasonal reconstruction" But in all previous applications of the inverse modeling approach using BIOME4 or related models, some sort of recon-struction or estimation of the seasonal variations in climate must have been involved, because BIOME4 requires monthly temperature, precipitation and cloudiness (or sun-shine) data as input.

Lines 95-96: "the forcing factor we used for MH is essential the seasonal change." I think that what's going on here is that the midHolocene CMIP5/PMIP3 experiment is essentially one that looks at the response of the models to changes in the seasonality of insolation, and that you are attempting to derive reconstructions of both summer and winter temperature and precipitation to compare with the simulations.

Line 101: If you're referring to radiocarbon ages, this should be written as 6000 $\pm$ 500 14C yr BP)

Line 102: Spell out "three".

Line 105: I don't understand the notion of "distinct" pollen records. Distinct in the sense

of "unique" or distinct in the sense of "clearly readable"?

Line 107: Criterion 2 seems to be combining two things: sampling resolution and data present within the age range. Please reword.

Line 108: How far?

Line 109: "or by regression"

Line 113: Fix the Webb (1985) citation. (Webb, T. III, etc.)

Lines 110-113: Reorganize sentence to describe the ranking scheme first, and the results second.

Line 114: Add a citation for the concept of "biomization" (which will be a mystery to modelers).

Line 116: CQPD. Not in references. Also "sedimentary" what?

Lines 117-119: Add region names to Fig. 1?

Lines 120-135: There seem to be two tasks described in the paragraph: 1) interpolation of modern climate data (from some unspecified source, and by some by some unspecified approach) to the locations of the pollen data, and 2) interpolation of biome scores onto a regular grid using ANN. I suggest breaking this paragraph up, while providing more information on the first task.

Line 129: If the ANN is calibrated using present-day biomes, then I don't see how it can be used to interpolate anomalies. Or was it the case that present-day and paleo biomes were independently interpolated onto the grid, after which anomalies were calculated.

Lines 142-143: What are "climate anomalies in the present day"?

Line 145: Delete "in which the PI experiment was defined."

Line 146: Here it would be good to refer explicitly to the "midHolocene" experiment.

Line 153: Spell out eight and five.

Line 156: "in order to calculate" These variables could also be calculated on the models' native grids. The motivation for interpolation onto a common grid is simply to get the data onto a common grid.

Line 160: Either delete the hyphens here, or put them into other instances of "biogeography" or biogeochemistry".

Line 162: "sunshine percentage (relative to cloud cover)" I don't know what this means.

Line 164: "main input variables" Are the any others? If not, the variables described are the input variables.

Line 171: Bigelow: not in references.

Line 173: Were the climate variables downscaled in any way (as in the apply-the-anomalies approach, Harrison et al., 1998, J. Climate, Harrison et al., 2014, Climate Dynamics). If not, then the climate fields will not contain the spatial variability of modern climate that in topographically complex areas can have a major impact on vegetation. Fig. S7 attests to the existence of bias in the PI simulations. If the simulated climate values are used directly, than a quantitative estimate of the bias (as in Table 6 for the present-day reconstructions) should be provided.

Line 174: "more than 30 years" How much more? Why not use the same number of years for each model?

Line 176: Replace "model-data discrepancies" with "differences between simulated (by the climate-model output) and reconstructed (from pollen). . ."

Line 183: Replace "estimate" with "describe".

Line 192-194: I'm not sure why you're describing the interpolation of biome data again.

Line 197: "Inverse Vegetation Model" See earlier comments.

Line 208: I'm not sure why CO2 concentrations and soil characteristics are being per-turbed (i.e. estimated by the inverse approach). We know CO2, and earlier you argued that soils were assumed not to differ. Also, Table 3 implies that anomalies (or better put, long-term mean differences between present and past) were iteratively generated, which implies that, as is standard procedure, they were applied to present-day climate values and passed to the biome model. If so, what were those present-day values?

Lines 216-218: I don't know if I'm reading Table 6 correctly, but if I am, the slopes and intercepts are anything but close to 1.0 and 0.0. Only for the case of Pjan is the slope within two standard errors of 1.0, and only for MAP and Pjan is the intercept within two standard errors of 0.0. It would be useful to see scatter diagrams of the observed and estimated values for each variable.

Line 224: The "collected data" is your data set, right? How was the comparison statistic calculated?

Lines 226-239: How are the changes or differences in the reconstructions calculated? As differences between the mid-Holocene reconstructions and present-day observa-tions, or present-day inverse-approach estimates? There is considerable bias in the estimates for the present day (Table 6). How would that contribute to the mismatch between simulations and observations?

Section 3.1 (throughout): No information on the uncertainties of the reconstructions is given. These are customarily obtained from the variability of the "feasible" climate vectors generated in the optimization step in the inverse approach (e.g. Izumi and Bartlein, 2016, Fig. 3). For that matter, there is no information on the spatial variability of the simulations. Uncertainties for both could be displayed by plotting boxplots in Fig. 3, as opposed to bar graphs.

Line 249: "a decreasing trend" Conventionally, trends are described in the sense of a change from one time to another, or the change over a fixed period of time, so here, if the mid-Holocene MTWA values are lower than present, the trend would be positive

or increasing over time (i.e. from the mid-Holocene to present). Check the discussion of precipitation trends too. It would be best to simply drop the notion of "trends" and concentrate on the change between midHolocene and PI.

Line 265: "more detailed information about the geographic distribution of simulated temperature..."

Section 3.1 (overall): It would be interesting to see a comparison for Pjan, the single variable with an intercept of 0.0 and a slope of 1.0.

Line 273: "which would introduce a bias..." That's certainly plausible, but right now it's simply a conjecture.

Line 309: "However, none of the models succeed in capturing these features,..." I agree. However, the differences between the simulated and reconstructed biomes for the midHolocene simulations strike me as apparently similar in magnitude to those for the PI, and casual comparison of Figs. 5 and S7 suggests to me that some of the patterns of disagreement in the midHolocene case are inherited from the PI. This makes me wonder again about the protocol followed for generating the midHolocene simulations (see comment on line 173).

Line 310: What are "enhanced vegetation conditions"?

Line 311: "...a cumulating inconsistency in the model-data comparisons ... because of the vegetation-climate feedbacks." Except for the two AOV models, vegetation-climate feedback is only present in the real, as opposed to simulated, climate, i.e. in the reconstructions.

Line 315-316: "wetter and warmer in MTWA, colder in MTCO" This makes no sense. You might say "higher temperatures in the warmest month of the year," but did you indeed look at precipitation in the warmest month? I think what you want to say is "higher (than present) July precipitation and MTWA, lower than present MTCO" or something like that.

Line 318: Trend again. Data show higher-than-present MTCO during the mid-Holocene while models simulate lower-than-present MTCO.

Line 319: My reading of Fig. 3 shows that CNRM-CM5 and HadGEM2-ES are consistent with all of the other models in simulating lower-than-present MTCO.

Line 322: "among models"

Line 323-324: Replace "shed light" with "raises" (the question). ("Shedding light" implies that the variability referred to would answer the question.)

Line 326: Replace "amplitude" with "amplitude and pattern". (You emphasize pattern as much as area.) Also, it's not the failure of the models to simulate vegetation change that's important, it's the fact that (apart from HadGEM2-ES and HadGEM2-CC) they can't, because the vegetation is not interactive. However, can't albedo still vary, through variations in soil color and snow cover?

Line 337: "Reconstruction showed..." I thought you were talking about the two AOV models. This sentence implies that you estimated the overall albedo change from the vegetation reconstruction, and compared with the two models with interactive vegetation. Is that right? If not, please explain a bit more.

Line 348-349: "should act" or "most likely would act" (We don't really know if it would.)

Line 351-353: It may well be the case that cloud radiative feedback (or rather, inadequate simulation of that) could play a role in the data-model mismatch, but if so, that points to a completely different kind of model inadequacy, involving atmospheric circulation, moisture flux, and cloud-producing or cloud-suppressing mechanisms. Those mechanisms have been implicated in explaining the mismatch between simulations and reconstructions in the Eurasian midcontinent (Bartlein et al., 2017, GRL).

Line 354: Taylor (and fix reference too).

Lines 363-364: "counteract ... while they enhance..." This needs a little explanation,

and a tie into the figure.

Line 374-376: Alternatively, the mid-Holocene vegetation could be specified as part of the experimental design. Unfortunately, this would eliminate using vegetation and vegetation-derived reconstructions as benchmarking targets.

Technical comments:

I concur with the Editor and other referees that some work needs to be done on the references and data-availability aspects of the paper.

References: Format varies from reference to reference.

Tables 4, 5 & 6: Replace commas with periods (decimal points).

Fig. 7: Define dotted horizontal and vertical lines.

Maps (throughout): Why does the "nine-dash line" inset vary in size and shape? I realize that the inset has to be there for geopolitical reasons, but why does it change from map to map?

Fig. S7: What is the white horizontal line?

SI, p. 1: Dallmeyer et al. (2017), not in references.

SI, p. 3: Material at the bottom of the table is hard to read. Please reformat into a table-like arrangement.

SI, p. 5: Add citations to original data sources.

---

## Author Comment (AC1) · 5 Feb 2019

We greatly appreciate the constructive comments and suggestions on the previous version of the manuscript from editor. We have attempted to address every point raised. The following is the outline of the changes we have made, with reference to the order of the comments made by the editor.

Comments from editor: 1. Archiving data Input data Comments: Ideally, the pollen assemblages for each site, at least for the 6ka should be made available (e.g. in trusted archives). The biographic reference must be included for each of the sites. Many citations in Table 1 are not listed in the reference, list.

Response: The biographic reference for each site is included in Table 1 (on pages 42-

45 in revised version), and the citations have been added in the reference list. About the pollen assemblages, we provide the biome type at 6 ka for each pollen site in Table S4 (on pages 5-12 in revised version).

Comments: All the original data must be archived in a vetted repository and a citation should be linked to them.

Response: The original data can't be shared without the permission from authors of them, but we have already give the reference of them in Table 1, full information could be obtained if the readers send request to the authors.

Comments: Climate simulations: the citations in Table 3 are not listed in the reference list. This should be amended.

Response: The citations in Table 3 are added in reference list (line 888-890, 536-539, 803-808, 581-583, 632-636, 437-441, 740-753, 504-508, 515-525, 836-839, 540-548, 918-922 in revised version).

Output data Comments: all reconstructed climate variables and the biomes for each site at 6ka should be archived (the values for the colored dots in Figs. 2-4).

Response: these values are provided in Table S4 (on pages 5-12 in revised version).

Comments: the simulated values for each of the climate variables (the colored bars in Figs. 2-4).

Response: For the simulated values, we think the grid mean values in the colored bars in Figs. 2-4, and the spatial patterns in Figs. S1-6 are enough to show the discrepancies between model-data for climate change during mid-Holocene over China. Of course, if the editor insists it's necessary, we can also provide the simulated values for each pollen site of the climate variables from all 13 models.

Comments: basic metadata for each site (the information listed in Table 1)

Response: Provided in Table 1.

2. The "data availability" section Comments: it should include the doi/url that links to the output data from this study and the reiterates the location of the input data (above). Alternatively, locations of the input data can be provided in the table.

Response: We prefer to provide our data directly in a Table (like Table S4) instead of putting them in a vetted repository.

3. Other reminders Comments: Of course, the authors might also want to make the digital data for their maps available, or anything that they think might be useful for future studies.

Response: For now, we think all these information about our input and output data are enough, of course, we are open-minded to other possible requests.

Comments: A special attention to the reference list is required. The references marked as red in the attached file are missing in the reference list. The author should also check lines 483, 637 and 694 of their reference list. Is CQPD 2000 or 2001 or both? And also please note the supplement to this comment: http://www.clim-past-discuss.net/cp-2018-145-Ec1-supplement.pdf.

Response: It's CQPD 2000, and all these missing and errors in the reference list are modified in the revised version.

The revised version of manuscript and supplementary information are enclose below as supplement.zip.

Please also note the supplement to this comment:
https://www.clim-past-discuss.net/cp-2018-145/cp-2018-145-AC1-supplement.zip

---

## Author Comment (AC2) · 5 Feb 2019

We greatly appreciate the constructive comments and suggestions on the previous version of the manuscript from Reviewer #1. We have attempted to address every point raised. The following is the outline of the changes we have made, with reference to the order of the comments made by the referee.

Comments from Reviewer #1: The model-data comparison of climate change during mid-Holocene (MH) is an important issue to validate the results from Global Circulation Model (GCM) against the proxies gathered from dataset. Based on the new pollen dataset and Inverse Vegetation Model (IVM). This study provided a quantitative reconstruction of climate variables during MH over China was provided and compared to the

simulation results from 13 models in PMIP3. A large discrepancy on the temperature anomaly between model-data at both annual and seasonal scale was depicted, mainly due to the failure of capturing vegetation change during MH by models, which is very helpful for better understanding the climatic changes during MH, and also pinpoints the possible way to reconcile model and data by accurately simulating the non-linear responses of vegetation and hydrology in GCMs. The manuscript can be accepted for publication after minor revision. A few basic comments and some issues to deal with as follow: 1. Since it's a quantitative model-data comparison based on pollen dataset, in which 91 records were digitized from published papers. More detailed information about the data should be provided, like the age control, pollen assemblages from around 6 ka at each site.

RE: the required information has been added in the Table 1 (page 42-45) and Table S4 (page 5-12 in Supplementary Information).

2. As mentioned in the manuscript, there is a difference in vegetation inputs for the MH period among models in PMIP3, a table for detail information should be given.

RE: we added a new Table S5 in the supplementary information (page 13 in Supplementary Information).

3. The disparity of temperature anomaly during MH among models could be resulted from the difference in pre-industrial (PI) simulation. Authors should prove that there is no any clear relationship between PI temperature and temperature change (MH-PI).

RE: Fig.R1, as attached below, demonstrates that there is no any clear relationship between PI temperature and temperature change (MH-PI), for both annual and seasonal scale, which means the disparity of temperature anomaly during MH among models doesn't come from the difference in PI simulation.

4. Some references are missing in the reference list. Such as the citations in Table 3.

RE: we have added the citations from Table 3 in the reference list (line 888-890, 536-

539, 803-808, 581-583, 632-636, 437-441, 740-753, 504-508, 515-525, 836-839, 540-548, 918-922 in revised version) .

The revised version of manuscript and supplementary information are enclosed below as supplement zip.

Please also note the supplement to this comment:
https://www.clim-past-discuss.net/cp-2018-145/cp-2018-145-AC2-supplement.zip
* * *
[Figure]

[Figure]

**Fig. 1.**

**Table S5. Vegetation setting for the mid-Holocene among models in PMIP3**

| Model | L A I | Stomatal Resistance Function Of | Vegetation Time Variation |
|---|---|---|---|
| CCSM4 | Prognostic | CO2 \| Light \| Temperature \| Water availability | Prescribed (varying from files) |
| MIROC-ESM | Prescribed | CO2 \| Light \| Temperature \| Water availability | Prescribed (varying from files) |
| BCC-CSM1.1 | Prognostic | CO2 \| Light \| Temperature \| Water availability | Prescribed (varying from files) |
| CNRM-CM5 | Prescribed | Light \| Temperature \| Water availability | Fixed (not varying) |
| CSIRO-MK3.6.0 | Prescribed | Light \| Temperature \| Water availability | Prescribed (varying from files) |
| GISS-E2-R | Prescribed | CO2 \| Light \| Temperature \| Water availability | Fixed (not varying) |
| IPSL-CM5A-LR | Prognostic | CO2 \| Light \| Temperature \| Water availability | Prescribed (varying from files) |
| MPI-ESM-P | Prognostic | CO2 \| Water availability | Fixed (not varying) |
| MRI-CGCM3 | Prescribed | CO2 \| Light \| Water availability | Prescribed (varying from files) |
| HadGEM2-ES | Prognostic | CO2 \| Light \| Temperature \| Water availability | Dynamical (varying from simulation) |
| HadGEM2-CC | Prognostic | CO2 \| Light \| Temperature \| Water availability | Dynamical (varying from simulation) |
| FGOALS-g2 | Prescribed | no data | Prescribed (varying from files) |
| FGOALS-s2 | Prescribed | no data | Prescribed (varying from files) |

**Fig. 2.**

---

## Author Comment (AC4) · 5 Feb 2019

We greatly appreciate the constructive comments and suggestions on the previous version of the manuscript from Reviewer #3. We have attempted to address every point raised. The following is the outline of the changes we have made, with reference to the order of the comments made by the referee.

Comments from Reviewer #3:

The manuscript entitled "Mid-Holocene climate change over China: model-data discrepancy" by Lin et al. presented a study on model-data comparison by using the pollen data collection in China and PMIP3 mid-Holocene simulations. From the large discrepancy showed in model-data comparison, both in annual mean, warmest month Printer-friendly version

and coldest month, they conclude that the major reason that PMIP3 simulations do not agree with data is because the vegetation distribution is not properly represented in climate models, where most models do not include dynamical vegetation and the prescribed MH vegetation map is the same as preindustrial. The MH vegetation issues have been recognized in recent years and many efforts are made to reconstruct a better MH land cover map, this includes the PAGES working group on Landcover6k. Therefore a good vegetation map from China would be expected to contribute to an eventual global land-cover map during the mid-Holocene and benefit the paleoclimate community. However, the current work has a somewhat mislead focus and I have the following major concerns.

General comments from Reviewer #3:

1. The reconstructed mid-Holocene climate in their study is largely depend on the pollen data collection. I am not an expert on pollen data, but I am wondering if all the published data use the same standard on data process. Can they be synthesized by Webb1-7 standard and put together for comparison? I hope a reviewer from pollen community may have some insights on the data process. There are no discussion on the potential uncertainties on collected data, at least one comparison with other proxy data can provide the cross-proxy verification. The authors emphasized three original data but no detailed information, which are important if they are not published. When the significant differences are found in model-data comparisons, the uncertainties from the data should be discussed as well. One can not regard reconstruction is the truth. We need to know how reliable is the reconstructed climate from pollen data, given that the IVF method used to reconstruct the climate is a crude estimate. Otherwise it is dangerous if this paper is published and people take for granted that this is the climate (and vegetation map) in China during mid-Holocene.

RE: Thanks for this very important comment, we will answer it point by point: Firstly, can the pollen records collected from papers be synthesized by Webb 1-7? Yes, we need to do some data processing before we use the collected pollen records to recon-

CPD
struct the climates. Firstly, the published papers only give the pollen diagram, not the pollen assemblages. So we need to digitize the pollen diagram for obtaining the pollen assemblages, and then use biomization to get the biome scores and biome types. Secondly, for age control, different dating methods are used in the collected pollen records, we use CalPal 2007 (Weninger et al., 2007) to correct 14C age into calendar age so that they can be contrasted each other. For lacustrine records, if the specific carbon pool age is mentioned in the literature, the calendar age is corrected after deducting the carbon pool. Otherwise, the influence of carbon pool is not considered. The age series of records were obtained by linear regression or linear interpolation of adjacent dating data. After these preprocessing, a unified chronological standard for all pollen records is built, and then the classification of age control followed the standard of Webb 1-7.

Secondly, what's the potential uncertainties on data reconstruction? How reliable are the pollen data used in this study? How about the cross-proxy verification? For the reliability of pollen data, we have compared them with the BIOME6000 (Fig. 2, page 50), the match between collected data and the BIOME6000 is more than 90% for both MH and PI. For the potential uncertainties on data reconstruction, IVM is relied on the BIOME4, a global vegetation model. Because of the particularity of vegetation types in the monsoon region of China, the BIOME4 needs further improvement of vegetation simulation accuracy in this area. This possible bias in simulating vegetation will lead to uncertainty in reconstruction. In this version, we added more information in Table S4 in supplementary information, we gave the climate variables reconstructed from IVM at each site. Moreover, we also showed the bias on data reconstruction by giving the median value (for instance, column named MTCO) and values indicating the 5% (MTCO1)-95% (MTCO2) uncertainty bands. For the cross-proxy verification, we compared our reconstruction with previous studies over China based on multiple proxies (including pollen, lake core, palaeosol, ice core, peat and stalagmite). Compared to PI, most reconstructions reproduced a warmer and wetter annual condition during MH (Fig. 1 as below), same as our study. In other word, this discrepancy between

**CPD**
model-data for climate change over China during MH is common and robust in reconstructions derived from different proxies. Our study just reinforces the picture given by the discrepancies between PMIP simulation and pollen data derived from a synthesis of the literature.

Thirdly, about the three original data, thanks for the reminder. They have been published, and we added the information in Table 1, as well as the reference list (line 956-958, page 41 and line 866-869, page 37).

2. The BIOME4 produced vegetation pattern in fig5 is determined by the input climate variables from the model, given the supplementary figures s1-s6 and previous studies by Jiang et al. (2012) have already show different climate patterns produced by different models, therefore the mismatch in vegetation pattern and reconstructed map in Fig5 is expected. I don't think this mismatch can be used to argue that the modelled MH climate is not good because they did not use a correct vegetation map and include the vegetation-climate interaction. Those vegetation patterns produced by BIMOE4 are not used in PMIP experiment setup, it would make more sense if authors compare the reconstruction and PMIP prescribed land cover map, or compare BIOME4 produced vegetation map with the ones produced by those climate models (for example HadGEM2-ES) that have dynamical vegetation to gain some understanding on vegetation-climate feedback.

RE: We totally agree with the reviewer, it's a very efficient way to test our proposal if we can run the simulation with the reconstructed vegetation in GCM. However, as far as we know, prescribing the vegetation in a coupled GCM is not easy. For instance, if we want to use the reconstructed vegetation in Orchidee (the vegetation module of IPSL), we need to modify numerous parameters to make sure that the experiment with new vegetation condition will not be killed by the model due to its mismatch in climate variables. Moreover, the GCM models in PMIP3 have their own vegetation module, it definitely takes much time to do such test, that's why in this paper we choose BIOME4 to evaluate MH vegetation simulation against the reconstructed result. Actually, we

CPD
already plan to conduct this experiment in Orchidee to further test the proposal of this paper, it's an ongoing work. Although we can't prescribe the MH vegetation with our reconstructed results in all PMIP3 models, we have succeeded to conduct such test in CESM version 1.0.5. The CESM version 1.0.5, developed at the National Center for Atmospheric Research, is a widely used coupled model with dynamic atmosphere (CAM4), land (CLM4), ocean (POP2), and sea-ice (CICE4) components (Gent et al., 2011). Here, we use  $\sim 2^{\circ}$  resolution for the CAM4, configured by  $\sim 1.9^{\circ}$  (latitude)  $\times 2.5^{\circ}$ (longitude) in the horizontal direction and 26 layers in the vertical direction. The POP2 adopts a finer grid, with a nominal 1° horizontal resolution and 60 layers in the vertical direction. The land and sea-ice components have the same horizontal grids as the atmosphere and ocean components, respectively. Two experiments were conducted, including a mid-Holocene (MH) experiment (6 ka) with original vegetation setting (prescribed as PI vegetation for MH) and a MH experiment with reconstructed vegetation (6 ka VEG). In detail, experiment 6 ka used the MH orbital parameters (Eccentricity=0.018682; Obliguity=24.105°; Angular precession=0.87°) and modern vegetation (Salzmann et al., 2008). Compared to experiment 6 ka, experiment 6 ka VEG used our reconstructed vegetation in China. Except for the changed vegetation, all other boundary conditions were kept unchanged in these two experiments, including the solar constant (1365 W m-2), modern topography and ice sheet, and pre-industrial greenhouse gases (CO2 = 280 ppmv; CH4 = 760 ppbv; N2O = 270 ppbv). Experiment 6 ka was initiated from the default pre-industrial simulation and run for 500 model years. Experiment 6 ka VEG was initiated from model year 301 of experiment 6 ka and run for another 200 model years. We analyzed the computed climatological means of the last 50 model years from each experiment here. The new-added Fig.8 in manuscript (enclosed here as Fig. 2) shows the climate anomalies between two simulations (6 ka VEG minus 6 ka), for both annual and seasonal scale. For temperature, it's clear that the 6 ka VEG simulation reproduces the warmer annual ( $\sim$ 0.3 K on average) and winter temperature ( $\sim 0.6$  K on average), especially the winter temperature. For precipitation, the reconstructed vegetation leads to higher annual and seasonal precipitation,

**CPD**
which can also reconcile the discrepancy of increase amplitude for precipitation during MH between model-data (data reproduced larger amplitude than model, revealed by our study). So it's true that the mismatch between model-data in MH vegetation has significant influence on the discrepancy of climate, this is consistent with our proposal in this study. Each model has different sensitivity to the boundary change, further work should be carried out in more models to test the influence of vegetation on climate, this is an ongoing work.

Specific comments:

1. The abstract need to provide more information from this work, now only contains motivation and conclusion. And the conclusion in abstract actually is a speculation, did not come from the results of this work.

RE: We modified the abstract as following: The mid-Holocene period (MH) has long been an ideal target for the validation of Global Circulation Model (GCM) results against reconstructions gathered in global datasets. These studies aimed to test the GCM sensitivity mainly to the seasonal changes induced by the orbital parameters (precession). Despite widespread agreement between model results and data on the MH climate, some important differences still exist. There is no consensus on the continental size of the MH thermal climate response, which makes regional quantitative reconstruction critical to obtain a comprehensive understanding of the MH climate patterns. Here, we compare the annual and seasonal outputs from the most recent Paleoclimate Modelling Intercomparison Projects Phase 3 (PMIP3) models with an updated synthesis of climate reconstruction over China, including, for the first time, a seasonal cycle of temperature and precipitation. Our results indicate that the main discrepancies between model-data for MH climates are the annual and winter mean temperature. A warmer-than-present climate condition are derived from pollen data for both annual mean temperature ( $\sim$ 0.7 K on average) and winter mean temperature ( $\sim$ 1 K on average), while most of the models provide a linear response driven by the seasonal forcing (a decreased annual mean temperature with a warmer summer and colder winter). By

CPD
conducting simulations in BIOME4 and CESM, we show that to capture the seasonal pattern reconstructed by data, it is critical to assess surface processes. These results pinpoint the crucial importance of including the non-linear of the surface water and energy balance to vegetation changes.

2. Take line 49 as an example, 0.5K should be write as 0.5 K, follow SI standard, there is a space between number and unit. May correct throughout the manuscript.

RE: Corrected. Line 25, 26, 55, 73, 77, 78, 79, 237, 238, 240, 245, 248, 250, 255, 257, 259, 260, 388.

3. Line 116, "The new sites", if it is new, the data information should be described, otherwise they are unknown.

RE: Corrected by adding the description of new sites "91 digitized data and three original data" on page 6 line 123 in revised version.

4. Line 120, what is cloudiness, how are they measured? Because this is not a common variable, should be described.

RE: "Cloudiness" means the "Total Cloud Fraction", it is calculated for the whole atmospheric column, as seen from the surface or the top of atmosphere. Include both large-scale and convective cloud. The standard output name in PMIP is "clt". It's an inverse measure of sunshine (corrected in page 6 line 128 in revised version).

5. Line 129, how do you determine the anomalies for biome scores? What is the purpose of this paragraph L120-L139, to produce reconstruction in Fig5?

RE: To determine the anomalies for biome scores, we first use the biomization (Prentice et al., 1996) to get the biome score calculated from pollen taxa percentage for both MH and PI. And then we get the biome score anomaly (MH-PI). The purpose of line 120-139 is to demonstrate the scheme of artificial neural network (ANN) used in our study, and by using this interpolation method, we get the reconstructed spatial pattern of vegetation in Fig. 5 with red rectangle. The schematic diagram of ANN is provided as

CPD
below (Fig. 3).

6. Line 143 to Line 147, on description of PMIP is a bit strange, what do you mean"in which the PI experiment was denied". "The main variability between MH and PI" should be "The main forcing between MH and PI".

RE:From Line 143 to Line 145, we previously wrote as "In its third phase (PMIP3), the models were identical to those used in the CMIP5 experiments, in which the PI experiment was defined", here we mean that the protocol of PI is defined in CMIP5, and the models of PMIP3 followed that from CMIP5. Now according to the suggestion from you and Patrick Bartlein, we deleted the words "in which the PI experiment was defined". We corrected the "variability" into "forcing" (on page 7 line 156 in revised version).

7. Line 156, "interpolated to a common 2.5 grid", why do you think 2.5 is a common grid, given the pollen data are very local, 2.5 degree grid is too coarse.

RE: The "common" here means "uniform", we have corrected (on page 7 line 166 in revised version) . For the model resolution, yes, our study focus on China area, not globe, but for simulation, 2.5 degree is not very coarse even for local study. Moreover, some global models used in our study have lower than 2.5°\*2.5° resolution, like IPSL-CM5A-LR (96\*96), it will not make more sense even if we interpolate it into higher resolution.

8. Line 161-162, How do you obtain the sunshine data from observation and model? Should be described more specific.

RE: The sunshine data could be calculated as an inverse measure of cloudiness. Cloudiness means the "total cloud fraction", it's an output of model which named as "clt". We first obtained the "clt" from each model, then calculate the sunshine based on "clt". In the new version of manuscript (on page 8 line 172 in revised version), we described more specific "sunshine percentage (an inverse measure of cloud area
fraction)".

9. Line 184, "Weighting the attributes is subjective", will it cause uncertainties?

RE: As we mentioned in the manuscript, weighting the attributes is subjective because there is no obvious theoretical basis for relative significance. The attributes values listed in Table 4 and Table 5 are according to the previous studies (Skyes et al., 1999; Ni et al., 2000). It may cause some bias, however, it is not likely that different ecologists would assign greatly differing values (Skyes et al., 1999).

10. Line 191, from Zhang et al., 2010, the reference can not be found in reference list.

RE: Added in the reference list (on page 40 lines 933-935 in revised version).

11. I am wondering if the warmest month and coldest month changes between MH and PI (and between the models), or always July and January? Give there is a change in seasonality in MH, authors should mention this.

RE: For reconstruction, we obtained the biological climate based on pollen records, and the warmest month and coldest month are not always July and January for both MH and PI. and for models, we calculated the mean temperature for every month, and selected the warmest and coldest one to compare with the reconstruction. So the change in seasonality during MH doesn't influence our comparison for MTWA and MTCO.

12. Line 261, "with a decrease in the northeastern regions", also decrease in east monsoon region at Yangzi river valley.

RE: We have added the words "with a decrease in the northeastern regions and east monsoon region at Yangtze River valley" (on page 12 line 271 in revised version).

13. Line 310-312, "this failure to capture ..", see above general comment 2.

RE: See the answers to comment 2.

CPD
14. Line 320, "triggered" is a weird word.

RE: We corrected it into "caused" (on page 14 line 331 in revised version ).

15. Fig 7 on feedback discussion, how do you determine the feedbacks from the cloud cover or surface cover? In Line 356 the authors mentioned the "surface albedo and cloud change are calculated . . .", I don't understand why the changes in forcing can be regarded as feedback, physically it is a climate response to forcing.

RE: The albedo feedback is not identical to the changes in forcing, and it doesn't directly response to forcing. In fact, our philosophy to calculate the albedo feedback is: the forcing change during MH (mainly the seasonal solar radiation change) firstly leads to the seasonal climate change, and accordingly, the vegetation type at that period will be different from PI. Then, the changes in vegetation type have a feedback on climate through the albedo variation. This feedback can be calculated by measuring the changes in radiative fluxes at both Earth's surface and at the top of atmosphere. For instance, if the land surface changes from bare soil to forest, the surface albedo will decrease and the net radiation at land surface increase, in this case, the surface albedo has a positive feedback on the net shortwave flux. The detailed information about how to quantify the feedback are shown in Taylor et al., (2007) and Braconnot and Kageyama (2015).

16. Line 733, "Importance" should be "Important".

RE: Corrected.

17. Table 6, should give more information for meteorological data, how long, and give which month is the warmest coldest month, in line 742, should be "warmest month".

RE: Corrected.

18. Line 744, "stand error" means "standard deviation"?

RE: Corrected.
19. Figure 1, should you mark your three original data in this map separately?

RE: The three original data are marked in the Fig. 1 in revised version of manuscript (on page 49) as green circles.

20. Figure 4, the huge annual precipitation anomaly in reconstruction, how reliable is it? I highly suspect it. The unit for precipitation is mm, does it mean annual 240 mm equal 20 mm/month? I suggest you use mm/month to avoid confusion.

RE: Yes, from our reconstruction, the annual precipitation anomaly (MH-PI) is huge. This increase in mean annual precipitation (MAP) is mainly due to the increased intensity of monsoon in eastern area over China, which brings much higher precipitation during summer, and results in an increased MAP. In Table 6, the regression coefficient (R) between the reconstructed modern MAP by inverse vegetation models (IVM) and observed meteorological values is 0.94, which means the MAP reproduced from the IVM is reliable during present day. But it's also true that there are some bias in MAP reconstruction, in Table S4, we give the median value (MAP) and values indicating the 5% (MAP1)-95% (MAP2) uncertainty bands to show the bias in IVM reconstruction.

21. Figure 6, Line 848 to 850, why do you give the abbreviation, they are not in the figure.

RE: Corrected (page 54 in revised version).

The revised version of manuscript and supplementary information are enclosed as below as supplement.zip.

Reference: Weninger, B., Jöris, O., Danzeglocke, U., 2007. CalPal-2007. Cologne Radiocarbon Calibration and Palaeoclimate Research Package. http://www.calpal.de/. Jiang, D., Lang, X., Tian, Z., and Wang, T.: Considerable Model–Data Mismatch in Temperature over China during the Mid-Holocene: Results of PMIP Simulations, Journal of Climate, 25, 4135-4153, 2012. Gent, P.R., Danabasoglu, G., Donner, L.J., Holland, M.M., Hunke, E.C., Jayne, S.R., Lawrence, D.M., Neale, R.B., Rasch, P.J.,

CPD
Vertenstein, M., Worley, P.H., Yang, Z., and Zhang, M.: The community climate system model version 4, Journal of Climate, 24, 4973-4991, 2011. Salzmann, U., Haywood, A.M., Lunt, D.J., Valdes, P.J., and Hill, D.J.: A new global biome reconstruction and data-model comparison for the middle Pliocene, Global Ecology and Biogeography, 17, 432-447, 2008. Prentice, I. C., Guiot, J., Huntley, B., Jolly, D., and Cheddadi, R.: Reconstructing biomes from palaeoecological data: A general method and its application to European pollen data at 0 and 6 ka, Climate Dynamics, 12, 185-194, 1996. Sykes, M.T., Prentice, I.C., and Laarif, F.: Quantifying the impact of global climate change on potential natural vegetation, Climatic Change, 41, 37-52, 1999. Ni, J., Sykes, M. T., Prentice, I. C., and Cramer, W.: Modelling the vegetation of China using the process-based equilibrium terrestrial biosphere model BIOME3, Global Ecology and Biogeography, 9, 463-479, 2000. Taylor, K.E., Crucifix, M., Braconnot, P., Hewitt, C. D., Doutriaux. C., Broccoli, A. J., Mitchell, J. F. B., Webb, M. J.: Estimating shortwave radiative forcing and response in climate models, J. Clim, 20, 2530-2543, 2007. Braconnot, P., and Kagevama, M.: Shortwave forcing and feedbacks in Last Glacial Maximum and Mid-Holocene PMIP3 simulations, Philosophical Transactions of the Royal Society A: Mathematical, Physical and Engineering Sciences, 373, 2054-2060, 2015.

Please also note the supplement to this comment: https://www.clim-past-discuss.net/cp-2018-145/cp-2018-145-AC4-supplement.zip

---

## Author Comment (AC5) · 5 Feb 2019

We greatly appreciate the constructive comments and suggestions on the previous version of the manuscript from Prof. Bartlein. We have attempted to address every point raised. The following is the outline of the changes we have made, with reference to the order of the comments made by the referee.

General comments from the referee:

This paper presents an ambitious attempt at comparing simulations from the CMIP5/PMIP3 "midHolocene" archive with a new synthesis of fossil-pollen data for China. The pollen data are used in two ways: 1) to develop a set of quantitative reconstructions of several climate variables using an inverse-modeling approach (to

compare with the climate-model output), and 2) to develop a map of "megabiomes" for present and 6 ka (for direct comparison with vegetation simulated by BIOME4 using climate-model output). The paper shows that there is a considerable mismatch between the reconstructed and simulated climates and vegetation. The authors attribute this mismatch to experimental-design issues, in which vegetation and land-cover data in the climate models were fixed at present-day values, thereby limiting the ability of the climate models to correctly represent the potential impact of vegetation (and surface water- and energy-balance) feedback in the paleo simulations.

I have two reservations about the results and conclusions: First, there is insufficient information on the protocols adopted for generating the both present-day and paleo vegetation, as well as the paleo reconstructions. As for vegetation, Fig. 5 shows that there are large mismatches between the observed and simulated modern vegetation. These would naturally arise if the climate-model output were used directly to simulate the vegetation. We know that at their current resolutions, there is still considerable bias in present-day (or PI) climate simulations, and there is no reason to believe that those bias are the same in paleo simulations (or that they somehow go away). My impression of Fig. 5 and S7 is that those biases in simulated climate are indeed large, possibly swamping the real vegetation change, and so we're not really getting much insight into the nature of the mid-Holocene climate simulations, but instead learning about modern-day bias. As for reconstructed climate, despite the author's assertion otherwise, there is also considerable bias in the inverse-model reconstruction approach (Table 6). Only for Pjan does the regression between observed and fitted values not differ from one with a slope of 1.0 and an intercept of 0.0. It is not immediately clear how that bias might affect the reconstructed climate, but it reinforces the necessity of looking at the uncertainties in the reconstructions. Again, we may be learning more about the inverse approach than about model-data mismatches. Second, the attribution of the mismatches to the experimental design of the CMIP5/PMIP3 simulations, while plausible, is not really supported by any direct hypothesis tests, or by the consideration and dismissal of alternative hypotheses. The correlation between the temperature responses

and cloud-cover feedback (Figure 7) implicates at least one: inadequate simulation of atmospheric circulation as it may influence moisture flux or precipitation-generating mechanisms. I think that if the questions related to the protocol for data-model comparisons are answered, and due consideration given to other possible mechanisms for the mismatch, then the paper will ultimately be publishable.

Response to the major comments: Thanks for the very important comments, in conclusion, two main questions are proposed here:

1. The insufficient information on protocol for model-data comparison of vegetation.

RE: The referee is right when he pointed out that using climate for PD or 6 ka, there is a large mismatch between vegetation reconstructed by BIOME4 from model simulation and vegetation reconstructed from pollen data. Our aim here is to test whether the sensitivity of PMIP3 mid-Holocene simulations, mainly driven by insolation changes, may explain the vegetation changes observed from data. The trend depicted by all the models is cooler conditions during winter and warmer in summer which is consistent with a linear response to orbital forcing for 6 ka. On the contrary, the dataset shows for the seasonal response a warming during both seasons. We agree with Prof. Bartlein that the attribution of those non linear responses to vegetation is not really explained in our original manuscript. This is due to the fact that each model has different ways to account for vegetation, therefore our explanation was plausible but too speculative. To be able to convince the referee with quantitative arguments, we have conducted a supplementary experiment to demonstrate that our mechanism was appropriate. In this revised version, we succeeded to conduct such test in CESM version 1.0.5. The CESM version 1.0.5, developed at the National Center for Atmospheric Research, is a widely used coupled model with dynamic atmosphere (CAM4), land (CLM4), ocean (POP2), and sea-ice (CICE4) components (Gent et al., 2011). Here, we use $\sim2°$ resolution for the CAM4, configured by $\sim1.9°$ (latitude) $\times$ 2.5° (longitude) in the horizontal direction and 26 layers in the vertical direction. The POP2 adopts a finer grid, with a nominal 1° horizontal resolution and 60 layers in the vertical direction. The land and sea-ice components have the same horizontal grids as the atmosphere and ocean components, respectively. Two experiments were conducted, including a mid-Holocene (MH) experiment (6 ka) with original vegetation setting (prescribed as PI vegetation for MH) and a MH experiment with reconstructed vegetation (6 ka_VEG). In detail, experiment 6 ka used the MH orbital parameters (Eccentricity=0.018682; Obliquity=24.105°; Angular precession=0.87°) and modern vegetation (Salzmann et al., 2008). Compared to experiment 6 ka, experiment 6 ka_VEG used our reconstructed vegetation in China. Except for the changed vegetation, all other boundary conditions were kept unchanged in these two experiments, including the solar constant (1365 W m$-2$), modern topography and ice sheet, and pre-industrial greenhouse gases ($CO_2$ = 280 ppmv; $CH_4$ = 760 ppbv; $N_2O$ = 270 ppbv). Experiment 6 ka was initiated from the default pre-industrial simulation and run for 500 model years. Experiment 6 ka_VEG was initiated from model year 301 of experiment 6 ka and run for another 200 model years. We analyzed the computed climatological means of the last 50 model years from each experiment here. The new-added Fig.8 in manuscript (enclosed below as Fig. 1) shows the climate anomalies between two simulations (6 ka_VEG minus 6 ka), for both annual and seasonal scale. For temperature, it's clear that the 6 ka_VEG simulation reproduces the warmer annual (∼0.3 K on average) and winter temperature (∼0.6 K on average), especially the winter temperature. For precipitation, the reconstructed vegetation leads to higher annual and seasonal precipitation, which can also reconcile the discrepancy of increase amplitude for precipitation during MH between model-data (data reproduced larger amplitude than model, revealed by our study). This new result strongly suggests vegetation changes may explain a part of the mismatch, which is consistent with our proposal in this study. Nevertheless, there are certainly other possibilities and indeed models that better captured the hydrologic cycle and enhance the precipitation/ evaporation pattern could also explain differences between model and data. Each model has different sensitivity to the boundary change, further work will be carried out in more models to test the influence of vegetation on climate, this is an ongoing work. This response is on pages 16-17, lines 382-400 in the revised version.

2. More information about the considerable bias in the inverse-model reconstruction approach.

RE: In the climate reconstruction based on pollen data, the inverse-modeling approach considers the impact of $CO_2$, and provides sound logic to cope with the no-analogue problem. Moreover, the consequence of changing seasonality of climate forcing or climate response are also taken into account in the inverse-model reconstruction. The quantitative reconstructions derived from pollen data of Eurasia, Africa and Europe (Wu et al., 2007) at the LGM and the mid-Holocene, confirm the ability of the inverse vegetation model (IVM) method to provide spatially coherent patterns of palaeoclimate that are generally in agreement with previous reconstructions from climate proxies. However, the IVM approach is not a panacea. First, it is highly dependent on the quality of the vegetation model BIOME4, because of the particularity of vegetation types in the monsoon region of China, the BIOME4 needs further improvement of vegetation simulation accuracy in this area. This possible bias in simulating vegetation will lead to uncertainty in reconstruction. Second, the output of the model is not directly compared to the pollen data, the conversion of BIOME4 biomes to pollen biomes by the transfer matrix may add the source of uncertainty in reconstruction. These possible bias in climate reconstruction derived from IVM are described in the new version of manuscript (on page17 lines 401-411 in revised version). For the potential uncertainties on data reconstruction, besides the Table 6, we added more information in Table S4 in supplementary information, we gave the climate variables reconstructed from IVM at each site. Moreover, we also showed the bias on data reconstruction by giving the median value (for instance, column named MTCO) and values indicating the 5% (MTCO1)-95% (MTCO2) uncertainty bands.

Specific comments from the referee: Abstract: The abstract fails to disclose conclusions of paper.

RE: We modified the abstract as following: The mid-Holocene period (MH) has long been an ideal target for the validation of Global Circulation Model (GCM) results against

reconstructions gathered in global datasets. These studies aimed to test the GCM sensitivity mainly to the seasonal changes induced by the orbital parameters (precession). Despite widespread agreement between model results and data on the MH climate, some important differences still exist. There is no consensus on the continental size of the MH thermal climate response, which makes regional quantitative reconstruction critical to obtain a comprehensive understanding of the MH climate patterns. Here, we compare the annual and seasonal outputs from the most recent Paleoclimate Modelling Intercomparison Projects Phase 3 (PMIP3) models with an updated synthesis of climate reconstruction over China, including, for the first time, a seasonal cycle of temperature and precipitation. Our results indicate that the main discrepancies between model-data for MH climates are the annual and winter mean temperature. A warmer-than-present climate condition are derived from pollen data for both annual mean temperature ($\sim$0.7 K on average) and winter mean temperature ($\sim$1 K on average), while most of the models provide a linear response driven by the seasonal forcing (a decreased annual mean temperature with a warmer summer and colder winter). By conducting simulations in BIOME4 and CESM version 1.0.5, we show that to capture the seasonal pattern reconstructed by data, it is critical to assess surface processes. These results pinpoint the crucial importance of including the non-linear of the surface water and energy balance to vegetation changes.

Line 14: "proxy reconstructions" Aren't the reconstructions used here "real" reconstructions? I understand the notion of paleoclimatic evidence that can be used as a "proxy" for climate or other phenomena (like land cover). But the reconstructions here are actual reconstructions, not a stand-in or substitute for reconstructions.

RE: We have deleted the word "proxy" in Line 14 in revised version.

Line 18: "continental size" Are you referring to the area of the temperature anomaly or to terrestrial as opposed to marine responses?

RE: We are referring to the area of the temperature anomaly.

Line 20: New definition for PMIP?

RE: Corrected (on page 1 line 20 in revised version).

Line 22: "a seasonal cycle. . ."

RE: Corrected as "a seasonal cycle of temperature and precipitation" (on page 1 line 22 in revised version).

Line 25: "access surface processes" I don't know what this means.

RE: Sorry for the wrong spelling, it should be "assess" (on page 1 line 25 in revised version).

Line 27: "non-linear process associated with vegetation changes in hydrology and radiative forcing" Does this mean "non-linear responses in hydrology and radiative forcing to vegetation changes"? "Radiative forcing" in the context of the midHolocene experiment is usually reserved for describing the insolation forcing, so an alternative expression might be "non-linear response of the surface water and energy balance to vegetation changes" (which is what I think the paper is arguing for).

RE: Thanks for the suggested expression, we have corrected it (on page 2 line 30 in revised version).

Line 34: This definition of the age of the mid-Holocene is inconsistent with what is actually used in the paper (line 101). It might be good to distinguish between the midHolocene time slice, and the "midHolocene" CMIP5/PMIP3 experiment, throughout the paper.

RE: According to IntCal13 (Reimer et al., 2013), the mid-Holocene time slice 6000±500 14C yr BP is about 6800 Cal BP (the average value), which is not totally consistent with the "mid-Holocene" used in CMIP5/PMIP3 experiment (6000 Cal BP). We agree with the reviewer that this is a problem in model-data comparison for paleoclimate, but for a better comparison with BIOME6000 (which defined as 6000±500 14C yr BP), we

decided to choose the pollen data at 6000±500 14C yr BP in our study. Thanks to the comment, we will take care of this inconsistency and make better comparison of time slice in the future work.

Line 36: "an increase in insolation in the seasonal cycle" Replace with "an increase in the amplitude of the seasonal cycle of insolation. . ."

RE: Corrected. On page 2 line 40 in revised version.

Line 38: "climate response to changes in the seasonal distribution" It's not the response to the seasonal variations of insolation that you're looking at here, but instead the response to changes in the distribution.

RE: Corrected. On page 2 line 43 in revised version.

Line 42: "consistency of the dataset incorporating different proxies" I don't know what that means. ãĂĂ RE: We have changed it into "much work has been done to reconstruct the paleoclimate change based on different proxies" (on page 2 line 46 in revised version).

Line 45: Again, the data are real, not proxy.

RE: Corrected (on page 3 line 43 in revised version).

Line 47: "the source of discrepancies. . ."

RE: Corrected it as "the source of discrepancies between model and data . . ." (on page 3 line 52 in revised version).

Lines 50-51: But see Marsicek et al. (2018, Nature) the "Holocene conundrum" apparently arose from comparing apples and oranges. A different example might be more convincing.

RE: Yes, the reconstruction used in Marcott et al. (2013) is mainly from the marine records (∼80%) and the cooling trend is largely associated with North Atlantic. And

Interactive
comment

in Marsicek et al. (2018), they show a better consistency of temperature between model-data for Europe and North America continents during Holocene based on 642 sub-fossil pollen data. The different trends of pollen- and marine-based reconstruction indicate the spatial variability of annual temperature change during MH over the globe, which has already been investigated by Bartlein et al. (2010). Here, we use Liu et al. (2014) to pinpoint the decreased annual temperature in MH simulated by model, compared to PI.

Line 62: The sheer expanse of the country. . . Why should the synthesis of paleoclimatic data or simulations necessarily be restricted to political subdivisions? Extending the area of the comparison deeper into the interior of Eurasia would generate a bit more "leverage" in comparing the data and models, but I understand the logic of restricting the analysis to China.

RE: For this study, we only focus on China, but we agree that extending the area into Eurasia or globe is more comprehensive, which is carrying out by other colleges in our group now.

Line 64 (and elsewhere). The article "the" is required before "MH" in this context (i.e. when "MH" is being used as a noun). Elsewhere, as in line 55, where "MH" is used as a modifier of another word ("precipitation" in this context), the article is not used.

RE: Corrected.

Line 66: "warmer and wetter than present. . ."

RE: Corrected it as "warmer and wetter than present conditions" (on page 3 line 71 in revised version)..

Line 73: "colder than the baseline" What baseline? Present-day or preindustrial? RE: The baseline period of 36 models are different. 10 of 36 models refer to present-day, while others refer to preindustrial.

Line 75: "This study" Which study? Reword as "That study. . ." or more explicitly "Jiang

et al. (2013) were the first to point out the model-data discrepancy over China during the MH, but the lack of seasonal reconstructions in their study limits comparisons with simulations."?

RE: Corrected (on page 4 line 81 in revised version).

Line 83: Bartlein et al. didn't synthesize land-cover changes.

RE: Corrected.

Lines 86-91: The terminology here needs to be sorted out. The "process-based bio-geographic model" alluded to here is BIOME4, and it is employed in making inferences about past climates using an "inverse modeling through iterative forward modeling" (IMIFM) approach (Guiot et al. 2000; Wu et al., 2007, 2009). (See Izumi and Bartlein, 2016, GRL for further discussion.) So BIOME4 is the vegetation model, while the over-all approach (which employs that model) is "IMIFM" (or after that is all explained, simply "the inverse approach").

RE: Corrected (on page 4 lines 92-94 in revised version).

Line 91: "In the case of models. . ." Which models? Is it the case that you're evaluating the PMIP3 simulations made with state-of-the-art climate models using reconstructions of temperature and precipitation?

RE: Yes, we mean the PMIP3 models.

Line 94: "thanks to the seasonal reconstruction" But in all previous applications of the inverse modeling approach using BIOME4 or related models, some sort of recon-struction or estimation of the seasonal variations in climate must have been involved, because BIOME4 requires monthly temperature, precipitation and cloudiness (or sun-shine) data as input.

RE: Yes, the monthly climate variables are required in BIOME4 or related models, here we only emphasis that our study is to reconstruct the seasonal cycle of MH climate

change over China with a synthesis of pollen datasets.

Lines 95-96: "the forcing factor we used for MH is essential the seasonal change." I think that what's going on here is that the midHolocene CMIP5/PMIP3 experiment is essentially one that looks at the response of the models to changes in the seasonality of insolation, and that you are attempting to derive reconstructions of both summer and winter temperature and precipitation to compare with the simulations.

RE: Corrected, according to your suggestion (on page 5 line 100 in revised version).

Line 101: If you're referring to radiocarbon ages, this should be written as 6000 $\pm$ 500 14C yr BP)

RE: Corrected (on page 5 line 107 in revised version).

Line 102: Spell out "three".

RE: Corrected (on page 5 line 108 in revised version).

Line 105: I don't understand the notion of "distinct" pollen records. Distinct in the sense of "unique" or distinct in the sense of "clearly readable"?

RE: We have corrected it into "clearly readable" (on page 5 line 111 in revised version).

Line 107: Criterion 2 seems to be combining two things: sampling resolution and data present within the age range. Please reword.

RE: Corrected (on page 5 lines 112-114 in revised version).

Line 108: How far?

RE: We abandon the pollen records in our dataset if the published paper mentions the influence of human activity on the pollen. (replaced the "far away" by "abandon the pollen records if the published paper mentions the influence of human activity" on page 5 line 114 in revised version.

Line 109: "or by regression"

RE: Corrected (on page 5 line 116 in revised version).

Line 113: Fix the Webb (1985) citation. (Webb, T. III, etc.)

RE: Corrected (on page 5 line 119 in revised version).

Lines 110-113: Reorganize sentence to describe the ranking scheme first, and the results second.

RE: Corrected (on page 5 lines 116-120 in revised version).

Line 114: Add a citation for the concept of "biomization" (which will be a mystery to modelers).

RE: Corrected, we have added Prentice et al., 1996 (on page 5 line 120 in revised version).

Line 116: CQPD. Not in references. Also "sedimentary" what?

RE: The CQPD reference was added, and it should be "sediment" (on page 6 line 123 in revised version).

Lines 117-119: Add region names to Fig. 1?

RE: Corrected, we added the "Tibetan Plateau" and "Loess Plateau".

Lines 120-135: There seem to be two tasks described in the paragraph: 1) interpolation of modern climate data (from some unspecified source, and by some by some unspecified approach) to the locations of the pollen data, and 2) interpolation of biome scores onto a regular grid using ANN. I suggest breaking this paragraph up, while providing more information on the first task.

RE: The modern climate data are based on the datasets (1951-2001) from 657 meteorological observation stations over China, we also added the data source in the manuscript.

Line 129: If the ANN is calibrated using present-day biomes, then I don't see how it can

be used to interpolate anomalies. Or was it the case that present-day and paleo biomes were independently interpolated onto the grid, after which anomalies were calculated.

RE: The high spatial coverage of present-day pollen records and the application of ANN in interpolating the biome scores makes it possible to reconstruct the past spatial variability with a few pollen sites. In our study, at each pollen site, we firstly used the biomization to get the biome scores for both present-day (PD) and mid-Holocene (MH). Then we calculated the biome score anomalies between two periods (MH-PD). Based on the artificial neural network (ANN), we got the interpolated spatial pattern of biome scores for both PD and anomalies (MH-PD). The spatial pattern of MH biome scores were obtained by overlay the PD pattern with anomalies pattern (MH-PD). Finally, the biome with the highest index is attributed to each grid point, and thus, the spatial pattern of MH vegetation was obtained. The detailed scheme is provided in the enclosed Fig. 2 as below.

Lines 142-143: What are "climate anomalies in the present day"?

RE: Corrected it into "climate anomalies in the past periods" (on page 7 line 152-153 in revised version).

Line 145: Delete "in which the PI experiment was defined."

RE: Corrected.

Line 146: Here it would be good to refer explicitly to the "midHolocene" experiment.

RE: Corrected.

Line 153: Spell out eight and five.

RE: Corrected (on page 7 line 163 in revised version).

Line 156: "in order to calculate" These variables could also be calculated on the models' native grids. The motivation for interpolation onto a common grid is simply to get the data onto a common grid.

RE: Corrected (on pager 7 lines 167-168 in revised version).

Line 160: Either delete the hyphens here, or put them into other instances of "biogeography" or biogeochemistry".

RE: Corrected (on page 8 line 170 in revised version).

Line 162: "sunshine percentage (relative to cloud cover)" I don't know what this means.

RE: The sunshine percentage is related to cloud cover (not "relative"). We have corrected it as "an inverse measure od cloud area fraction (on page 8 line 172 in revised version).

Line 171: Bigelow: not in references.

RE: Corrected (on pages 19-20 lines 453-459 in revised version).

Line 173: Were the climate variables downscaled in any way (as in the apply-the anomalies approach, Harrison et al., 1998, J. Climate, Harrison et al., 2014, Climate Dynamics). If not, then the climate fields will not contain the spatial variability of modern climate that in topographically complex areas can have a major impact on vegetation. Fig. S7 attests to the existence of bias in the PI simulations. If the simulated climate values are used directly, than a quantitative estimate of the bias (as in Table 6 for the present-day reconstructions) should be provided.

RE: We directly used the climate fields from models without downscaling. But according to the Taylor diagrams (Figure 1, enclosed below as Fig. 3) from Jiang et al. (2016, Int. J. Climatol), the GCMs from PMIP is reliable to simulate the mean state and year-to-year variability of surface air temperature and precipitation over China for present day even without downscaling. Of course, we agree with the reviewer that there are bias and in the future work, we will try to downscale the climate variables before applying them into regional study.

Line 174: "more than 30 years" How much more? Why not use the same number of

years for each model?

RE: Corrected (on page 8 line 184 in revised version).

Line 176: Replace "model-data discrepancies" with "differences between simulated (by the climate-model output) and reconstructed (from pollen). . ."

RE: Corrected (on page 8 lines 186-187 in revised version).

Line 183: Replace "estimate" with "describe".

RE: CorrectedïijĹon page 9 line 194 in revised versionïijĽ.

Line 192-194: I'm not sure why you're describing the interpolation of biome data again. ãĂĂ RE: We have deleted this sentence.

Line 197: "Inverse Vegetation Model" See earlier comments.

RE: Corrected (on page 9 line 205 in revised version).

Line 208: I'm not sure why $CO_2$ concentrations and soil characteristics are being perturbed (i.e. estimated by the inverse approach). We know $CO_2$, and earlier you argued that soils were assumed not to differ. Also, Table 3 implies that anomalies (or better put, long-term mean differences between present and past) were iteratively generated, which implies that, as is standard procedure, they were applied to present-day climate values and passed to the biome model. If so, what were those present-day values?

RE: Sorry for the inaccurate expression here, the $CO_2$ concentration and soil are being perturbed by model. For the soil properties, because of a lack of paleosol data, soil characteristics were assumed to have been the same during the MH. While the atmospheric $CO_2$ concentration for the MH was taken from ice core records (EPICA community members 2004), and set at 270 ppmv; the modern $CO_2$ concentration was set at 340 ppmv, because most of the modern pollen samples were collected during 1970s and 1980s.

Lines 216-218: I don't know if I'm reading Table 6 correctly, but if I am, the slopes and intercepts are anything but close to 1.0 and 0.0. Only for the case of Pjan is the slope within two standard errors of 1.0, and only for MAP and Pjan is the intercept within two standard errors of 0.0. It would be useful to see scatter diagrams of the observed and estimated values for each variable.

RE: According to your suggestion, we added more information in Table S4, in which you can find the median value and values indicating the 5% -95% uncertainty bands for each variable reconstructed by IVM at every pollen site. From Table 6, there are bias between high and low values of some variables, which indicates the uncertainty in IVM reconstruction. However, we compared our reconstruction with previous studies in Fig. 4. Compared to PI, most reconstructions reproduced a warmer and wetter annual condition during MH, generally same as our study. In other word, this discrepancy between model-data for climate change over China during MH is common and robust in reconstructions derived from different proxies. Our study reinforces the picture given by the discrepancies between PMIP simulation and pollen data derived from a synthesis of the literature.

Line 224: The "collected data" is your data set, right? How was the comparison statistic calculated?

RE: Yes, the collected data is the dataset used in this study. The comparison statistic calculated by the match number of pollen sites. We categorized our pollen records into megabiomes, and 145 of 159 (more than 90%) pollen data match well with the BIOME 6000 during MH, while the match number is 149 for PI.

Lines 226-239: How are the changes or differences in the reconstructions calculated? As differences between the mid-Holocene reconstructions and present-day observations, or present-day inverse-approach estimates? There is considerable bias in the estimates for the present day (Table 6). How would that contribute to the mismatch between simulations and observations? Section 3.1 (throughout): No information on

the uncertainties of the reconstructions is given. These are customarily obtained from the variability of the "feasible" climate vectors generated in the optimization step in the inverse approach (e.g. Izumi and Bartlein, 2016, Fig. 3). For that matter, there is no information on the spatial variability of the simulations. Uncertainties for both could be displayed by plotting boxplots in Fig. 3, as opposed to bar graphs.

RE: The changes or differences in the paleoclimate reconstructions are calculated as the differences of biome scores between mid-Holocene and present-day times by inverse approach. And the considerable bias listed in Table 6 between observation and estimation for present-day will add the uncertainty in IVM climate reconstruction. We agree with the reviewer that boxplot is a good way to display the uncertainty, but it can't show the bias at each site. So we give the columns for every variable indicating 5-95% uncertainty bands. More detailed information about the uncertainties from reconstruction can be found in Table S4.

Line 249: "a decreasing trend" Conventionally, trends are described in the sense of a change from one time to another, or the change over a fixed period of time, so here, if the mid-Holocene MTWA values are lower than present, the trend would be positive or increasing over time (i.e. from the mid-Holocene to present). Check the discussion of precipitation trends too. It would be best to simply drop the notion of "trends" and concentrate on the change between midHolocene and PI.

RE: Corrected.

Line 265: "more detailed information about the geographic distribution of simulated temperature. . ." Section 3.1 (overall): It would be interesting to see a comparison for Pjan, the single variable with an intercept of 0.0 and a slope of 1.0.

RE: Corrected (on page 12 line 276-277). For Pjan, although its intercept and slope are most close to 0 and 1 when compared to other parameters, the amount of Pjan is small in China. As East Asian monsoon area, the annual precipitation is effected much more by summer rainfall (Pjul ) rather than winter rainfall (Pjan).

Line 273: "which would introduce a bias. . ." That's certainly plausible, but right now it's simply a conjecture.

RE: Yes, it's true that we haven't quantified the impact of different MH vegetation setting on the role of vegetation-atmosphere interaction in the MH climates among all PMIP3 models. But by giving Fig.5, Fig. 7 and Fig.8, we think that the failure to capture MH vegetation change has influence on model-data discrepancy for climate change.

Line 309: "However, none of the models succeed in capturing these features,. . ." I agree. However, the differences between the simulated and reconstructed biomes for the midHolocene simulations strike me as apparently similar in magnitude to those for the PI, and casual comparison of Figs. 5 and S7 suggests to me that some of the patterns of disagreement in the midHolocene case are inherited from the PI. This makes me wonder again about the protocol followed for generating the midHolocene simulations (see comment on line 173).

RE: According to the Taylor diagrams (Figure 1) from Jiang et al. (2016, Int. J. Climatol), the GCMs from PMIP is reliable to simulate the mean state and year-to-year variability of surface air temperature and precipitation over China for present day even without downscaling. However, the BIOME4 is a globe model, because of the particularity of vegetation types in the monsoon region of China, the BIOME4 needs further improvement of vegetation simulation accuracy in this area. This possible bias in simulating vegetation will lead to uncertainty in reconstruction. We agree with the reviewer that the downscaling is very important in applying global model into regional study, which will be taken into account in our future studies. In this study, to be able to convince the referee with quantitative arguments, we decided to conduct a supplementary experiment to demonstrate that our mechanism was appropriate. This new modeling strongly suggests vegetation changes may explain a large part of the mismatch (as shown in Fig. R1), which is very consistent with our proposal in this study. Nevertheless, there are certainly other possibilities and indeed models that better captured the hydrologic cycle and enhance the precipitation/ evaporation pattern could also explain

differences between model and data.

Line 310: What are "enhanced vegetation conditions"?

RE: The "enhanced vegetation conditions" refers to the transition from grassland to forest in the northeast during MH. We have modified it as "transition from grassland into forest".

Line 311: ". . .a cumulating inconsistency in the model-data comparisons . . . because of the vegetation-climate feedbacks." Except for the two AOV models, vegetation-climate feedback is only present in the real, as opposed to simulated, climate, i.e. in the reconstructions.

RE: Yes, all models except for the two AOV models present the real vegetation-climate feedback in PI, but they failed to present the real feedback in MH. The vegetation during MH is prescribed as PI in these 11 models, which means no change of such vegetation-climate feedback for PI and MH among them. This will lead to a cumulating inconsistency.

Line 315-316: "wetter and warmer in MTWA, colder in MTCO" This makes no sense. You might say "higher temperatures in the warmest month of the year," but did you indeed look at precipitation in the warmest month? I think what you want to say is "higher (than present) July precipitation and MTWA, lower than present MTCO" or something like that.

RE: Corrected (on page 14 lines 325-326 in revised version).

Line 318: Trend again. Data show higher-than-present MTCO during the mid-Holocene while models simulate lower-than-present MTCO.

RE: Corrected.

Line 319: My reading of Fig. 3 shows that CNRM-CM5 and HadGEM2-ES are consistent with all of the other models in simulating lower-than-present MTCO. ãĂĂ RE: We

have corrected the expression of this sentence. On page 14 line 329-330.

Line 322: "among models"

RE: Corrected (on page 14 line 338 in revised version).

Line 323-324: Replace "shed light" with "raises" (the question). ("Shedding light" implies that the variability referred to would answer the question.)

RE: Corrected (on page 14 line 335 in revised version).

Line 326: Replace "amplitude" with "amplitude and pattern". (You emphasize pattern as much as area.) Also, it's not the failure of the models to simulate vegetation change that's important, it's the fact that (apart from HadGEM2-ES and HadGEM2-CC) they can't, because the vegetation is not interactive. However, can't albedo still vary, through variations in soil color and snow cover?

RE: The "amplitude" has been replaced by "amplitude and pattern". To the second comment, yes, the albedo could vary through the variation in soil and snow cover, so we checked the monthly surface albedo change (MH-PI) of all models with prescribed vegetation. The Table enclosed below as Fig. 5 indicates that the surface albedo change caused by snow cover between two periods is very small (no more than 0.005), which could be neglected in this study. About the soil color, to our knowledge, it is prescribed as PI during MH in PMIP3, so this impact on albedo change is also negligible for our study.

Line 337: "Reconstruction showed. . ." I thought you were talking about the two AOV models. This sentence implies that you estimated the overall albedo change from the vegetation reconstruction, and compared with the two models with interactive vegetation. Is that right? If not, please explain a bit more.

RE: Yes, it's right.

Line 348-349: "should act" or "most likely would act" (We don't really know if it would.)

RE: Corrected.

Line 351-353: It may well be the case that cloud radiative feedback (or rather, inadequate simulation of that) could play a role in the data-model mismatch, but if so, that points to a completely different kind of model inadequacy, involving atmospheric circulation, moisture flux, and cloud-producing or cloud-suppressing mechanisms. Those mechanisms have been implicated in explaining the mismatch between simulations and reconstructions in the Eurasian midcontinent (Bartlein et al., 2017, GRL).

RE: It's a very important comment. We agree with your proposal that cloud radiative feedback may play a role in model-data mismatch, which indicates another kind of inadequacy. For this study, we simply focuses on the surface land change, we are not able to quantify the possible influence of mechanisms related to cloud on this model-data discrepancy for now, but we can do more in the future.

Line 354: Taylor (and fix reference too).

RE: Corrected (on page 16 line 366 in revised version).

Technical comments: I concur with the Editor and other referees that some work needs to be Corrected on the references and data-availability aspects of the paper. References: Format varies from reference to reference. Tables 4, 5 & 6: Replace commas with periods (decimal points). Fig. 7: Define dotted horizontal and vertical lines. Maps (throughout): Why does the "nine-dash line" inset vary in size and shape? I realize that the inset has to be there for geopolitical reasons, but why does it change from map to map? Fig. S7: What is the white horizontal line? SI, p. 1: Dallmeyer et al. (2017), not in references. SI, p. 3: Material at the bottom of the table is hard to read. Please reformat into a table-like arrangement. SI, p. 5: Add citations to original data sources.

RE: All these technical comments mentioned above have been done in the new manuscript.

The revised version of manuscript and supplementary information are enclosed below

as supplement.zip.

Please also note the supplement to this comment:
https://www.clim-past-discuss.net/cp-2018-145/cp-2018-145-AC5-supplement.zip

—————————————————————

[Figure]

**Figure 1.** Climate anomalies between the two experiments (6 ka and 6 ka_VEG) conducted in CESM version 1.0.5. The anomalies (6 ka_VEG-6 ka) of temperature and precipitation at both annual and seasonal scale are presented, and all these climate variables are calculated as the last 50-year means from two simulations.

**Fig. 1.**

Biomization

Biome score (MH)    Biome score (PD)

(minus)

Biome score anomalies (MH-PD)

(Artificial neural network)

Spatial pattern of biome scores anomalies (MH-PD)    Spatial pattern of biome scores (PD)

(Artificial neural network)

(overlay)

Spatial pattern of biome scores MH

(choose the highest biome score)

Spatial pattern of vegetation MH

**Figure 2**. The schematic diagram of artificial neural network.

**Fig. 2.**

[Figure]

Figure 1. Taylor diagrams displaying normalized pattern statistics of climatological (a) annual, (b) DJF, (c) MAM, (d) JJA, and (e) SON temperatures over China between 77 GCMs and observation for the period 1961–2000. The radial co-ordinate gives the standard deviation normalized by the observed value, and the angular co-ordinate gives the correlation with observation. The normalized CRMSE between a GCM and observation (marked as REF) is their distance apart. Numbers indicate GCMs listed in Table 1. Colour coding is green for TAR, orange for AR4, and blue for AR5 GCMs. Red and purple asterisks indicate the ensemble mean and the median of the 77 GCMs, respectively.

**Fig. 3.**

[Figure]

**Figure 4**. Comparison between our climate reconstruction and previous reconstruction. (a) Previous temperature results. Diamond is the qualitative reconstruction, red is the temperature increase and green is the temperature decrease; Circle is quantitative reconstruction; (b) Mean annual temperature reconstruction in this study; (c) Previous precipitation results, diamond is the qualitative reconstruction, red is the precipitation increase and green is the precipitation decrease; Circle is quantitative reconstruction; (d) Mean annual precipitation reconstruction in this study.

**Fig. 4.**

Table. The monthly albedo change caused by soil color and snow cover among models.

| | Model | Jan | Feb | Mar | Apr | May | Jun | Jul | Aug | Sep | Oct | Nov | Dec |
|---|---|---|---|---|---|---|---|---|---|---|---|---|---|
| PI | bcc-csm1-1 | 0.149 | 0.156 | 0.156 | 0.154 | 0.142 | 0.127 | 0.136 | 0.143 | 0.147 | 0.153 | 0.152 | 0.145 |
| MH | bcc-csm1-1 | 0.149 | 0.156 | 0.157 | 0.155 | 0.142 | 0.126 | 0.134 | 0.140 | 0.142 | 0.148 | 0.147 | 0.143 |
| MH-PI | anomaly | -0.001 | 0.000 | 0.001 | 0.001 | 0.000 | -0.001 | -0.001 | -0.003 | -0.004 | -0.004 | -0.006 | -0.002 |
| PI | CCSM4 | 0.170 | 0.172 | 0.173 | 0.174 | 0.164 | 0.145 | 0.143 | 0.153 | 0.159 | 0.168 | 0.168 | 0.164 |
| MH | CCSM4 | 0.170 | 0.174 | 0.175 | 0.176 | 0.166 | 0.146 | 0.142 | 0.150 | 0.156 | 0.167 | 0.166 | 0.160 |
| MH-PI | anomaly | 0.001 | 0.001 | 0.003 | 0.002 | 0.002 | 0.001 | -0.001 | -0.003 | -0.002 | -0.001 | -0.003 | -0.003 |
| PI | CNRM-CM5 | 0.151 | 0.164 | 0.164 | 0.161 | 0.146 | 0.128 | 0.130 | 0.140 | 0.143 | 0.152 | 0.152 | 0.141 |
| MH | CNRM-CM5 | 0.149 | 0.161 | 0.161 | 0.159 | 0.144 | 0.124 | 0.130 | 0.137 | 0.138 | 0.145 | 0.147 | 0.140 |
| MH-PI | anomaly | -0.002 | -0.003 | -0.003 | -0.002 | -0.002 | -0.003 | -0.001 | -0.003 | -0.005 | -0.007 | -0.004 | -0.001 |
| PI | CSIRO-Mk3-6-0 | 0.169 | 0.181 | 0.178 | 0.171 | 0.156 | 0.140 | 0.147 | 0.161 | 0.167 | 0.171 | 0.169 | 0.162 |
| MH | CSIRO-Mk3-6-0 | 0.170 | 0.180 | 0.179 | 0.172 | 0.157 | 0.139 | 0.145 | 0.161 | 0.164 | 0.168 | 0.164 | 0.162 |
| MH-PI | anomaly | 0.001 | 0.000 | 0.001 | 0.001 | 0.000 | -0.001 | -0.001 | 0.000 | -0.003 | -0.004 | -0.005 | -0.001 |
| PI | FGOALS-g2 | 0.170 | 0.172 | 0.170 | 0.168 | 0.154 | 0.137 | 0.141 | 0.155 | 0.156 | 0.164 | 0.168 | 0.161 |
| MH | FGOALS-g2 | 0.172 | 0.175 | 0.173 | 0.173 | 0.159 | 0.141 | 0.142 | 0.156 | 0.158 | 0.167 | 0.168 | 0.163 |
| MH-PI | anomaly | 0.002 | 0.003 | 0.004 | 0.005 | 0.005 | 0.004 | 0.001 | 0.002 | 0.002 | 0.003 | 0.000 | 0.002 |
| PI | FGOALS-s2 | 0.165 | 0.173 | 0.171 | 0.170 | 0.161 | 0.148 | 0.153 | 0.164 | 0.165 | 0.168 | 0.166 | 0.161 |
| MH | FGOALS-s2 | 0.164 | 0.174 | 0.173 | 0.172 | 0.162 | 0.142 | 0.150 | 0.159 | 0.160 | 0.165 | 0.160 | 0.156 |
| MH-PI | anomaly | 0.000 | 0.001 | 0.002 | 0.002 | 0.001 | -0.006 | -0.003 | -0.005 | -0.005 | -0.004 | -0.005 | -0.005 |
| PI | GISS-E2-R | 0.144 | 0.153 | 0.154 | 0.150 | 0.131 | 0.114 | 0.119 | 0.126 | 0.132 | 0.138 | 0.138 | 0.134 |
| MH | GISS-E2-R | 0.144 | 0.153 | 0.153 | 0.149 | 0.131 | 0.114 | 0.111 | 0.122 | 0.124 | 0.129 | 0.132 | 0.130 |
| MH-PI | anomaly | 0.000 | -0.001 | -0.001 | 0.000 | 0.000 | 0.000 | -0.007 | -0.004 | -0.008 | -0.009 | -0.006 | -0.003 |
| PI | IPSL-CM5A-LR | 0.149 | 0.158 | 0.161 | 0.159 | 0.148 | 0.133 | 0.140 | 0.149 | 0.152 | 0.155 | 0.148 | 0.142 |
| MH | IPSL-CM5A-LR | 0.149 | 0.158 | 0.162 | 0.160 | 0.149 | 0.133 | 0.140 | 0.147 | 0.150 | 0.152 | 0.144 | 0.139 |
| MH-PI | anomaly | 0.000 | 0.000 | 0.001 | 0.001 | 0.001 | 0.000 | 0.000 | -0.001 | -0.002 | -0.003 | -0.004 | -0.002 |
| PI | MIROC-ESM | 0.162 | 0.171 | 0.173 | 0.167 | 0.146 | 0.123 | 0.129 | 0.136 | 0.143 | 0.153 | 0.155 | 0.151 |
| MH | MIROC-ESM | 0.164 | 0.174 | 0.176 | 0.172 | 0.150 | 0.124 | 0.131 | 0.137 | 0.144 | 0.154 | 0.154 | 0.152 |
| MH-PI | anomaly | 0.002 | 0.002 | 0.003 | 0.005 | 0.004 | 0.001 | 0.002 | 0.000 | 0.000 | 0.001 | -0.002 | 0.001 |
| PI | MPI-ESM-P | 0.170 | 0.164 | 0.160 | 0.155 | 0.153 | 0.155 | 0.151 | 0.145 | 0.145 | 0.151 | 0.161 | 0.173 |
| MH | MPI-ESM-P | 0.169 | 0.163 | 0.160 | 0.157 | 0.154 | 0.154 | 0.148 | 0.141 | 0.141 | 0.147 | 0.160 | 0.171 |
| MH-PI | anomaly | -0.001 | -0.001 | 0.000 | 0.001 | 0.001 | -0.001 | -0.003 | -0.004 | -0.004 | -0.004 | -0.001 | -0.002 |
| PI | MRI-CGCM3 | 0.183 | 0.194 | 0.194 | 0.190 | 0.173 | 0.156 | 0.158 | 0.172 | 0.176 | 0.187 | 0.189 | 0.180 |
| MH | MRI-CGCM3 | 0.183 | 0.195 | 0.196 | 0.191 | 0.174 | 0.154 | 0.156 | 0.168 | 0.172 | 0.183 | 0.183 | 0.175 |
| MH-PI | anomaly | 0.000 | 0.001 | 0.001 | 0.002 | 0.001 | -0.001 | -0.002 | -0.003 | -0.005 | -0.004 | -0.005 | -0.005 |

**Fig. 5.**

---

## Author Response (AR1)

**Response to the editor and reviewers**

We greatly appreciate the constructive comments and suggestions on the previous version of the manuscript from the editor and reviewers. We have attempted to address every point raised. The following is the point-by-point reply, with reference to the order of the comments made by the editor and reviewers.

The main changes we have made in the revised version are listed as below:

1. The referee suggested to add more information about our reconstruction in this study. We have provided the Table 1(on pages 42-45), Table S4 (on pages 5-8 in Supplementary Information), and Table S5 (on pages 8-13 in Supplementary Information) to show the detailed information of the pollen site, the biome scores of each record and the reconstructed result of each variable from IVM. We also give the boxplots of temperature and precipitation anomaly (MH-PI) in Figure 3 and Figure 4 to better illustrate the discrepancy between model-data of climate change over China during MH.

2. Following the referee comment, we did the cross-proxy validation of our reconstruction. In the revised version, we compared our reconstruction with previous studies over China based on multiple proxies (including pollen, lake core, palaeosol, ice core, peat and stalagmite). Compared to PI, most reconstructions reproduced a warmer and wetter annual condition during MH, same as our study. In other word, this discrepancy between model-data for climate change over China during MH is common and robust in reconstructions derived from different proxies. The results are shown in Fig. 7 on page 58 in the revised paper.

3. The referee raised an important issue on "One way to test the proposal of the authors would be prescribing the reconstructed vegetation in a climate model to see how the model results would be altered." Although we can't prescribe the MH vegetation with our reconstructed results in all PMIP3 models, we have succeeded to conduct such test in CESM version 1.0.5. The new-added Fig.9 (on page 60 in the revised paper) shows the climate anomalies between two simulations (6 ka_VEG minus 6 ka), for both annual and seasonal scale. For temperature, it's clear that the 6 ka_VEG

simulation reproduces the warmer annual (~0.3 K on average) and winter temperature (~0.6 K on average), especially the winter temperature. For precipitation, the reconstructed vegetation leads to higher annual and seasonal precipitation, which can also reconcile the discrepancy of increase amplitude for precipitation during MH between model-data (data reproduced larger amplitude than model, revealed by our study). So it's true that the mismatch between model-data in MH vegetation has significant influence on the discrepancy of climate, this is consistent with our proposal in this study.

4. Two referees indicated that the abstract need to provide more information from this work, we added sentences "Our results indicate that the main discrepancies between model and data for the MH climate are the annual and winter mean temperature. A warmer-than-present climate condition is derived from pollen data for both annual mean temperature (~0.7 K on average) and winter mean temperature (~1 K on average), while most of the models provide both colder-than-present annual and winter mean temperature and a relatively warmer summer, showing linear response driven by the seasonal forcing. By conducting simulations in BIOME4 and CESM, we show that the surface processes are the key factors drawing the uncertainties between models and data. These results pinpoint the crucial importance of including the non-linear responses of the surface water and energy balance to vegetation changes" on pages 1-2 lines 22-31 in the revised paper.

5. Following the referee's comment, we added the model-data comparison of Pjan in the revised paper on page 55 in the revised paper.

6. All the references mentioned in our paper are included in the reference list in the revised paper.

Other modifications for specific comments are indicated in each response and marked as underlined words in the marked-up manuscript below.

**Response to the editor**

**1. Archiving data**

**Input data**

**Comments:** Ideally, the pollen assemblages for each site, at least for the 6ka should be made available (e.g. in trusted archives). The biographic reference must be included for each of the sites. Many citations in Table 1 are not listed in the reference, list.

**Response:** The biographic reference for each site is included in Table 1 (on pages 42-45 in revised version), and the citations have been added in the reference list. About the pollen assemblages, we provide the biome scores at 6 ka for each pollen site in Table S4 (on pages 5-8 in revised version). The full dataset of pollen records used in our study are from Li et al. (2019), it is available upon the request.

**Comments:** All the original data must be archived in a vetted repository and a citation should be linked to them.

**Response:** For the 94 collected pollen records, we give the reference of each pollen record in Table 1, and we also give the Biome type at each site in Table S5, as well as the biome scores (Table S4). For the 65 pollen data from Chinese Quaternary Pollen Database, we need to obey the data usage rule, and the detailed information for these pollen records is not available since they are not public data yet.

**Comments:** Climate simulations: the citations in Table 3 are not listed in the reference list. This should be amended.

**Response:** The citations in Table 3 are added in reference list (line 888-890, 536-539, 803-808, 581-583, 632-636, 437-441, 740-753, 504-508, 515-525, 836-839, 540-548, 918-922 in revised version).

**Output data**

**Comments:** all reconstructed climate variables and the biomes for each site at 6ka should be archived (the values for the colored dots in Figs. 2-4).

**Response:** these values are provided in Table S5 (on pages 8-13 in revised version).

**Comments:** the simulated values for each of the climate variables (the colored bars in Figs. 3-4).

**Response:** The simulated values for climate variables from each model are provided in Table S6 and Table S7 on pages 13-14 in Supplementary Information.

**Comments:** basic metadata for each site (the information listed in Table 1)

**Response:** Provided in Table 1.

**2. The "data availability" section**

**Comments:** it should include the doi/url that links to the output data from this study and the reiterates the location of the input data (above). Alternatively, locations of the input data can be provided in the table.

**Response:** The data from Chinese Quaternary Pollen Dataset are not available yet, for other data used in our study, we prefer to provide them directly in a Table (like Table S4, Table S5) in the SI instead of putting them in a vetted repository.

**3. Other reminders**

**Comments:** Of course, the authors might also want to make the digital data for their maps available, or anything that they think might be useful for future studies.

**Response:** For now, we think all the information about our input and output data are enough, of course, we are open-minded to other possible requests.

**Comments:** A special attention to the reference list is required. The references marked as red in the attached file are missing in the reference list. The author should also check lines 483, 637 and 694 of their reference list. Is CQPD 2000 or 2001 or both? And also please note the supplement to this comment: http://www.clim-past-discuss.net/cp-2018-145-Ec1-supplement.pdf.

**Response:** It's CQPD 2000, and all these missing and errors in the reference list are modified in the revised version.

**Reference**

Li, Q., Wu, H., Yu, Y., Sun, A., and Luo, Y.: Quantifying regional vegetation changes in China during three contrasting temperature intervals since the last glacial maximum, Journal of Asian Earth Sciences, http://doi.org/10.1016/j.jseaes.2018.10.013.

**Response to Reviewer #1**

**Comments from Reviewer #1:**

The model-data comparison of climate change during mid-Holocene (MH) is an important issue to validate the results from Global Circulation Model (GCM) against the proxies gathered from dataset. Based on the new pollen dataset and Inverse Vegetation Model (IVM). This study provided a quantitative reconstruction of climate variables during MH over China was provided and compared to the simulation results from 13 models in PMIP3. A large discrepancy on the temperature anomaly between model-data at both annual and seasonal scale was depicted, mainly due to the failure of capturing vegetation change during MH by models, which is very helpful for better understanding the climatic changes during MH, and also pinpoints the possible way to reconcile model and data by accurately simulating the non-linear responses of vegetation and hydrology in GCMs. The manuscript can be accepted for publication after minor revision. A few basic comments and some issues to deal with as follow:

1.  Since it's a quantitative model-data comparison based on pollen dataset, in which 91 records were digitized from published papers. More detailed information about the data should be provided, like the age control, pollen assemblages from around 6 ka at each site.

**Response:** the required information has been added in the Table 1 (page 45-48) and Table S5 (page 8-13 in Supplementary Information).

2. As mentioned in the manuscript, there is a difference in vegetation inputs for the MH period among models in PMIP3, a table for detail information should be given.

**Response:** we added a new Table S8 in the supplementary information (page 15 in Supplementary Information).

**Table S5. Vegetation setting for the mid-Holocene among models in PMIP3**

| Model | L A I | Stomatal Resistance Function Of | Vegetation Time Variation |
|---|---|---|---|
| CCSM4 | Prognostic | CO2 \| Light \| Temperature \| Water availability | Prescribed (varying from files) |
| MIROC-ESM | Prescribed | CO2 \| Light \| Temperature \| Water availability | Prescribed (varying from files) |
| BCC-CSM1.1 | Prognostic | CO2 \| Light \| Temperature \| Water availability | Prescribed (varying from files) |
| CNRM-CM5 | Prescribed | Light \| Temperature \| Water availability | Fixed (not varying) |
| CSIRO-MK3.6.0 | Prescribed | Light \| Temperature \| Water availability | Prescribed (varying from files) |
| GISS-E2-R | Prescribed | CO2 \| Light \| Temperature \| Water availability | Fixed (not varying) |
| IPSL-CM5A-LR | Prognostic | CO2 \| Light \| Temperature \| Water availability | Prescribed (varying from files) |
| MPI-ESM-P | Prognostic | CO2 \| Water availability | Fixed (not varying) |
| MRI-CGCM3 | Prescribed | CO2 \| Light \| Water availability | Prescribed (varying from files) |
| HadGEM2-ES | Prognostic | CO2 \| Light \| Temperature \| Water availability | Dynamical (varying from simulation) |
| HadGEM2-CC | Prognostic | CO2 \| Light \| Temperature \| Water availability | Dynamical (varying from simulation) |
| FGOALS-g2 | Prescribed | no data | Prescribed (varying from files) |
| FGOALS-s2 | Prescribed | no data | Prescribed (varying from files) |

3. The disparity of temperature anomaly during MH among models could be resulted from the difference in pre-industrial (PI) simulation. Authors should prove that there is no any clear relationship between PI temperature and temperature change (MH-PI).

**Response:** Fig.R1, as attached below, demonstrates that there is no any clear relationship between PI temperature and temperature change (MH-PI), for both annual and seasonal scale, which means the disparity of temperature anomaly during MH among models doesn't come from the difference in PI simulation.

[Figure]

Fig.R1 The relationship between PI temperature and temperature change (MH-PI)

4. Some references are missing in the reference list. Such as the citations in Table 3.

**Response:** we have added the citations from Table 3 in the reference list (line 943-945, 582-585, 869-874, 629-631, 685-689, 476-480, 804-812, 543-547, 558-568, 896-899, 586-594, 973-977 in revised version) .

**Response to Reviewer #2**

**General Comments:** Data-model comparison is often problematic especially at regional scale, for which there are many reasons. This paper presents an interesting analysis to investigate the possible impact of poor representation of vegetation in climate models on the model-data discrepancy over China. The authors compare the PMIP3 results with their "reconstruction" and propose that lack of vegetation dynamics is the main reason of model-data discrepancy in seasonal climate over China. The

results are clearly explained and the paper is well written. Especially, a large amount of data are presented and would be a remarkable contribution to the Holocene study. I would recommend its publication after the following comments are considered:

1. At least to my knowledge, regional diversity exists inside China regarding the timing of the mid-Holocene thermal maximum. However, the insolation of 6 ka BP is used in the PMIP3 simulations. In which degree might the model-data discrepancy be related to the forcing used in the climate models? Have the authors compared their reconstructions to simulations of other periods like 9 ka and 12 ka or to transient simulations to see whether the data-model comparison can be improved if different forcing is considered?

**Response:** Yes, there is a regional diversity exists over China regarding the timing of mid-Holocene thermal maximum (from ~8 ka to 4 ka). But our paper is focused on mid-Holocene time (defined as 6 ka in PMIP), not MH thermal maximum, and we also selected the pollen data at $6\pm0.5$ ka. So for us, in term of the consistency in time, it's better to do a model-data comparison with the 6 ka isolation forcing used in PMIP3, rather than 9 ka or 12 ka.

For the transient simulation, Liu et al.(2014) analyzed the model results from 22 ka in three coupled ocean-atmosphere models: CCSM3, FAMOUS and LOVECLIM. It turns out that all three models reproduced a colder than present annual mean temperature during Holocene, no matter at 12 ka, 9 ka or 6 ka, which is consistent with our results. And we agree with the reviewer that 9 ka and 12 ka are also very important periods to understand the mechanism of climate change during MH, we plan to do this comparison in the future work.

2. One way to test the proposal of the authors would be prescribing the reconstructed vegetation in a climate model to see how the model results would be altered and whether the model would reproduce more realistic results when compare to other proxy data.

**Response:** We agree with the reviewer, it's a very efficient way to test our proposal if

we can run the simulation with the reconstructed vegetation in GCMs. However, as far as we know, prescribing the vegetation in a coupled GCM is not easy. Moreover, the GCM models in PMIP3 have their own vegetation module, it definitely takes much time to do such test, that's why in this paper we choose BIOME4 to evaluate MH vegetation simulation against the reconstructed result.

Although we can't prescribe the MH vegetation with our reconstructed results in all PMIP3 models, we succeeded to conduct such test in CESM version 1.0.5. This version, developed at the National Center for Atmospheric Research, is a widely used coupled model with dynamic atmosphere (CAM4), land (CLM4), ocean (POP2), and sea-ice (CICE4) components (Gent et al., 2011). Here, we use ~2° resolution for the CAM4, configured by ~1.9° (latitude) × 2.5° (longitude) in the horizontal direction and 26 layers in the vertical direction. The POP2 adopts a finer grid, with a nominal 1° horizontal resolutions and 60 layers in the vertical direction. The land and sea-ice components have the same horizontal grids as the atmosphere and ocean components, respectively.

Two experiments were conducted, including a mid-Holocene (MH) experiment (6 ka) with original vegetation setting (prescribed as PI vegetation for MH) and a MH experiment with reconstructed vegetation (6 ka_VEG). In detail, experiment 6 ka used the MH orbital parameters (Eccentricity=0.018682; Obliquity=24.105°; Angular precession=0.87°) and modern vegetation (Salzmann et al., 2008). Compared to experiment 6 ka, experiment 6 ka_VEG used our reconstructed vegetation in China. Except for the changed vegetation, all other boundary conditions were kept unchanged in these two experiments, including the solar constant (1365 W m−2), modern topography and ice sheet, and pre-industrial greenhouse gases ($CO_2$ = 280 ppmv; $CH_4$ = 760 ppbv; $N_2O$ = 270 ppbv). Experiment 6 ka was initiated from the default pre-industrial simulation and run for 500 model years. Experiment 6 ka_VEG was initiated from model year 301 of experiment 6 ka and run for another 200 model years. We analyzed the computed climatological means of the last 50 model years from each experiment here.

Fig. 9 in the revised paper (enclosed as below) shows the climate anomalies between

two simulations (6 ka_VEG minus 6 ka), for both annual and seasonal scale. For temperature, it's clear that the 6 ka_VEG simulation reproduces the warmer annual (~0.3 K for grid mean) and winter temperature (~0.6 K for grid mean), especially the winter temperature. For precipitation, the reconstructed vegetation leads to higher annual and seasonal precipitation, which can also reconcile the discrepancy of increase amplitude for precipitation during MH between model-data (data reproduced larger amplitude than model, revealed by our study). So it's true that the mismatch between model-data in MH vegetation has significant influence on the discrepancy of climate, this is consistent with our proposal in this study.

Each model has different sensitivity to the boundary change, further work should be carried out in more models to test the influence of vegetation on climate, this is an ongoing work.

This response is on pages 17-18, lines 406-426 in the revised version.

[Figure]

**Figure 9.** Climate anomalies between the two experiments (6 ka and 6 ka_VEG) conducted in CESM version 1.0.5. The anomalies (6 ka_VEG-6 ka) of temperature and precipitation at both annual and seasonal scale are presented, and all these climate variables are calculated as the last 50-year means from two simulations.

3. The spatial resolution of all the GCMS is very coarse when regional diversity within China is considered. The regional details of topography are not necessarily well represented. I wonder in which degree the model-data mismatch is related to rough topography used in the climate models.

**Response:** Thanks for the important comment, yes, we should consider the possible influence of rough topography on the model-data discrepancy, especially the Tibetan

Plateau (TP). Numerical simulations had been widely utilized to investigate the climate response of the uplift of TP, and it is also indicated by previous studies that most of the experiments use coarse resolution GCMs have deficiency in describing the small-scale topography and hence climate (Wang et al., 2005; Gao et al., 2008; Jiang et al., 2016).

In PMIP3, the topography for mid-Holocene is same as CMIP5 PI, and thus, each model has the same topography boundary for both MH and PI, the difference between them concerning the topography is the interpolation due to their different resolutions. Among the 13 models used in this paper, MRI-CGCM3 has the highest resolution (Atmosphere: 320*160*L48; Ocean: 364*368*L51), while IPSL-CM5A-LR has the lowest one (Atmosphere: 96*96*L39; Ocean: 182*149*L31). The possible influence of topography on the model-data mismatch captured in our study could be tested from two points:

Firstly, for the model with high resolution. In Fig. R2 (enclosed below), we give the actual modern orography and the interpolated orography used in MRI-CGCM3 and IPSL-CM5A-LR. For MRI-CGCM3, the topography is very close to the observation, so for this model, the model-data discrepancy during MH over China is not related to rough topography.

[Figure]

**Figure R2.** The topography comparison between models and observation

Secondly, for the model with course resolution. When we compare the topography of observation and that used in IPSL-CM5A-LR, it's true that the course version of model will lead to biases in topography when the regional diversity is discussed. To quantify such influence, we compare the results of IPSL-CM5A-LR and IPSL-CM5A-MR (Fig. R3). The difference in topography caused by model resolution has influence on some small regional climate, but no significant change for general pattern.

[Figure]

**Figure R3.** The preindustrial climate comparison between simulation and observation. Tas means temperature above 2m surface, pr means precipitation.

4. Line 372: the authors consider the poor capacity of vegetation modelling in climate models to be the major reason for model-data discrepancy. Before the author test for other reasons like those related to topography, soil types, selected climate forcing, I am not very convinced that vegetation is the major reason.

**Response:** For the selected climate forcing and topography, we have already gave the answer above (in question 1 and question 3). For soil type, to our knowledge, there is no relative research to quantify the soil type effect on climate over China during MH. But it's certainly true that soil type change could lead to climate anomaly through surface albedo variation and hydrology processes. For instance, if the bare soil transferred into vegetated soil, the regional climate could be warmer due to the decreased surface albedo. But for the further quantification of this contribution, it will

be done in the future work.

In this paper, we only focus on the vegetation influence on the model-data discrepancy, and we do present and prove the mismatch between model-data in MH vegetation could partly account for the discrepancy of climate. We don't emphasis that the vegetation influence is the main reason. And according to your suggestion, we add the sentence: "Moreover, besides the vegetation influence, to which extent this model-data discrepancy is related to rough topography, soil type and other possible factors should be investigated in the future work." (on page 19, line 440-442).

5. Two models of 13 use dynamical vegetation model. According to your analyses, is there obvious advantage of using AOV instead of AO?

**Response:** As we mentioned in the manuscript, only 2 models (HadGEM2-ES and HadGEM2-CC) in PMIP3 has the dynamic vegetation simulation for mid-Holocene. However, the main vegetation changes during MH demonstrated by these two models are very different. HadGEM2-ES simulated increased tree coverage (~15%) and a decreased bare soil fraction (~6%), while HadGEM2-CC depicts a ~3% decrease in tree fraction and a ~1% increase in bare soil (Fig. S9 in supplementary information, on page 27 in the revised paper).

We made a rough calculation of albedo variance caused solely by vegetation change for both two models and for our reconstruction, based on the area fraction and albedo value of each vegetation type (Betts, 2000; Bonfils et al., 2001; Oguntunde et al., 2006; Bonan, 2008). Reconstruction showed vegetation changes during MH leading to a ~1.8% decrease in albedo when snow-free, with a much larger impact (~4.2% decrease) when snow-covered. The results from HadGEM2-ES are highly consistent with the albedo changes from the reconstruction, featuring a ~1.4% (~6.5%) decrease without (with) snow, while HadGEM2-CC produces an increased albedo value during MH (~0.22% for snow-free, ~1.9% with snow-cover), depending on its vegetation simulation.

The difference in simulating MH vegetation distribution between these two AOVGCM will influence their ability in capturing the climate change during MH. From Fig. 3 in the manuscript (on page 54), we can see that HadGEM2-ES succeeded to capture the increased annual temperature anomaly (~0.42 K), with relatively higher MTWA and MTCO among models, while HadGEM2-CC showed similar results with other models.

In conclusion, according to our analysis, there is an obvious advantage of using AOVGCM instead of AOGCM when we discussing about the MH climate, but the premise is that the AOVGCM can simulate accurate vegetation distribution.

**6.** Line 36: do you mean "an increase in the seasonal cycle of insolation"?

**Response:** Thanks for the correction, we mean "an increase in the amplitude of the seasonal cycle of insolation" in Line 36, and that has been modified in the new version of manuscript on page 2 line 40 in revised version.

7. I wonder how the pollen data of PI was collected. Were they collected from the surface? Is there any influence from human activities?

**Response:** Yes, the pollen data of PI are collected from the surface, but we only choose the sites without or with little influence of human activity. Moreover, when we collected the pollen data, we abandon the pollen records if the published paper mentions the influence of human activity (on page 5 line 113 in revised paper), so the key point here is that weather can we trust the climate reconstruction if the pollen records are influenced by human activities during PI. To clarify this issue, we have examined the statistical correlations between observed meteorological values and reconstructed climates by inverse vegetation model at the sample sites for PI (Table 6 in manuscript). The regression coefficients are very high (from 0.75 to 0.95), which means that IVM is able to reconstruct the PI climates over China based on the pollen data. In other words,

the PI pollen data are reliable to obtain the climate parameters, and the human activity has no significant effect on it.

8.   In Figure 3: is the anomaly relative to PI? How was the grid mean value calculated?

**Response:** Yes, it's the anomaly relative to PI. About the grid mean value, we firstly extract the simulated values at each pollen site, and then calculate the grid mean value in ncl.

9.   Line 320: PMIP3

**Response:** Thanks, we have modified it.

**Reference:**

Wang, B., Ding, Q., Fu, X., Kang, I., Jin, K., Shukla, J., Francisco, D.: Fundamental challenge in simulation and prediction of summer monsoon rainfall, Geophysical Research Letters, 32, L15711, 2005.

Gao, X., Shi, Y., Song, R., Giorgi, F., Wang, Y., Zhang, D.: Reduction of future monsoon precipitation over China: comparison between a high resolution RCM simulation and the driving GCM. Meteorology and Atmospheric Physics, 100, 73–86, 2008.

Jiang, D., Tian, Z., Liang, X.: Reliability of climate models for China through the IPCC Third to Fifth Assessment Reports, International Journal of Climatology, 36, 1114–1133, 2016.

Betts, R. A.: Offset of the potential carbon sink from boreal forestation by decreases in surface albedo, Nature, 408, doi: 10.1038/nature. 35041545, 2000.

Bonfils, C., de Noblet-Ducoudré, N, Braconnot, P., and Joussaume, S.: Hot Desert Albedo and Climate Change: Mid-Holocene Monsoon in North Africa, Journal of Climate, 14, 3724–3737, 2001.

Bonan, G. B.: Forests and Climate Change: Forcings, Feedbacks, and the Climate

Benefits of Forests, Science, 320, 1444-1449, 2008.

Oguntunde, P. G., Ajayi, A. E., and Giesen, N.: Tillage and surface moisture effects on bare-soil albedo of a tropical loamy sand, Soil and Tillage Research, 85, 107-114, 2006.

**Response to Reviewer #3**

**General comments:** The manuscript entitled "Mid-Holocene climate change over China: model-data discrepancy" by Lin et al. presented a study on model-data comparison by using the pollen data collection in China and PMIP3 mid-Holocene simulations. From the large discrepancy showed in model-data comparison, both in annual mean, warmest month and coldest month, they conclude that the major reason that PMIP3 simulations do not agree with data is because the vegetation distribution is not properly represented in climate models, where most models do not include dynamical vegetation and the prescribed MH vegetation map is the same as preindustrial. The MH vegetation issues have been recognized in recent years and many efforts are made to reconstruct a better MH land cover map, this includes the PAGES working group on Landcover6k. Therefore, a good vegetation map from China would be expected to contribute to an eventual global land-cover map during the mid-Holocene and benefit the paleoclimate community. However, the current work has a somewhat mislead focus and I have the following major concerns.

1.  The reconstructed mid-Holocene climate in their study is largely depend on the pollen data collection. I am not an expert on pollen data, but I am wondering if all the published data use the same standard on data process. Can they be synthesized by Webb1-7 standard and put together for comparison? I hope a reviewer from pollen community may have some insights on the data process. There are no discussion on the potential uncertainties on collected data, at least one comparison with other proxy data can provide the cross-proxy verification. The authors emphasized three original data but no detailed information, which are important if

they are not published. When the significant differences are found in model-data comparisons, the uncertainties from the data should be discussed as well. One can not regard reconstruction is the truth. We need to know how reliable is the reconstructed climate from pollen data, given that the IVM method used to reconstruct the climate is a crude estimate. Otherwise it is dangerous if this paper is published and people take for granted that this is the climate (and vegetation map) in China during mid-Holocene.

**Response:** Thanks for this very important comment, we will answer it point by point:

**Firstly, can the pollen records collected from papers be synthesized by Webb 1-7?**

Yes, we need to do some data processing before we use the collected pollen records to reconstruct the climates. Firstly, the published papers only give the pollen diagram, not the pollen assemblages. So we need to digitize the pollen diagram for obtaining the pollen assemblages, and then use biomization to get the biome scores and biome types. Secondly, for age control, different dating methods are used in the collected pollen records, we use CalPal 2007 (Weninger et al., 2007) to correct $^{14}C$ age into calendar age so that they can be contrasted with each other. For lacustrine records, if the specific carbon pool age is mentioned in the literature, the calendar age is corrected after deducting the carbon pool. Otherwise, the influence of carbon pool is not considered. The age series of records were obtained by linear regression or linear interpolation of adjacent dating data. After these preprocessing, a unified chronological standard for all pollen records is built, and then the classification of age control followed the standard of Webb 1-7.

The description of these data processing are added in the revised manuscript from on pages 5-6 lines 115 to line 124 in the revised paper.

**Secondly, what's the potential uncertainties on data reconstruction? How reliable are the pollen data used in this study? How about the cross-proxy verification?**

For the reliability of pollen data, we have compared them with the BIOME6000 (Fig. 2, page 53), the match between collected data and the BIOME6000 is more than 90% for both MH and PI.

For the potential uncertainties on data reconstruction, IVM is relied on the BIOME4, a global vegetation model. Because of the particularity of vegetation types in the monsoon region of China, the BIOME4 needs further improvement of vegetation simulation accuracy in this area. This possible bias in simulating vegetation will lead to uncertainty in reconstruction. In this version, we added more information in Table S5 in supplementary information (on page 8-13 in the SI), we gave the climate variables reconstructed from IVM at each site. Moreover, we also showed the bias on data reconstruction by giving the median value (for instance, column named MTCO) and values indicating the 5% (MTCO1)-95% (MTCO2) uncertainty bands.

For the cross-proxy verification, we compared our reconstruction with previous studies over China based on multiple proxies (including pollen, lake core, palaeosol, ice core, peat and stalagmite). Compared to PI, most reconstructions reproduced a warmer and wetter annual condition during MH (Fig. 7 as below, on page 58 in revised paper), same as our study. In other word, this discrepancy between model-data for climate change over China during MH is common and robust in reconstructions derived from different proxies. Our study just reinforces the picture given by the discrepancies between PMIP simulation and pollen data derived from a synthesis of the literature.

[Figure]

Figure 8. Comparison between our climate reconstruction and previous reconstruction. (a) Previous temperature results. Diamond is the qualitative reconstruction, red is the temperature increase and green is the temperature decrease; Circle is quantitative reconstruction; (b) Mean annual temperature reconstruction in this study; (c) Previous precipitation results, diamond is the qualitative reconstruction, red is the precipitation increase and green is the precipitation decrease; Circle is quantitative reconstruction; (d) Mean annual precipitation reconstruction in this study.

Thirdly, about the three original data, thanks for the reminder. They have been published, and we added the information in Table 1, as well as the reference list (lines 1011-1013, page 44 and lines 921-924, page 40).

2. The BIOME4 produced vegetation pattern in fig5 is determined by the input climate variables from the model, given the supplementary figures s1-s6 and previous studies by Jiang et al. (2012) have already show different climate patterns produced by different models, therefore the mismatch in vegetation pattern and reconstructed map in Fig5 is expected. I don't think this mismatch can be used to argue that the modelled MH climate is not good because they did not use a correct vegetation map and include the vegetation-climate interaction. Those vegetation patterns produced by BIMOE4 are not used in PMIP experiment setup, it would make more sense if authors compare the reconstruction and PMIP prescribed land cover map, or compare BIOME4 produced vegetation map with the ones produced by those climate models (for example HadGEM2-ES) that have dynamical vegetation to gain some understanding on vegetation-climate feedback.

**Response:** We totally agree with the reviewer, it's a very efficient way to test our proposal if we can run the simulation with the reconstructed vegetation in GCM. However, as far as we know, prescribing the vegetation in a coupled GCM is not easy. For instance, if we want to use the reconstructed vegetation in Orchidee (the vegetation module of IPSL), we need to modify numerous parameters to make sure that the experiment with new vegetation condition will not be killed by the model due to its mismatch in climate variables. Moreover, the GCM models in PMIP3 have their own vegetation module, it definitely takes much time to do such test, that's why in this paper

we choose BIOME4 to evaluate MH vegetation simulation against the reconstructed result. Actually, we already plan to conduct this experiment in Orchidee to further test the proposal of this paper, it's an ongoing work.

Although we can't prescribe the MH vegetation with our reconstructed results in all PMIP3 models, we have succeeded to conduct such test in CESM version 1.0.5. The CESM version 1.0.5, developed at the National Center for Atmospheric Research, is a widely used coupled model with dynamic atmosphere (CAM4), land (CLM4), ocean (POP2), and sea-ice (CICE4) components (Gent et al., 2011). Here, we use ~2° resolution for the CAM4, configured by ~1.9° (latitude) × 2.5° (longitude) in the horizontal direction and 26 layers in the vertical direction. The POP2 adopts a finer grid, with a nominal 1° horizontal resolution and 60 layers in the vertical direction. The land and sea-ice components have the same horizontal grids as the atmosphere and ocean components, respectively.

Two experiments were conducted, including a mid-Holocene (MH) experiment (6 ka) with original vegetation setting (prescribed as PI vegetation for MH) and a MH experiment with reconstructed vegetation (6 ka_VEG). In detail, experiment 6 ka used the MH orbital parameters (Eccentricity=0.018682; Obliquity=24.105°; Angular precession=0.87°) and modern vegetation (Salzmann et al., 2008). Compared to experiment 6 ka, experiment 6 ka_VEG used our reconstructed vegetation in China. Except for the changed vegetation, all other boundary conditions were kept unchanged in these two experiments, including the solar constant (1365 W m$^{-2}$), modern topography and ice sheet, and pre-industrial greenhouse gases ($CO_2$ = 280 ppmv; $CH_4$ = 760 ppbv; $N_2O$ = 270 ppbv). Experiment 6 ka was initiated from the default pre-industrial simulation and run for 500 model years. Experiment 6 ka_VEG was initiated from model year 301 of experiment 6 ka and run for another 200 model years. We analyzed the computed climatological means of the last 50 model years from each experiment here.

The new-added Fig.9 on page 60 in the revised paper (enclosed below) shows the climate anomalies between two simulations (6 ka_VEG minus 6 ka), for both annual and seasonal scale. For temperature, it's clear that the 6 ka_VEG simulation reproduces

the warmer annual (~0.3 K on average) and winter temperature (~0.6 K on average), especially the winter temperature. For precipitation, the reconstructed vegetation leads to higher annual and seasonal precipitation, which can also reconcile the discrepancy of increase amplitude for precipitation during MH between model-data (data reproduced larger amplitude than model, revealed by our study). So it's true that the mismatch between model-data in MH vegetation has significant influence on the discrepancy of climate, this is consistent with our proposal in this study.

Each model has different sensitivity to the boundary change, further work should be carried out in more models to test the influence of vegetation on climate, this is an ongoing work.

[Figure]

**Figure 9.** Climate anomalies between the two experiments (6 ka and 6 ka_VEG) conducted in

CESM version 1.0.5. The anomalies (6 ka_VEG-6 ka) of temperature and precipitation at both annual and seasonal scale are presented, and all these climate variables are calculated as the last 50-year means from two simulations.

**Specific comments:**

1. The abstract need to provide more information from this work, now only contains motivation and conclusion. And the conclusion in abstract actually is a speculation, did not come from the results of this work.

**Response:** We modified the abstract as following (on page 1-2 line 22-31 in the revised paper):

The mid-Holocene period (MH) has long been an ideal target for the validation of Global Circulation Model (GCM) results against reconstructions gathered in global datasets. These studies aimed to test the GCM sensitivity mainly to the seasonal changes induced by the orbital parameters (precession). Despite widespread agreement between model results and data on the MH climate, some important differences still exist. There is no consensus on the continental size of the MH thermal climate response, which makes regional quantitative reconstruction critical to obtain a comprehensive understanding of the MH climate patterns. Here, we compare the annual and seasonal outputs from the most recent Paleoclimate Modelling Intercomparison Projects Phase 3 (PMIP3) models with an updated synthesis of climate reconstruction over China, including, for the first time, a seasonal cycle of temperature and precipitation. Our results indicate that the main discrepancies between model-data for MH climates are the annual and winter mean temperature. A warmer-than-present climate condition are derived from pollen data for both annual mean temperature (~0.7 K on average) and winter mean temperature (~1 K on average), while most of the models provide a linear response driven by the seasonal forcing (a decreased annual mean temperature with a warmer summer and colder winter). By conducting simulations in BIOME4 and CESM, we show that to capture the seasonal pattern reconstructed by data, it is critical to assess surface processes. These results pinpoint the crucial importance of including the non-linear of the surface water and energy balance to vegetation changes.

2. Take line 49 as an example, 0.5K should be write as 0.5 K, follow SI standard, there is a space between number and unit. May correct throughout the manuscript.

**Response:** Corrected.

3. Line 116, "The new sites", if it is new, the data information should be described, otherwise they are unknown.

**Response:** Corrected by adding the description of new sites "91 digitized data and three original data" on page 6 line 128 in revised version.

4. Line 120, what is cloudiness, how are they measured? Because this is not a common variable, should be described.

**Response:** "Cloudiness" means the "Total Cloud Fraction", it is calculated for the whole atmospheric column, as seen from the surface or the top of atmosphere. Include both large-scale and convective cloud. The standard output name in PMIP is "clt". It's an inverse measure of sunshine (corrected in page 6 line 133 in revised version).

5. Line 129, how do you determine the anomalies for biome scores? What is the purpose of this paragraph L120-L139, to produce reconstruction in Fig5?

**Response:** To determine the anomalies for biome scores, we first use the biomization (Prentice et al., 1996) to get the biome score calculated from pollen taxa percentage for both MH and PI. And then we get the biome score anomaly (MH-PI). The purpose of line 120-139 is to demonstrate the scheme of artificial neural network (ANN) used in our study, and by using this interpolation method, we get the reconstructed spatial pattern of vegetation in Fig. 5 with red rectangle. The schematic diagram of ANN is provided as below (Fig. R3).

[Figure]

**Figure R3.** The schematic diagram of artificial neural network.

6. Line 143 to Line 147, on description of PMIP is a bit strange, what do you mean "in which the PI experiment was denied". "The main variability between MH and PI" should be "The main forcing between MH and PI".

**Response:** From Line 143 to Line 145, we previously wrote as "In its third phase (PMIP3), the models were identical to those used in the CMIP5 experiments, in which the PI experiment was defined", here we mean that the protocol of PI is defined in CMIP5, and the models of PMIP3 followed that from CMIP5. Now according to the suggestion from you and Patrick Bartlein, we deleted the words "in which the PI experiment was defined".

We corrected the "variability" into "forcing" (on page 7 line 161 in revised version).

7. Line 156, "interpolated to a common 2.5 grid", why do you think 2.5 is a common grid, given the pollen data are very local, 2.5 degree grid is too coarse.

**Response:** The "common" here means "uniform", we have corrected (on page 7 line 171 in revised paper). For the model resolution, yes, our study focus on China area, not globe, but for simulation, 2.5 degree is not very coarse even for local study. Moreover, some global models used in our study have lower than 2.5°*2.5° resolution, like IPSL-

CM5A-LR (96*96), it will not make more sense even if we interpolate it into higher resolution.

8. Line 161-162, How do you obtain the sunshine data from observation and model? Should be described more specific.

**Response:** The sunshine data could be calculated as an inverse measure of cloudiness. Cloudiness means the "total cloud fraction", it's an output of model which named as "clt". We first obtained the "clt" from each model, then calculate the sunshine based on "clt". In the new version of manuscript (on page 8 line 177 in revised version), we described more specific "sunshine percentage (an inverse measure of cloud area fraction)".

9. Line 184, "Weighting the attributes is subjective", will it cause uncertainties?

**Response:** As we mentioned in the manuscript, weighting the attributes is subjective because there is no obvious theoretical basis for relative significance. The attributes values listed in Table 4 and Table 5 are according to the previous studies (Skyes et al., 1999; Ni et al., 2000). It may cause some bias, however, it is not likely that different ecologists would assign greatly differing values (Skyes et al., 1999).

10. Line 191, from Zhang et al., 2010, the reference can not be found in reference list.

**Response:** Added in the reference list (on page 40 lines 933-935 in revised version).

11. I am wondering if the warmest month and coldest month changes between MH and PI (and between the models), or always July and January? Give there is a change in seasonality in MH, authors should mention this.

**Response:** For reconstruction, we obtained the biological climate based on pollen records, and the warmest month and coldest month are not always July and January for both MH and PI. And for models, we calculated the mean temperature for every month, and selected the warmest and coldest one to compare with the reconstruction. So the change in seasonality during MH doesn't influence our comparison for MTWA and

MTCO.

12. Line 261, "with a decrease in the northeastern regions", also decrease in east monsoon region at Yangzi river valley.

**Response:** We have added the words "with a decrease in the northeastern regions and east monsoon region at Yangtze River valley" (on page 12 line 275 in revised version).

13. Line 310-312, "this failure to capture ..", see above general comment 2.

**Response:** See the answers to comment 2.

14. Line 320, "triggered" is a weird word.

**Response:** We have deleted this word in revised version.

15. Fig 7 on feedback discussion, how do you determine the feedbacks from the cloud cover or surface cover? In Line 356 the authors mentioned the "surface albedo and cloud change are calculated . . .", I don't understand why the changes in forcing can be regarded as feedback, physically it is a climate response to forcing.

**Response:** The albedo feedback is not identical to the changes in forcing, and it doesn't directly response to forcing. In fact, our philosophy to calculate the albedo feedback is: the forcing change during MH (mainly the seasonal solar radiation change) firstly leads to the seasonal climate change, and accordingly, the vegetation type at that period will be different from PI. Then, the changes in vegetation type have a feedback on climate through the albedo variation. This feedback can be calculated by measuring the changes in radiative fluxes at both Earth's surface and at the top of atmosphere. For instance, if the land surface changes from bare soil to forest, the surface albedo will decrease and the net radiation at land surface increase, in this case, the surface albedo has a positive feedback on the net shortwave flux. The detailed information about how to quantify the feedback are shown in Taylor et al. (2007) and Braconnot and Kageyama (2015).

16. Line 733, "Importance" should be "Important".

**Response:** Corrected.

17. Table 6, should give more information for meteorological data, how long, and give which month is the warmest coldest month, in line 742, should be "warmest month".

**Response:** We have added the data source in the description of Table 6 "data source: China Climate Bureau, China Ground Meteorological Record Monthly Report, 1951-2001" on page 51 lines 1052-1053 in revised paper. For the warmest month and coldest month, it depends on the mean temperature at each site, for instance, the warmest month could be June or July, so we can't give the exact month.

18. Line 744, "stand error" means "standard deviation"?

**Response:** Corrected.

19. Figure 1, should you mark your three original data in this map separately?

**Response:** The three original data are marked in the Fig. 1 in revised version of manuscript (on page 52) as green circles.

20. Figure 4, the huge annual precipitation anomaly in reconstruction, how reliable is it? I highly suspect it. The unit for precipitation is mm, does it mean annual 240 mm equal 20 mm/month? I suggest you use mm/month to avoid confusion.

**Response:** Yes, from our reconstruction, the annual precipitation anomaly (MH-PI) is huge. This increase in mean annual precipitation (MAP) is mainly due to the increased intensity of monsoon in eastern area over China, which brings much higher precipitation during summer, and results in an increased MAP. In Table 6, the regression coefficient (R) between the reconstructed modern MAP by inverse vegetation models (IVM) and observed meteorological values is 0.94, which means the MAP reproduced from the IVM is reliable during present day. But it's also true that there are some bias in MAP reconstruction, in Table S5, we give the median value (MAP) and values indicating the 5% (MAP1)-95% (MAP2) uncertainty bands to show the bias in IVM reconstruction.

21. Figure 6, Line 848 to 850, why do you give the abbreviation, they are not in the figure.

**Response:** Corrected (on page 57 in revised version).

**Response to Prof. Bartlein**

**General comments from the referee:**

This paper presents an ambitious attempt at comparing simulations from the CMIP5/PMIP3 "midHolocene" archive with a new synthesis of fossil-pollen data for China. The pollen data are used in two ways: 1) to develop a set of quantitative reconstructions of several climate variables using an inverse-modeling approach (to compare with the climate-model output), and 2) to develop a map of "megabiomes" for present and 6 ka (for direct comparison with vegetation simulated by BIOME4 using climate-model output). The paper shows that there is a considerable mismatch between the reconstructed and simulated climates and vegetation. The authors attribute this mismatch to experimental-design issues, in which vegetation and land-cover data in the climate models were fixed at present-day values, thereby limiting the ability of the climate models to correctly represent the potential impact of vegetation (and surface water- and energy-balance) feedback in the paleo simulations.

I have two reservations about the results and conclusions:

First, there is insufficient information on the protocols adopted for generating the both present-day and paleo vegetation, as well as the paleo reconstructions. As for vegetation, Fig. 5 shows that there are large mismatches between the observed and simulated modern vegetation. These would naturally arise if the climate-model output were used directly to simulate the vegetation. We know that at their current resolutions, there is still considerable bias in present-day (or PI) climate simulations, and there is no reason to believe that those bias are the same in paleo simulations (or that they somehow go away). My impression of Fig. 5 and S7 is that those biases in simulated climate are indeed large, possibly swamping the real vegetation change, and so we're not really getting much insight into the nature of the mid-Holocene climate simulations, but instead learning about modern-day bias. As for reconstructed climate, despite the author's assertion otherwise, there is also considerable bias in the inverse-model reconstruction approach (Table 6). Only for Pjan does the regression between observed and fitted values not differ from one with a slope of 1.0 and an intercept of 0.0. It is not immediately clear how that bias might affect the reconstructed climate, but it reinforces the necessity of looking at the uncertainties in the reconstructions. Again, we may be learning more about the inverse approach than about model-data mismatches.

Second, the attribution of the mismatches to the experimental design of the CMIP5/PMIP3 simulations, while plausible, is not really supported by any direct hypothesis tests, or by the consideration and dismissal of alternative hypotheses. The correlation between the temperature responses and cloud-cover feedback (Figure 8) implicates at least one: inadequate simulation of atmospheric circulation as it may influence moisture flux or precipitation-generating mechanisms.

I think that if the questions related to the protocol for data-model comparisons are answered, and due consideration given to other possible mechanisms for the mismatch, then the paper will ultimately be publishable.

**Response to the major comments:**

Thanks for the very important comments, in conclusion, two main questions are proposed here:

1. **The insufficient information on protocol for model-data comparison of vegetation**.

**Response:** The referee is right when he pointed out that using climate for PD or 6 ka, there is a large mismatch between vegetation reconstructed by BIOME4 from model simulation and vegetation reconstructed from pollen data. Our aim here is to test whether the sensitivity of PMIP3 mid-Holocene simulations, mainly driven by insolation changes, may explain the vegetation changes observed from data. The trend depicted by all the models is cooler conditions during winter and warmer in summer which is consistent with a linear response to orbital forcing for 6 ka. On the contrary, the dataset shows for the seasonal response a warming during both seasons.

We agree with Prof. Bartlein that the attribution of those non linear responses to vegetation is not really explained in our original manuscript. This is due to the fact that each model has different ways to account for vegetation, therefore our explanation was plausible but too speculative. To be able to convince the referee with quantitative arguments, we have conducted a supplementary experiment to demonstrate that our mechanism was appropriate.

In this revised version, we succeeded to conduct such test in CESM version 1.0.5. The CESM version 1.0.5, developed at the National Center for Atmospheric Research, is a widely used coupled model with dynamic atmosphere (CAM4), land (CLM4), ocean (POP2), and sea-ice (CICE4) components (Gent et al., 2011). Here, we use ~2° resolution for the CAM4, configured by ~1.9° (latitude) × 2.5° (longitude) in the horizontal direction and 26 layers in the vertical direction. The POP2 adopts a finer grid, with a nominal 1° horizontal resolution and 60 layers in the vertical direction. The land and sea-ice components have the same horizontal grids as the atmosphere and ocean components, respectively.

Two experiments were conducted, including a mid-Holocene (MH) experiment (6 ka) with original vegetation setting (prescribed as PI vegetation for MH) and a MH experiment with reconstructed vegetation (6 ka_VEG). In detail, experiment 6 ka used the MH orbital parameters (Eccentricity=0.018682; Obliquity=24.105°; Angular precession=0.87°) and modern vegetation (Salzmann et al., 2008). Compared to

experiment 6 ka, experiment 6 ka_VEG used our reconstructed vegetation in China. Except for the changed vegetation, all other boundary conditions were kept unchanged in these two experiments, including the solar constant (1365 W m−2), modern topography and ice sheet, and pre-industrial greenhouse gases ($CO_2$ = 280 ppmv; $CH_4$ = 760 ppbv; $N_2O$ = 270 ppbv). Experiment 6 ka was initiated from the default pre-industrial simulation and run for 500 model years. Experiment 6 ka_VEG was initiated from model year 301 of experiment 6 ka and run for another 200 model years. We analyzed the computed climatological means of the last 50 model years from each experiment here.

The new-added Fig.9 in manuscript (enclosed below) shows the climate anomalies between two simulations (6 ka_VEG minus 6 ka), for both annual and seasonal scale. For temperature, it's clear that the 6 ka_VEG simulation reproduces the warmer annual (~0.3 K on average) and winter temperature (~0.6 K on average), especially the winter temperature. For precipitation, the reconstructed vegetation leads to higher annual and seasonal precipitation, which can also reconcile the discrepancy of increase amplitude for precipitation during MH between model-data (data reproduced larger amplitude than model, revealed by our study). This new result strongly suggests vegetation changes may explain a part of the mismatch, which is consistent with our proposal in this study. Nevertheless, there are certainly other possibilities and indeed models that better captured the hydrologic cycle and enhance the precipitation/ evaporation pattern could also explain differences between model and data.

Each model has different sensitivity to the boundary change, further work will be carried out in more models to test the influence of vegetation on climate, this is an ongoing work.

This response is on pages 17-18, lines 410-430 in the revised paper.

[Figure]

**Figure 9.** Climate anomalies between the two experiments (6 ka and 6 ka_VEG) conducted in CESM version 1.0.5. The anomalies (6 ka_VEG-6 ka) of temperature and precipitation at both annual and seasonal scale are presented, and all these climate variables are calculated as the last 50-year means from two simulations.

**2. More information about the considerable bias in the inverse-model reconstruction approach.**

**Response:** In the climate reconstruction based on pollen data, the inverse-modeling approach considers the impact of $CO_2$, and provides sound logic to cope with the no-analogue problem. Moreover, the consequence of changing seasonality of climate forcing or climate response are also taken into account in the inverse-model reconstruction. The quantitative reconstructions derived from pollen data of Eurasia, Africa and Europe (Wu et al., 2007) at the LGM and the mid-Holocene, confirm the

ability of the inverse vegetation model (IVM) method to provide spatially coherent patterns of palaeoclimate that are generally in agreement with previous reconstructions from climate proxies.

However, the IVM approach is not a panacea. First, it is highly dependent on the quality of the vegetation model BIOME4, because of the particularity of vegetation types in the monsoon region of China, the BIOME4 needs further improvement of vegetation simulation accuracy in this area. This possible bias in simulating vegetation will lead to uncertainty in reconstruction. Second, the output of the model is not directly compared to the pollen data, the conversion of BIOME4 biomes to pollen biomes by the transfer matrix may add the source of uncertainty in reconstruction. These possible bias in climate reconstruction derived from IVM are described in the new version of manuscript (on page 15 lines 343-353 in revised version).

For the potential uncertainties on data reconstruction, besides the Table 6, we added more information in Table S5 in supplementary information, we gave the climate variables reconstructed from IVM at each site. We also showed the bias on data reconstruction by giving the median value (for instance, column named MTCO) and values indicating the 5% (MTCO1)-95% (MTCO2) uncertainty bands. Moreover, to validate our reconstruction, we compared the results with numerical previous studies concerning MH climate change over China based on multiple proxies (including pollen, lake core, palaeosol, ice core, peat and stalagmite), the relative references and detailed information are listed in Supplementary Information (Table S9 and Table S10). As shown in Figure 7 on page 59 in the revised paper (enclosed below). Compared to PI, most reconstructions reproduced a warmer and wetter annual condition during MH, same as our study. In other words, this discrepancy between model-data for climate change over China during MH is common and robust in reconstructions derived from different proxies. Our study just reinforces the picture given by the discrepancies between PMIP simulation and pollen data derived from a synthesis of the literature. And thus, our reconstruction is reliable even with the potential uncertainties mentioned above.

[Figure]

Figure 7. Comparison between our climate reconstruction and previous reconstruction. (a) Previous temperature results. Diamond is the qualitative reconstruction, red is the temperature increase and green is the temperature decrease; Circle is quantitative reconstruction; (b) Mean annual temperature reconstruction in this study; (c) Previous precipitation results, diamond is the qualitative reconstruction, red is the precipitation increase and green is the precipitation decrease; Circle is quantitative reconstruction; (d) Mean annual precipitation reconstruction in this study.

**Specific comments from the referee:**

Abstract: The abstract fails to disclose conclusions of paper.

**Response:** We modified the abstract as following (on pages 1-2 Line 12-31 in the revised paper):

The mid-Holocene period (MH) has long been an ideal target for the validation of Global Circulation Model (GCM) results against reconstructions gathered in global datasets. These studies aimed to test the GCM sensitivity mainly to the seasonal changes induced by the orbital parameters (precession). Despite widespread agreement between model results and data on the MH climate, some important differences still exist. There is no consensus on the continental size of the MH thermal climate response,

which makes regional quantitative reconstruction critical to obtain a comprehensive understanding of the MH climate patterns. Here, we compare the annual and seasonal outputs from the most recent Paleoclimate Modelling Intercomparison Projects Phase 3 (PMIP3) models with an updated synthesis of climate reconstruction over China, including, for the first time, a seasonal cycle of temperature and precipitation. Our results indicate that the main discrepancies between model-data for MH climates are the annual and winter mean temperature. A warmer-than-present climate condition are derived from pollen data for both annual mean temperature (~0.7 K on average) and winter mean temperature (~1 K on average), while most of the models provide a linear response driven by the seasonal forcing (a decreased annual mean temperature with a warmer summer and colder winter). By conducting simulations in BIOME4 and CESM version 1.0.5, we show that to capture the seasonal pattern reconstructed by data, it is critical to assess surface processes. These results pinpoint the crucial importance of including the non-linear of the surface water and energy balance to vegetation changes.

Line 14: "proxy reconstructions" Aren't the reconstructions used here "real" reconstructions? I understand the notion of paleoclimatic evidence that can be used as a "proxy" for climate or other phenomena (like land cover). But the reconstructions here are actual reconstructions, not a stand-in or substitute for reconstructions.
**Response:** We have deleted the word "proxy" in Line 14 in revised version.

Line 18: "continental size" Are you referring to the area of the temperature anomaly or to terrestrial as opposed to marine responses?
**Response:** We are referring to the area of the temperature anomaly.

Line 20: New definition for PMIP?
**Response:** Corrected as "Paleoclimate Modelling Intercomparison Projects Phase 3" (on page 1 line 21 in revised version).

Line 22: "a seasonal cycle. . ."

**Response:** Corrected as "a seasonal cycle of temperature and precipitation" (on page 1 line 22 in revised version).

Line 25: "access surface processes" I don't know what this means.
**Response:** Sorry for the wrong spelling, it should be "assess".

Line 27: "non-linear process associated with vegetation changes in hydrology and radiative forcing" Does this mean "non-linear responses in hydrology and radiative forcing to vegetation changes"? "Radiative forcing" in the context of the midHolocene experiment is usually reserved for describing the insolation forcing, so an alternative expression might be "non-linear response of the surface water and energy balance to vegetation changes" (which is what I think the paper is arguing for).
**Response:** Thanks for the suggested expression, we have corrected it (on page 2 lines 30-31 in revised version).

Line 34: This definition of the age of the mid-Holocene is inconsistent with what is actually used in the paper (line 101). It might be good to distinguish between the midHolocene time slice, and the "midHolocene" CMIP5/PMIP3 experiment, throughout the paper.
**Response:** According to IntCal13 (Reimer et al., 2013), the mid-Holocene time slice 6000±500 $^{14}$C yr BP is about 6800 Cal BP (the average value), which is not totally consistent with the "mid-Holocene" used in CMIP5/PMIP3 experiment (6000 Cal BP). We agree with the reviewer that this is a problem in model-data comparison for paleoclimate, but for a better comparison with BIOME6000 (which defined as 6000±500 $^{14}$C yr BP), we decided to choose the pollen data at 6000±500 $^{14}$C yr BP in our study.

Thanks to the comment, we will take care of this inconsistency and make better comparison of time slice in the future work.

Line 36: "an increase in insolation in the seasonal cycle" Replace with "an increase in the amplitude of the seasonal cycle of insolation. . ."

**Response:** Corrected. On page 2 line 40 in revised version.

Line 38: "climate response to changes in the seasonal distribution" It's not the response to the seasonal variations of insolation that you're looking at here, but instead the response to changes in the distribution.

**Response:** Corrected. On page 2 line 43 in revised version.

Line 42: "consistency of the dataset incorporating different proxies" I don't know what that means.

**Response:** We have changed it into "much work has been done to reconstruct the paleoclimate change based on different proxies" (on page 2 line 46 in revised version).

Line 45: Again, the data are real, not proxy.

**Response:** Corrected (on page 3 line 49 in revised version).

Line 47: "the source of discrepancies. . ."

**Response:** Corrected it as "the source of discrepancies between model and data …" (on page 3 line 52 in revised version).

Lines 50-51: But see Marsicek et al. (2018, Nature) the "Holocene conundrum" apparently arose from comparing apples and oranges. A different example might be more convincing.

**Response:** Yes, the reconstruction used in Marcott et al. (2013) is mainly from the marine records (~80%) and the cooling trend is largely associated with North Atlantic. And in Marsicek et al. (2018), they show a better consistency of temperature between model-data for Europe and North America continents during Holocene based on 642 sub-fossil pollen data. The different trends of pollen- and marine-based reconstruction indicate the spatial variability of annual temperature change during MH over the globe,

which has already been investigated by Bartlein et al. (2010). Here, we use Liu et al. (2014) to pinpoint the decreased annual temperature in MH simulated by model, compared to PI.

Line 62: The sheer expanse of the country. . . Why should the synthesis of paleoclimatic data or simulations necessarily be restricted to political subdivisions? Extending the area of the comparison deeper into the interior of Eurasia would generate a bit more "leverage" in comparing the data and models, but I understand the logic of restricting the analysis to China.

**Response:** For this study, we only focus on China, but we agree that extending the area into Eurasia or globe is more comprehensive, which is carrying out by other colleges in our group now.

Line 64 (and elsewhere). The article "the" is required before "MH" in this context (i.e. when "MH" is being used as a noun). Elsewhere, as in line 55, where "MH" is used as a modifier of another word ("precipitation" in this context), the article is not used.

**Response:** Corrected.

Line 66: "warmer and wetter than present. . ."

**Response:** Corrected it as "warmer and wetter than present conditions" (on page 3 line 71 in revised version).

Line 73: "colder than the baseline" What baseline? Present-day or preindustrial?

**Response:** The baseline period of 36 models are different. 10 of 36 models refer to present-day, while others refer to preindustrial.

Line 75: "This study" Which study? Reword as "That study. . ." or more explicitly "Jiang et al. (2013) were the first to point out the model-data discrepancy over China during the MH, but the lack of seasonal reconstructions in their study limits comparisons with simulations."?

**Response:** Corrected as "Jiang et al. (2012) were the first to point out the model-data discrepancy over China during the MH, but the lack of seasonal reconstructions in their study limits comparisons with simulations."(on page 4 lines 80-82 in revised version).

Line 83: Bartlein et al. didn't synthesize land-cover changes.
**Response:** Corrected.

Lines 86-91: The terminology here needs to be sorted out. The "process-based biogeographic model" alluded to here is BIOME4, and it is employed in making inferences about past climates using an "inverse modeling through iterative forward modeling" (IMIFM) approach (Guiot et al. 2000; Wu et al., 2007, 2009). (See Izumi and Bartlein, 2016, GRL for further discussion.) So BIOME4 is the vegetation model, while the overall approach (which employs that model) is "IMIFM" (or after that is all explained, simply "the inverse approach").

**Response:** Thanks for the professional and detailed explanation. We have modified this description into "we firstly used the quantitative method of biomization to reconstruct vegetation types during the MH based on a new synthesis of pollen datasets, and then used the Inverse Vegetation Model (Guiot et al., 2000; Wu et al., 2007) to obtain the annual, the mean temperature of the warmest month (MTWA) and the mean temperature of the coldest month (MTCO) climate features over China for the MH." (on page 4 lines 92-94 in revised version).

Line 91: "In the case of models. . ." Which models? Is it the case that you're evaluating the PMIP3 simulations made with state-of-the-art climate models using reconstructions of temperature and precipitation?

**Response:** Yes, we mean the PMIP3 models. And we changed this sentence into "In the case of PMIP3 models, we present a comprehensive evaluation of the PMIP3 simulations made with state-of-art climate models using reconstructions of temperature and precipitation." (on page 4, lines 94-96 in revised paper).

Line 94: "thanks to the seasonal reconstruction" But in all previous applications of the inverse modeling approach using BIOME4 or related models, some sort of reconstruction or estimation of the seasonal variations in climate must have been involved, because BIOME4 requires monthly temperature, precipitation and cloudiness (or sunshine) data as input.

**Response:** Yes, the monthly climate variables are required in BIOME4 or related models, here we only emphasis that our study is to reconstruct the seasonal cycle of MH climate change over China with a synthesis of pollen datasets. We have deleted the words "thanks to the seasonal reconstruction".

Lines 95-96: "the forcing factor we used for MH is essential the seasonal change." I think that what's going on here is that the midHolocene CMIP5/PMIP3 experiment is essentially one that looks at the response of the models to changes in the seasonality of insolation, and that you are attempting to derive reconstructions of both summer and winter temperature and precipitation to compare with the simulations.

**Response:** Corrected, according to your suggestion (on page 5 lines 98-102 in revised version).

Line 101: If you're referring to radiocarbon ages, this should be written as 6000 ± 500 14C yr BP)

**Response:** Corrected (on page 5 line 106 in revised version).

Line 102: Spell out "three".

**Response:** Corrected (on page 5 line 107 in revised version).

Line 105: I don't understand the notion of "distinct" pollen records. Distinct in the sense of "unique" or distinct in the sense of "clearly readable"?

**Response:** We have corrected it into "clearly readable" (on page 5 line 110 in revised version).

Line 107: Criterion 2 seems to be combining two things: sampling resolution and data present within the age range. Please reword.

**Response:** Corrected as "including the pollen taxa during 6000±500 14C yr BP period with a minimum sampling resolution of 1000 years per sample" (on page 5 lines 111-113 in revised version).

Line 108: How far?

**Response:** We abandon the pollen records in our dataset if the published paper mentions the influence of human activity on the pollen. (replaced the "far away" by "abandon the pollen records if the published paper mentions the influence of human activity" on page 5 lines 113-114 in revised version.

Line 109: "or by regression"

**Response:** Corrected (on page 5 line 121 in revised version).

Line 113: Fix the Webb (1985) citation. (Webb, T. III, etc.)

**Response:** Corrected (on page 6 line 124 in revised version).

Lines 110-113: Reorganize sentence to describe the ranking scheme first, and the results second.

**Response:** Corrected (on pages 5-6 lines 114-124 in revised version).

Line 114: Add a citation for the concept of "biomization" (which will be a mystery to modelers).

**Response:** Corrected, we have added Prentice et al., 1996 (on page 6 line 125 in revised version).

Line 116: CQPD. Not in references. Also "sedimentary" what?

**Response:** The CQPD reference was added, and it should be "sediment" (on page 6 line 128 in revised version).

Lines 117-119: Add region names to Fig. 1?

**Response:** Corrected, we added the "Tibetan Plateau" and "Loess Plateau".

Lines 120-135: There seem to be two tasks described in the paragraph: 1) interpolation of modern climate data (from some unspecified source, and by some by some unspecified approach) to the locations of the pollen data, and 2) interpolation of biome scores onto a regular grid using ANN. I suggest breaking this paragraph up, while providing more information on the first task.

**Response:** The modern climate data are based on the datasets (1951-2001) from 657 meteorological observation stations over China, we also added the data source in the manuscript.

Line 129: If the ANN is calibrated using present-day biomes, then I don't see how it can be used to interpolate anomalies. Or was it the case that present-day and paleo biomes were independently interpolated onto the grid, after which anomalies were calculated.

**Response:** The high spatial coverage of present-day pollen records and the application of ANN in interpolating the biome scores makes it possible to reconstruct the past spatial variability with a few pollen sites. In our study, at each pollen site, we firstly used the biomization to get the biome scores for both present-day (PD) and mid-Holocene (MH). Then we calculated the biome score anomalies between two periods (MH-PD). Based on the artificial neural network (ANN), we got the interpolated spatial pattern of biome scores for both PD and anomalies (MH-PD). The spatial pattern of MH biome scores was obtained by overlay the PD pattern with anomalies pattern (MH-PD). Finally, the biome with the highest index is attributed to each grid point, and thus, the spatial pattern of MH vegetation was obtained. The detailed scheme is provided in the enclosed Fig. R2 as below.

[Figure]

**Figure R2.** The schematic diagram of artificial neural network.

Lines 142-143: What are "climate anomalies in the present day"?

**Response:** Corrected it into "climate anomalies in the past periods" (on page 7 line 157 in revised version).

Line 145: Delete "in which the PI experiment was defined."

**Response:** Corrected.

Line 146: Here it would be good to refer explicitly to the "midHolocene" experiment.

**Response:** Corrected.

Line 153: Spell out eight and five.

**Response:** Corrected (on page 7 line 167 in revised version).

Line 156: "in order to calculate" These variables could also be calculated on the models' native grids. The motivation for interpolation onto a common grid is simply to get the data onto a common grid.

**Response:** Corrected as "in order to get the bioclimatic variables (e.g. MAT, MAP,

MTWM, MTCO, July precipitation) onto a common grid for comparison with the reconstruction results" (on pager 7 lines 171-172 in revised version).

Line 160: Either delete the hyphens here, or put them into other instances of "biogeography" or biogeochemistry".
**Response:** Corrected (on page 8 line 174 in revised version).

Line 162: "sunshine percentage (relative to cloud cover)" I don't know what this means.
**Response:** The sunshine percentage is related to cloud cover (not "relative"). We have corrected it as "an inverse measure of cloud area fraction (on page 8 line 176 in revised version).

Line 171: Bigelow: not in references.
**Response:** Corrected (on pages 21 lines 490-496 in revised version).

Line 173: Were the climate variables downscaled in any way (as in the apply-the anomalies approach, Harrison et al., 1998, J. Climate, Harrison et al., 2014, Climate Dynamics). If not, then the climate fields will not contain the spatial variability of modern climate that in topographically complex areas can have a major impact on vegetation. Fig. S7 attests to the existence of bias in the PI simulations. If the simulated climate values are used directly, then a quantitative estimate of the bias (as in Table 6 for the present-day reconstructions) should be provided.
**Response:** We directly used the climate fields from models without downscaling. We agree with the reviewer that there is bias between simulation and observation, especially the topographically complex areas. According to the Taylor diagrams (Figure 1 and Figure 7, the models used in our study are represented as NO. 33, 37, 45, 47, 49, 50, 58, 62, 63, 65, 70, 74 and 75 in blue color) from Jiang et al. (2016, Int. J. Climatol, https://rmets.onlinelibrary.wiley.com/doi/full/10.1002/joc.4406), the GCMs from PMIP3 is reliable to simulate the geographical distribution of surface air temperature and precipitation over China for present day even without downscaling. But there are

considerable bias between GCMs and observation for precipitation. Concerning this issue, we added the following sentences on page 17 lines 406-411 in the revised paper:

"However, the vegetation patterns produced by BIOME4 in Fig. 5 are not used in PMIP3 experiment setup, it's actually determined by the input variables from models. Previous study shows the GCMs from PMIP3 is reliable to simulate the geographical distribution of temperature and precipitation over China for present day without downscaling, but there is considerable bias between simulation and observation for precipitation (Jiang et al., 2016). Therefore, the disagreements of MH vegetation pattern possibly are inherited from the PI."

And thanks for the very important reminder, in the future work, we will try to downscale the climate variables before applying them into regional study.

Line 174: "more than 30 years" How much more? Why not use the same number of years for each model?

**Response:** Corrected (on page 8 line 188 in revised version).

Line 176: Replace "model-data discrepancies" with "differences between simulated (by the climate-model output) and reconstructed (from pollen). . ."

**Response:** Corrected (on page 8 lines 190-191 in revised version).

Line 183: Replace "estimate" with "describe".

**Response:** Corrected (on page 9 line 198 in revised version).

Line 192-194: I'm not sure why you're describing the interpolation of biome data again.

**Response:** We have deleted this sentence.

Line 197: "Inverse Vegetation Model" See earlier comments.

**Response:** Corrected (on page 9 line 210 in revised version).

Line 208: I'm not sure why $CO_2$ concentrations and soil characteristics are being

perturbed (i.e. estimated by the inverse approach). We know CO2, and earlier you argued that soils were assumed not to differ. Also, Table 3 implies that anomalies (or better put, long-term mean differences between present and past) were iteratively generated, which implies that, as is standard procedure, they were applied to present-day climate values and passed to the biome model. If so, what were those present-day values?

**Response:** Sorry for the inaccurate expression here, the $CO_2$ concentration and soil are being perturbed by model. For the soil properties, because of a lack of paleosol data, soil characteristics were assumed to have been the same during the MH. While the atmospheric $CO_2$ concentration for the MH was taken from ice core records (EPICA community members 2004), and set at 270 ppmv; the modern $CO_2$ concentration was set at 340 ppmv, because most of the modern pollen samples were collected during 1970s and 1980s. We have added these descriptions on page 6 lines 136-139 in revised paper:

"The MH soil properties and characteristics used in inverse vegetation model were kept the same with PI conditions, which are derived from the digital world soil map produced by the Food and Agricultural organization (FAO) (FAO, 1995). Atmospheric CO2 concentration for the MH was taken from ice core records (EPICA community members 2004), and was set at 270 ppmv."

Lines 216-218: I don't know if I'm reading Table 6 correctly, but if I am, the slopes and intercepts are anything but close to 1.0 and 0.0. Only for the case of Pjan is the slope within two standard errors of 1.0, and only for MAP and Pjan is the intercept within two standard errors of 0.0. It would be useful to see scatter diagrams of the observed and estimated values for each variable.

**Response:** According to your suggestion, we added the reconstruction result of Pjan in Fig. 4 (on page 56 in revised paper), and we also replace the bars with boxplots in Figure 3 and Figure 4 to show the variability of each model and reconstruction. We agree with the referee that there are some bias in IVM, but based on the comparison with numerical previous studies in China (Figure 7 on page 59 in the revised paper),

this discrepancy between model-data for climate change over China during MH is common and robust in reconstructions derived from different proxies. Our study reinforces the picture given by the discrepancies between PMIP simulation and pollen data derived from a synthesis of the literature.

Line 224: The "collected data" is your data set, right? How was the comparison statistic calculated?

**Response:** Yes, the collected data is the dataset used in this study. The comparison statistic calculated by the match number of pollen sites. We categorized our pollen records into megabiomes, and 145 of 159 (more than 90%) pollen data match well with the BIOME 6000 during MH, while the match number is 149 for PI. We have added the words "145 out of 159 sites" on page 10 line 238 in revised paper.

Lines 226-239: How are the changes or differences in the reconstructions calculated? As differences between the mid-Holocene reconstructions and present-day observations, or present-day inverse-approach estimates? There is considerable bias in the estimates for the present day (Table 6). How would that contribute to the mismatch between simulations and observations? Section 3.1 (throughout): No information on the uncertainties of the reconstructions is given. These are customarily obtained from the variability of the "feasible" climate vectors generated in the optimization step in the inverse approach (e.g. Izumi and Bartlein, 2016, Fig. 3). For that matter, there is no information on the spatial variability of the simulations. Uncertainties for both could be displayed by plotting boxplots in Fig. 3, as opposed to bar graphs.

**Response:** The changes or differences in the paleoclimate reconstructions are calculated as the differences of biome scores between mid-Holocene and present-day times by inverse approach. And the considerable bias listed in Table 6 between observation and estimation for present-day will add the uncertainty in IVM climate reconstruction. We agree with the reviewer and added the boxplots in Figure 3 and Figure 4 on page 55 and page 56 in revised paper. Considering the main purpose here is to show the climate discrepancy between each model and data, we choose to give the

boxplots of 13 models instead of showing MME result with different region. We also give the columns for every variable (derived from IVM at each site) indicating 5-95% uncertainty bands. More detailed information about the uncertainties from reconstruction can be found in Table S5 (on pages 8-13 in SI).

Line 249: "a decreasing trend" Conventionally, trends are described in the sense of a change from one time to another, or the change over a fixed period of time, so here, if the mid-Holocene MTWA values are lower than present, the trend would be positive or increasing over time (i.e. from the mid-Holocene to present). Check the discussion of precipitation trends too. It would be best to simply drop the notion of "trends" and concentrate on the change between midHolocene and PI.

**Response:** Corrected.

Line 265: "more detailed information about the geographic distribution of simulated temperature. . ." Section 3.1 (overall): It would be interesting to see a comparison for Pjan, the single variable with an intercept of 0.0 and a slope of 1.0.

**Response:** Corrected (on page 12 line 280-281). For Pjan, we have added the plot of Pjan in Figure 4. And we also added the description of model-data comparison for Pjan as:

   "For January precipitation, the reconstruction shows an overall increase in most region (~15 mm), except for the northwestern region, while MME indicates a slight decrease (~3 mm on average)." on page 12 lines 278-280 in revised paper.

Line 273: "which would introduce a bias. . ." That's certainly plausible, but right now it's simply a conjecture.

**Response:** Yes, it's true that we haven't quantified the impact of different MH vegetation setting on the role of vegetation-atmosphere interaction in the MH climates among all PMIP3 models. But by giving Fig.5, Fig. 8 and Fig.9, we think that the failure to capture MH vegetation change has influence on model-data discrepancy for climate change.

Line 309: "However, none of the models succeed in capturing these features,. . ." I agree. However, the differences between the simulated and reconstructed biomes for the midHolocene simulations strike me as apparently similar in magnitude to those for the PI, and casual comparison of Figs. 5 and S7 suggests to me that some of the patterns of disagreement in the midHolocene case are inherited from the PI. This makes me wonder again about the protocol followed for generating the midHolocene simulations (see comment on line 173).

**Response:** According to the Taylor diagrams (Figure 1 and Figure 7, the models used in our study are represented as NO. 33, 37, 45, 47, 49, 50, 58, 62, 63, 65, 70, 74 and 75 in blue color) from Jiang et al. (2016, Int. J. Climatol, https://rmets.onlinelibrary.wiley.com/doi/full/10.1002/joc.4406), the GCMs from PMIP3 is reliable to simulate the geographical distribution of surface air temperature and precipitation over China for present day even without downscaling. But there are considerable bias between GCMs and observation for precipitation. This possible bias in simulating vegetation will lead to the mismatch of MH vegetation pattern between model and observation. We agree with the reviewer that the downscaling is very important in applying global model into regional study, which will be taken into account in our future studies. In this study, to be able to convince the referee with quantitative arguments, we decided to conduct a supplementary experiment to demonstrate that our mechanism was appropriate. This new modeling strongly suggests vegetation changes may explain a large part of the mismatch (as shown in Fig. 9), which is very consistent with our proposal in this study. Nevertheless, there are certainly other possibilities and indeed models that better captured the hydrologic cycle and enhance the precipitation/ evaporation pattern could also explain differences between model and data.

Line 310: What are "enhanced vegetation conditions"?

**Response:** The "enhanced vegetation conditions" refers to the transition from grassland to forest in the northeast during MH. We have modified it as "transition from grassland into forest".

Line 311: ". . .a cumulating inconsistency in the model-data comparisons . . . because of the vegetation-climate feedbacks." Except for the two AOV models, vegetation-climate feedback is only present in the real, as opposed to simulated, climate, i.e. in the reconstructions.

**Response:** Yes, all models except for the two AOV models present the real vegetation-climate feedback in PI, but they failed to present the real feedback in MH. The vegetation during MH is prescribed as PI in these 11 models, which means no change of such vegetation-climate feedback for PI and MH among them. This will lead to a cumulating inconsistency.

Line 315-316: "wetter and warmer in MTWA, colder in MTCO" This makes no sense. You might say "higher temperatures in the warmest month of the year," but did you indeed look at precipitation in the warmest month? I think what you want to say is "higher (than present) July precipitation and MTWA, lower than present MTCO" or something like that.

**Response:** Corrected (on page 18 lines 432-433 in revised version).

Line 318: Trend again. Data show higher-than-present MTCO during the mid-Holocene while models simulate lower-than-present MTCO.

**Response:** Corrected.

Line 319: My reading of Fig. 3 shows that CNRM-CM5 and HadGEM2-ES are consistent with all of the other models in simulating lower-than-present MTCO.

**Response:** We have corrected the expression of this sentence. On page 18 line 436 in revised paper.

Line 322: "among models"

**Response:** We have deleted the sentence in the revised paper.

Line 323-324: Replace "shed light" with "raises" (the question). ("Shedding light" implies that the variability referred to would answer the question.)

**Response:** We have deleted the sentence in the revised paper.

Line 326: Replace "amplitude" with "amplitude and pattern". (You emphasize pattern as much as area.) Also, it's not the failure of the models to simulate vegetation change that's important, it's the fact that (apart from HadGEM2-ES and HadGEM2-CC) they can't, because the vegetation is not interactive. However, can't albedo still vary, through variations in soil color and snow cover?

**Response:** The "amplitude" has been replaced by "amplitude and pattern". To the second comment, yes, the albedo could vary through the variation in soil and snow cover, so we checked the monthly surface albedo change (MH-PI) of all models with prescribed vegetation. The Table enclosed below indicates that the surface albedo change caused by snow cover between two periods is very small (no more than 0.005), which could be neglected in this study. About the soil color, to our knowledge, it is prescribed as PI during MH in PMIP3, so this impact on albedo change is also negligible for our study.

Table. The monthly albedo change caused by soil color and snow cover among models.

|  | Model | Jan | Feb | Mar | Apr | May | Jun | Jul | Aug | Sep | Oct | Nov | Dec |
|---|---|---|---|---|---|---|---|---|---|---|---|---|---|
| *PI* | bcc-csm1-1 | 0.149 | 0.156 | 0.156 | 0.154 | 0.142 | 0.127 | 0.136 | 0.143 | 0.147 | 0.153 | 0.152 | 0.145 |
| *MH* | bcc-csm1-1 | 0.149 | 0.156 | 0.157 | 0.155 | 0.142 | 0.126 | 0.134 | 0.140 | 0.142 | 0.148 | 0.147 | 0.143 |
| *MH-PI* | anomaly | -0.001 | 0.000 | 0.001 | 0.001 | 0.000 | -0.001 | -0.001 | -0.003 | -0.004 | -0.004 | -0.006 | -0.002 |
| | | | | | | | | | | | | | |
| *PI* | CCSM4 | 0.170 | 0.172 | 0.173 | 0.174 | 0.164 | 0.145 | 0.143 | 0.153 | 0.159 | 0.168 | 0.168 | 0.164 |
| *MH* | CCSM4 | 0.170 | 0.174 | 0.175 | 0.176 | 0.166 | 0.146 | 0.142 | 0.150 | 0.156 | 0.167 | 0.166 | 0.160 |
| *MH-PI* | anomaly | 0.001 | 0.001 | 0.003 | 0.002 | 0.002 | 0.001 | -0.001 | -0.003 | -0.002 | -0.001 | -0.003 | -0.003 |
| | | | | | | | | | | | | | |
| *PI* | CNRM-CM5 | 0.151 | 0.164 | 0.164 | 0.161 | 0.146 | 0.128 | 0.130 | 0.140 | 0.143 | 0.152 | 0.152 | 0.141 |
| *MH* | CNRM-CM5 | 0.149 | 0.161 | 0.161 | 0.159 | 0.144 | 0.124 | 0.130 | 0.137 | 0.138 | 0.145 | 0.147 | 0.140 |
| *MH-PI* | anomaly | -0.002 | -0.003 | -0.003 | -0.002 | -0.002 | -0.003 | -0.001 | -0.003 | -0.005 | -0.007 | -0.004 | -0.001 |
| | | | | | | | | | | | | | |
| *PI* | CSIRO-Mk3-6-0 | 0.169 | 0.181 | 0.178 | 0.171 | 0.156 | 0.140 | 0.147 | 0.161 | 0.167 | 0.171 | 0.169 | 0.162 |
| *MH* | CSIRO-Mk3-6-0 | 0.170 | 0.180 | 0.179 | 0.172 | 0.157 | 0.139 | 0.145 | 0.161 | 0.164 | 0.168 | 0.164 | 0.162 |
| *MH-PI* | anomaly | 0.001 | 0.000 | 0.001 | 0.001 | 0.000 | -0.001 | -0.001 | 0.000 | -0.003 | -0.004 | -0.005 | -0.001 |
| | | | | | | | | | | | | | |
| *PI* | FGOALS-g2 | 0.170 | 0.172 | 0.170 | 0.168 | 0.154 | 0.137 | 0.141 | 0.155 | 0.156 | 0.164 | 0.168 | 0.161 |
| *MH* | FGOALS-g2 | 0.172 | 0.175 | 0.173 | 0.173 | 0.159 | 0.141 | 0.142 | 0.156 | 0.158 | 0.167 | 0.168 | 0.163 |
| *MH-PI* | anomaly | 0.002 | 0.003 | 0.004 | 0.005 | 0.005 | 0.004 | 0.001 | 0.002 | 0.002 | 0.003 | 0.000 | 0.002 |
| | | | | | | | | | | | | | |
| *PI* | FGOALS-s2 | 0.165 | 0.173 | 0.171 | 0.170 | 0.161 | 0.148 | 0.153 | 0.164 | 0.165 | 0.168 | 0.166 | 0.161 |
| *MH* | FGOALS-s2 | 0.164 | 0.174 | 0.173 | 0.172 | 0.162 | 0.142 | 0.150 | 0.159 | 0.160 | 0.165 | 0.160 | 0.156 |
| *MH-PI* | anomaly | 0.000 | 0.001 | 0.002 | 0.002 | 0.001 | -0.006 | -0.003 | -0.005 | -0.005 | -0.004 | -0.005 | -0.005 |
| | | | | | | | | | | | | | |
| *PI* | GISS-E2-R | 0.144 | 0.153 | 0.154 | 0.150 | 0.131 | 0.114 | 0.119 | 0.126 | 0.132 | 0.138 | 0.138 | 0.134 |
| *MH* | GISS-E2-R | 0.144 | 0.153 | 0.153 | 0.149 | 0.131 | 0.114 | 0.111 | 0.122 | 0.124 | 0.129 | 0.132 | 0.130 |
| *MH-PI* | anomaly | 0.000 | -0.001 | -0.001 | 0.000 | 0.000 | 0.000 | -0.007 | -0.004 | -0.008 | -0.009 | -0.006 | -0.003 |
| | | | | | | | | | | | | | |
| *PI* | IPSL-CM5A-LR | 0.149 | 0.158 | 0.161 | 0.159 | 0.148 | 0.133 | 0.140 | 0.149 | 0.152 | 0.155 | 0.148 | 0.142 |
| *MH* | IPSL-CM5A-LR | 0.149 | 0.158 | 0.162 | 0.160 | 0.149 | 0.133 | 0.140 | 0.147 | 0.150 | 0.152 | 0.144 | 0.139 |
| *MH-PI* | anomaly | 0.000 | 0.000 | 0.001 | 0.001 | 0.001 | 0.000 | 0.000 | -0.001 | -0.002 | -0.003 | -0.004 | -0.002 |
| | | | | | | | | | | | | | |
| *PI* | MIROC-ESM | 0.162 | 0.171 | 0.173 | 0.167 | 0.146 | 0.123 | 0.129 | 0.136 | 0.143 | 0.153 | 0.155 | 0.151 |
| *MH* | MIROC-ESM | 0.164 | 0.174 | 0.176 | 0.172 | 0.150 | 0.124 | 0.131 | 0.137 | 0.144 | 0.154 | 0.153 | 0.152 |
| *MH-PI* | anomaly | 0.002 | 0.002 | 0.003 | 0.005 | 0.004 | 0.001 | 0.002 | 0.000 | 0.000 | 0.001 | -0.002 | 0.001 |
| | | | | | | | | | | | | | |
| *PI* | MPI-ESM-P | 0.170 | 0.164 | 0.160 | 0.155 | 0.153 | 0.155 | 0.151 | 0.145 | 0.145 | 0.151 | 0.161 | 0.173 |
| *MH* | MPI-ESM-P | 0.169 | 0.163 | 0.160 | 0.157 | 0.154 | 0.154 | 0.148 | 0.141 | 0.141 | 0.147 | 0.160 | 0.171 |
| *MH-PI* | anomaly | -0.001 | -0.001 | 0.000 | 0.001 | 0.001 | -0.001 | -0.003 | -0.004 | -0.004 | -0.004 | -0.001 | -0.002 |
| | | | | | | | | | | | | | |
| *PI* | MRI-CGCM3 | 0.183 | 0.194 | 0.194 | 0.190 | 0.173 | 0.156 | 0.158 | 0.172 | 0.176 | 0.187 | 0.189 | 0.180 |
| *MH* | MRI-CGCM3 | 0.183 | 0.195 | 0.196 | 0.191 | 0.174 | 0.154 | 0.156 | 0.168 | 0.172 | 0.183 | 0.183 | 0.175 |
| *MH-PI* | anomaly | 0.000 | 0.001 | 0.001 | 0.002 | 0.001 | -0.001 | -0.002 | -0.003 | -0.005 | -0.004 | -0.005 | -0.005 |

Line 337: "Reconstruction showed. . ." I thought you were talking about the two AOV models. This sentence implies that you estimated the overall albedo change from the vegetation reconstruction, and compared with the two models with interactive vegetation. Is that right? If not, please explain a bit more.

**Response:** Yes, it's right. We modified the sentence into "The overall albedo change from the vegetation reconstruction during the MH shows a ~1.8% decrease when snow-free, with a much larger impact (~4.2% decrease) when snow-covered" on page 16 lines 373-374 in revised paper.

Line 348-349: "should act" or "most likely would act" (We don't really know if it would.)

**Response:** Corrected.

Line 351-353: It may well be the case that cloud radiative feedback (or rather, inadequate simulation of that) could play a role in the data-model mismatch, but if so, that points to a completely different kind of model inadequacy, involving atmospheric circulation, moisture flux, and cloud-producing or cloud-suppressing mechanisms. Those mechanisms have been implicated in explaining the mismatch between simulations and reconstructions in the Eurasian midcontinent (Bartlein et al., 2017, GRL).

**Response:** It's a very important comment. We agree with your proposal that cloud radiative feedback may play a role in model-data mismatch, which indicates another kind of inadequacy. For this study, we simply focuses on the surface land change, we are not able to quantify the possible influence of mechanisms related to cloud on this model-data discrepancy for now, but we can do more in the future.

Line 354: Taylor (and fix reference too).

**Response:** Corrected.

Technical comments: I concur with the Editor and other referees that some work needs

to be Corrected on the references and data-availability aspects of the paper. References: Format varies from reference to reference. Tables 4, 5 & 6: Replace commas with periods (decimal points). Fig. 7: Define dotted horizontal and vertical lines. Maps (throughout): Why does the "nine-dash line" inset vary in size and shape? I realize that the inset has to be there for geopolitical reasons, but why does it change from map to map? Fig. S7: What is the white horizontal line? SI, p. 1: Dallmeyer et al. (2017), not in references. SI, p. 3: Material at the bottom of the table is hard to read. Please reformat into a table-like arrangement. SI, p. 5: Add citations to original data sources.

**Response:** The references format is uniform in the revised paper. We added the sentence on page 60 line 1195 to define the dotted horizontal and vertical lines in Fig. 8. For the nine-dash line inset, it's only a schematic diagram, the size and shape could be different from figures as long as we show the nine-dash line in the right location. The Dallmeyer et al. (2017) is included in the references, and we reformatted the material at the bottom of the table in the revised paper. For the citations in the SI, p. 5, we give the information in Table. 1 in the manuscript, and the main aim of able S4 is to show the reconstruction results from IVM at each site.

**The marked-up manuscript version**

[revised manuscript text omitted]

[Figure]

GISS-E2-R    HadGEM2-ES    CNRM-CM5    BCC-CSM1-1
FGOALS-s2    CSIRO-Mk3-6-0    MRI-CGCM3    IPSL-CM5A-LR
MPI-ESM-P    CCSM4    MIROC-ESM    FGOALS-g2
HadGEM2-CC

**Figure 8.** Scatter plots showing temperature, cloud cover feedback and surface albedo feedback changes during the MH. The values shown are the simulated 30-year mean anomaly (MH-PI) for the 13 models. **a**, annual mean temperature relative to the annual mean cloud cover feedback and d, annual surface albedo feedback. b, Summer (JJA) mean temperature relative to the summer mean cloud cover feedback and **e**, Summer surface albedo feedback.    **c**, Winter (DJF) mean temperature relative to the summer mean cloud cover feedback and **f**, Winter surface albedo feedback. The horizontal and vertical lines in plots represent the value of 0.

[Figure]

**Figure 9.** Climate anomalies between the two experiments (6 ka and 6 ka_VEG) conducted in CESM version 1.0.5. The anomalies (6 ka_VEG-6 ka) of temperature and precipitation at both annual and seasonal scale are presented, and all these climate variables are calculated as the last 50-year means from two simulations.

---

## Author Response (AR2)

**Response to the editor and reviewers**

We greatly appreciate the constructive comments and suggestions on the previous version of the manuscript from the editor and reviewers. The following is the point-by-point reply, with reference to the order of the comments made by the editor and reviewers.

According to the comments, the main changes we have made in this version are listed as below:

1. Both the referee and the editor suggested to include more information in the manuscript from our response to the review. We have added following the suggestions.

2. The data availability. For the data from China Quaternary Pollen Database, the original data is not public yet, and we can't share these data due to the rule of data usage. According to the suggestion from the editor, we added the statement and provided a contact in the data availability section (on page 22, lines 518-523): For the data from CQPD, the basic information (location, data supporter, age control and biome type of each site) can be found in CQPD (2000), while the original data are not publicly available yet. To whom request for the data, you can contact with Yunli Luo (lyl@ibcas.ac.cn, Institute of Botany, Chinese Academy of Sciences, Beijing, 100093, China), a core member and academic secretary of the CQPD.

The modifications for specific comments are indicated in each response and marked as underlined words in the marked-up manuscript below.

**Response to the editor**

**Comment:** Both there reviewers and myself are concerned that the responses to the review were NOT implemented in the manuscript. Referee#4 listed the most important comments/responses that do not appear in the manuscript. Moreover, according to the referee, your responses and the manuscript are giving contradictory information in a few cases. Therefore, I urge you to take into account in the manuscript all the responses that you gave to the referees.

**Response:** Thanks for the important comment. We have carefully checked the author's response.pdf, and the important clarifications and responses that we gave to the referees are now included in our manuscript or Supplementary Information. The main changes we have made in this version are listed as below:

1) Added the description of spatial variability for MH climate anomaly (on page 3, lines 64-71): For instance, the reconstruction used in Marcott et al. (2013) is mainly from the marine records (~80%), the cooling trend is largely associated with North Atlantic. And in Marsicek et al. (2018), they show a better consistency of temperature between model-data for Europe and North America continents during Holocene based on 642 sub-fossil pollen data. The different trends of pollen- and marine-based reconstruction indicate the spatial variability of annual temperature change during MH over the globe, which has already been investigated by Bartlein et al. (2010).

2) Added the definition of MH used in our study (on page 5, lines 114-119): Notably, according to IntCal13 (Reimer et al., 2013), the MH time slice 6000±500 14C yr BP is about 6800 Cal BP (the average value), which is not totally consistent with the "mid-Holocene" used in CMIP5/PMIP3 experiment (6000 Cal BP). But for a better comparison with BIOME6000 (in which the MH is defined as 6000±500 14C yr BP), we decided to choose the pollen data at 6000±500 14C yr BP in our study.

3) Added the description of ANN method (on page 7, lines 160-161 and lines 168-169). We also added the schematic diagram of ANN in Supplementary Information as Figure S1 on page 21.

4) A new-added section "Sensitivity test for vegetation feedback" (on pages 11-12, lines 248-269):

"To quantify the vegetation feedback on climate change during mid-Holocene over China, we did the sensitivity test in CESM version 1.0.5. This version, developed at the National Center for Atmospheric Research, is a widely used coupled model with dynamic atmosphere (CAM4), land (CLM4), ocean (POP2), and sea-ice (CICE4) components (Gent et al., 2011). Here, we use ~2° resolution for the CAM4, configured by ~1.9° (latitude) × 2.5° (longitude) in the horizontal direction and 26

layers in the vertical direction. The POP2 adopts a finer grid, with a nominal 1° horizontal resolutions and 60 layers in the vertical direction. The land and sea-ice components have the same horizontal grids as the atmosphere and ocean components, respectively.

Two experiments were conducted, including a mid-Holocene (MH) experiment (6 ka) with original vegetation setting (prescribed as PI vegetation for MH) and a MH experiment with reconstructed vegetation (6 ka_VEG). In detail, experiment 6 ka used the MH orbital parameters (Eccentricity=0.018682; Obliquity=24.105°; Angular precession=0.87°) and modern vegetation (Salzmann et al., 2008). Compared to experiment 6 ka, experiment 6 ka_VEG used our reconstructed vegetation in China. Except for the changed vegetation, all other boundary conditions were kept unchanged in these two experiments, including the solar constant (1365 W m−2), modern topography and ice sheet, and pre-industrial greenhouse gases ($CO_2$ = 280 ppmv; $CH_4$ = 760 ppbv; $N_2O$ = 270 ppbv). Experiment 6 ka was initiated from the default pre-industrial simulation and run for 500 model years. Experiment 6 ka_VEG was initiated from model year 301 of experiment 6 ka and run for another 200 model years. We analyzed the computed climatological means of the last 50 model years from each experiment here."

5) Added the clarification of model resolution issue (on pages 17-18, lines 395-416): "The discrepancies between model-data for MH climate change could also be resulted from the uncertainties in simulation. Firstly, the coarse spatial resolution of models. Previous study shows the GCMs from PMIP3 is reliable to simulate the geographical distribution of temperature and precipitation over China for present day without downscaling, but there is considerable bias between simulation and observation for precipitation (Jiang et al., 2016). In particular, the climate fields, directly used from the model output without downscaling, will not contain the spatial variability of modern climate that in topographically complex areas. And thus, it's necessary to check in which degree the model-data mismatch is related to rough topography used in the climate models. In PMIP3, MRI-CGCM3 has the highest resolution (Atmosphere: 320*160*L48; Ocean: 364*368*L51), while

IPSL-CM5A-LR has the lowest one (Atmosphere: 96*96*L39; Ocean: 182*149*L31). In Fig. 8 (enclosed below), we give the actual modern topography and the interpolated topography used in MRI-CGCM3 and IPSL-CM5A-LR. For MRI-CGCM3, the topography is very close to the observation, so for this model, the model-data discrepancy during MH over China is not related to the resolution. However, for the model with coarse resolution (IPSL-CM5A-LR), it's true that the coarse version of model will lead to bias in topography when the regional diversity is discussed. The variations in topography could influence the vegetation and hence the simulated climate. To quantify this impact, we compare the topography and PI climate results of IPSL-CM5A-LR and IPSL-CM5A-MR. Fig. 9 shows that the difference in topography caused by model resolution do has an impact on small scales (south region of the Tibetan Plateau), but not on the overall pattern. For a better comparison, the climate variables should be downscaled in the future work."

6) Added the Fig. 8 and Fig. 9 in the manuscript to illustrate the resolution issue (on pages 63 and 64).

7) Added the description of model bias in PI simulation (on page 18, lines 420-421), as well as the Figure S10 in Supplementary Information.

8) Added the comparison of AOVGCM and AOGCM (on page 19, lines 446-449): In conclusion, there is an obvious advantage of using AOVGCM instead of AOGCM when we discussing about the MH climate, but the premise is that the AOVGCM can simulate accurate vegetation distribution.

**Comment**: Moreover, as I already mentioned to you, according to CP (Copernicus) data policy, 'data that correspond to journal articles [are requested to be deposited] in reliable (public) data repositories, assigning digital object identifiers, and properly citing data sets as individual contributions' and the '[a]uthors are required to provide a statement on how their underlying research data can be accessed'.

The editor in chief of climate of the past and COPERNICUS journal agreed that it is not compulsory to deposit data in a public database. However, there must be a clear

statement on how the underlying data can be accessed.

First, the reference for CQPD (2000) is missing in the main paper. I assumed (but might be wrong) that it refers to Members of Quaternary Pollen Data Base in Institute of Geochemistry, Chinese Academy of Science: The evolution of the environment since 10000 years in the south of Liaoning Province, Science in China, 6, 603- 614, 1977 (in Chinese). Although this one is 1977. Could you please clarify this reference.

**Response**: The reference of CQPD (2000) is listed on page 36, lines 840-842.

**Comment**: Second, you must clearly state how the reader can obtain the same information that you got. Could you explain (in the paper, probably in the data availability section) the status of the CQPD (will the data ever be public, can outsiders find out who posted data and to whom the data should be asked?). Moreover, you must let readers know how they could access information from it. If the data are not publicly available (yet) you must identify the individual (name and contact details) to whom request for data should be sent. This information must be included in the 'data availability' section.

These requests seems to me to be fair. It will give the paper a minimum standard to be accepted for publication.

**Response:** we added the statement and provided a contact in the data availability (on page 22, lines 518-523): "For the data from CQPD, the basic information (location, data supporter, age control and biome type of each site) can be found in CQPD (2000), while the original data are not publicly available yet. To whom request for the data, you can contact with Yunli Luo (lyl@ibcas.ac.cn, Institute of Botany, Chinese Academy of Sciences, Beijing, 100093, China), a core member and academic secretary of the CQPD."

**Response to the reviewer**

I think the authors should be commended for their response to (four) wide-ranging and thorough reviews. In particular, the addition of the sensitivity test comparing simulations with present-day as opposed to reconstructed mid-Holocene vegetation is a solid addition, and greatly strengthens the assertion that data-model mismatches are attributable to experimental design issues. Unfortunately, the clarifications and additions are found in the authors' response as opposed to the revised manuscript. (A quick example: I wondered what "continental size" meant (line 18 in the original manuscript). This was clarified in the authors' response (p. 37), but not in the revised paper.) Some of the clarifications and explanations are quite substantial, and aren't even hinted at in the revised manuscript, or appear out of place. For example, the sensitivity test described in the author's response (p. 9) isn't really introduced until the end of the discussion section. The material on p. 9 of the authors' response should appear in the paper itself, partially in section 2, as well as in the discussion.

**Response**: Thanks for the suggestion, the clarification of "continental size" is added in the manuscript (line 18, page 1), and the introduction of the sensitivity test is now described in section 2 (lines 248-269, page 10-11), as well as in the discussion.

I'm still concerned about resolution issues, and the direct use of model output at GCM resolutions (as opposed to the standard "apply-the-anomalies" approach, but perhaps my concern would be alleviated if the fragmented discussion of that issue in the authors' response appeared in one place in the revised manuscript.

**Response:** We totally agree with the referee that the downscaling of GCMs is important for model-data comparison, which should be carried out in our future work. In this revised version, concerning the model resolution, we combined the fragmented discussion from the author's response into a paragraph in the discussion section in the manuscript (on pages 17-18, lines 395-416). The content is below:

"The discrepancies between model-data for MH climate change could also be resulted from the uncertainties in simulation. Firstly, the coarse spatial resolution of

models. Previous study shows the GCMs from PMIP3 is reliable to simulate the geographical distribution of temperature and precipitation over China for present day without downscaling, but there is considerable bias between simulation and observation for precipitation (Jiang et al., 2016). In particular, the climate fields, directly used from the model output without downscaling, will not contain the spatial variability of modern climate that in topographically complex areas. And thus, it's necessary to check in which degree the model-data mismatch is related to rough topography used in the climate models. In PMIP3, MRI-CGCM3 has the highest resolution (Atmosphere: 320*160*L48; Ocean: 364*368*L51), while IPSL-CM5A-LR has the lowest one (Atmosphere: 96*96*L39; Ocean: 182*149*L31). In Fig. 8, we give the actual modern topography and the interpolated topography used in MRI-CGCM3 and IPSL-CM5A-LR. For MRI-CGCM3, the topography is very close to the observation, so for this model, the model-data discrepancy during MH over China is not related to the resolution. However, for the model with coarse resolution (IPSL-CM5A-LR), it's true that the coarse version of model will lead to bias in topography when the regional diversity is discussed. The variations in topography could influence the vegetation and hence the simulated climate. To quantify this impact, we compare the topography and PI climate results of IPSL-CM5A-LR and IPSL-CM5A-MR. Fig. 9 shows that the difference in topography caused by model resolution do has an impact on small scales (e.g. south region of the Tibetan Plateau), but not on the overall pattern. For a better comparison, the climate variables should be downscaled in the future work."

Although the authors' response will be available to readers, its structure, consisting of responses to the individual reviewers in turn, make it less useful, and I worry that readers likely ignore the in any case, assuming that if the paper has been published, all necessary changes have been made.

**Response:** As the referee said, thanks to the 4 reviewers and the editor, our paper received a wide-ranging and thorough review. For us, response to the individual reviewers in turn is the most direct and clear way. But concerning the numerical pages of our response (123 pages), we agree that readers are likely to ignore all necessary

changes that have been made in the revised version. According to the suggestion from both referee and the editor, the important and substantial changes will be implemented in the manuscript this time, as opposed to only in the response.pdf.

**Specific comments:**

Fig. R1 (Model bias discussion in the authors' response): I think that piControl values should be plotted on the x-axis, to conform to the convention of the predictor on the x-axis, response on the y-axis. Also, with the usual FGOAL-s2 outlier removed, then I do see a relationship for JJA. Fig. R1 does not appear anywhere in the revised version of the manuscript. Is there a further supplemental file that contains the "R" figures?

**Response:** we modified the figure according to the suggestion, and this figure is added in the supplementary as Figure S10 (on page 30). We also added the clarification of this issue in the manuscript, on page 18, lines 419-424.

Fig. R2 (Model resolution): Again, this material belongs in the paper, as opposed to the authors' response .pdf, where it is likely to be overlooked.

**Response:** The Fig. R2 was included in the manuscript, and renamed as Figure 8, on page 63.

Authors' response p 13: "The difference in topography caused by model resolution has influence on some small regional climate, but no significant change for general pattern." I think the idea is that resolution has an impact on small (or regional) scales, but not on the overall pattern. But those small or regional-scale variations in climate can have a large impact on vegetation and hence reconstructed climate.

**Response:** According to the referee's suggestion, we have modified the description as "Figure 9 shows that the difference in topography caused by model resolution do has an impact on small scales (south region of the Tibetan Plateau), but not on the overall

pattern. For a better comparison, the climate variables should be downscaled in the future work" on page 17, lines 413-416.

AOV vs. AO discussion (authors' response, p. 14): An abbreviated version of this discussion should appear in the paper.

**Response**: The comparison between two AOVGCM has already been described in the last revised version (on pages 17-18, lines 416-433). In this version, we added the AOV vs. AO discussion on page 19, lines 446-449 as "In conclusion, there is an obvious advantage of using AOVGCM instead of AOGCM when we discussing about the MH climate, but the premise is that the AOVGCM can simulate accurate vegetation distribution."

Reviewer 3's point 2: (More material describing the sensitivity test.) It would be worth pointing out (in the paper) that if vegetation were prescribed using reconstructed biomes, then that would reduce the power of the biome-based climate reconstructions (owing to the potential for circularity: "Prescribe the vegetation to get the vegetation.). I don't think that's really the case, but the possibility of circularity will worry some readers and should be acknowledged.

**Response**: For the description of sensitivity test, we have added a new section (sensitivity test for vegetation feedback) in part two, in which the model and experiment design for vegetation feedback are introduced (on pages 10-11, lines 248-269). For the potential circularity, we described it as "However, it should also be noted that prescribing the vegetation with reconstructed biomes would reduce the power of the biome-based climate reconstruction, owing to the potential circularity (prescribe the vegetation to get the vegetation)" on page 21, lines 505-508.

Authors' response p. 24 (and p. 26): "[Cloudiness is] an inverse measure of sunshine…". No it isn't (New et al., 1999, J. Climate 12:829-856). There is a strong latitudinal gradient in the relationship between the two.

**Response**: Thanks for the correction, we will pay attention to the difference between the two notions.

Line 21: "Project" (singular, not plural) (https://pmip3.lsce.ipsl.fr)

**Response**: Corrected on page 1, line 21.

Line 38: definition of "mid-Holocene" Again, the definition is clarified in the authors' response, but not the paper.

**Response:** The definition of mid-Holocene is now clarified in the manuscript on page 5, lines 114-118.

Line 52: (Source of discrepancies paragraph). The authors' response handles my concern well, but again, this should appear in the paper. (I'm not just trying to boost my citation count here. Marsicek et al. clearly dismiss the "Holocene conundrum", and so using the conundrum as a motivation for the current paper seems inadvisable—readers that know about Marsicek et al., will wonder why the conundrum is being used to motivate the current paper, and those data don't will be left with the impression that there's a sever mismatch.)

**Response**: Thanks for the important suggestion. We have added this issue in the source of discrepancies paragraph on page 3, lines 64-71. The description was wrote as below: "For instance, the reconstruction used in Marcott et al. (2013) is mainly from the marine records (~80%), the cooling trend is largely associated with North Atlantic. And in Marsicek et al. (2018), they show a better consistency of temperature between model-data for Europe and North America continents during Holocene based on 642 sub-fossil pollen data. The different trends of pollen- and marine-based reconstruction indicate the spatial variability of annual temperature change during MH over the globe, which has already been investigated by Bartlein et al. (2010)."

Line 141: (ANN description). Again, the authors' reply clarifies the approach, but that clarification does not appear in the paper, nor does Fig. "R2". (Moreover, there are two figures labelled "R2" in the response (p. 25 and p. 45)—the second (p. 45) probably should be R3.

**Response**: We added the ANN description:"In our study, at each pollen site, we firstly used the biomization to get the biome scores for both present-day and MH" on page 7, lines 160-161. And we also give the schematic diagram of ANN in Supplementary Information as Figure S1.

Table S5: What does "(varying from files)" mean?

**Response**: Although models use the prescribed vegetation for MH, the vegetation pattern is varying from the parameters settings, for instance, the leaf area index. Here, "varying from files" means "varying from files with different parameter settings". To make it more clear, we have modified it into "varying from parameters".

[revised manuscript text omitted]

---

## Author Response (AR3)

**Response to the editor**

We greatly appreciate the constructive comments and suggestions on the previous version of the manuscript from the editor. The following is the point-by-point reply, with reference to the order of the comments made by the editor and reviewers.

**Comment:** Thank you for taking the time to answer the reviewers comments and to include them into your manuscript. Once more I want to congratulate you for the large amount of work performed in 'rescuing' data.

There are still some items that must be improved before publication. I bet that you will honestly make these changes.

1. I understood that the initial manuscript was thoroughly edited for English. Many pieces of text have been included in the manuscript. Therefore a full check for English is required. As an example, the English of section 4.2 as well as of the data availability section is poor. Other sections may suffer from the same problem.

In particular, I don't understand that the simulations are reliable for temperature and precipitation but the bias is considerable for precipitation.

**Response:** According to the suggestion, we have fully checked the English of our manuscript, and the corrections can be found in the revised manuscript as below. For instance, we modified the sentence 'the simulations are reliable for temperature and precipitation but the bias is considerable for precipitation' into 'previous studies show that the GCMs from PMIP3 are reliable to simulate the geographical distribution of temperature and precipitation over China for present day. However, compared with observation, most models have topography-related cold biases (Jiang et al., 2016)' on page 17, lines 397-400.

2. Tables S6 and S7; Figures 3 and 4. Are the values (in the tables and in the box plot) an average over China or the mean of the values at the different points where there are data?

**Response:** The values in Tables S6 and S7; Figures 3 and 4 are the mean of the values at different points where there are data. We have added this remark in the description of these data in tables and figures on page 14 line 324.

3. Authors' response p13 (from the reviewer comments). You indicated that you modified the description. Indeed you included most of the reviewer's comment, except 'But those small or regional-scale variations in climate can have a large impact on vegetation and hence reconstructed climate'. My understanding is that this sentence is as important (if not more) than the previous one. Could you please include it in the manuscript.

**Response:** According to the suggestion, we have added the sentence in the manuscript on page 18, lines 415-416.

4. A big effort has been made for the references. However, there are still some issues.

a. The following references are both cited as (Li et al., 2011).

Li, X., Zhao, K., Dodson, J., and Zhou, X.: Moisture dynamics in central Asia for the last 15 kyr: new evidence from Yili Valley, Xinjiang, NW China, Quaternary Science Reviews, 30, 23-34, 2011.

Li, C., Wu, Y., and Hou, X.: Holocene vegetation and climate in Northeast China revealed from Jingbo Lake sediment, Quaternary International, 229, 67-73, 2011.

According to Copernicus Publications house standards they should be cited in the text as (X. Li et al., 2011) and (C. Li et al., 2011). Please check your manuscript.

**Response:** We have modified the citation in the manuscript on page 49, page 50 and page 51, Table 1.

b. I identified the same issue for (Liu et al., 2007), (Zhang et al., 2007), (Wang et al., 2010).

**Response:** We have modified the citation in the manuscript on page 4 line 93, pages 49-51, Table 1.

c.   More classically, they are also two different references corresponding to (Li et al, 2003). They should be (Li et al, 2003a) and (Li et al, 2003b).

**Response:** According to the suggestion, we have modified the citation in the manuscript on page 50.

d.   There is no reference corresponding to (Yang et al., 2001).

**Response:** We have added the citation on page 46 lines 1067-1069:

Yang, Y., Huang, C., Wang, S., and Kong, Z: Study on the mire development and palaeogeographical environment change since the early period of the Holocene in the east part of the Xiliaohe Plain, Scientia Geographica Sinica, 21, 242-249, 2001 (in Chinese with English abstract).

e.   Most probably the reference 'Members of the China Quaternary Pollen Data Base' for (CQPD, 2000) will not be accepted. You should check with Copernicus on how to proceed. Indeed, if you indicate CQPD in the text, as a reference, it should be find AS IS in the list of references. At the moment it is not the case.

**Response:** According to previous experience (see Li et al., Large-scale vegetation history in China and its response to climate change since the Last Glacial Maximum, Quaternary International, 2019), it's ok to cite CQPD like this. But we agree with the editor that this citation could cause some confusion to the readers, we will check it with Copernicus if it isn't accepted.

f.   The reference Members, M.P. should be MARGO Project Members.

**Response:** Corrected on page 36, line 852.

5.   The data availability section. I appreciate your effort here. Now the colleagues interested in CQPD data know how to contact the authors to possibly use the data. There are still 91+3 dataset for which nothing is mentioned. My understanding is that you digitized most of them (91). It would be extremely useful if these data were available as well. Can they be requested to one of the co-authors of this paper? Could

that be mentioned in the data availability section? For the first three datasets in Table 1, you indicate that it is 'original data'. Then their digital version should be available somewhere. Could you mention where/how to get the values?

**Response**: For the 91 digitized data, they can be requested to the co-author of our paper Qin Li, we also provided the email address of her in the data section (on page 22 lines 516). About the 3 original data, they are given by the original author of these papers, which can also be archived by Qin Li.

6. Table 2. The eccentricity value for the PI should be 0.016724. Angular precession is not the correct astronomical term (neither in the table nor in the text). I would suggest 'longitude of the perihelion'. Please check all the values in that table.

**Response**: Thanks for the reminder, we have corrected the 'Angular precession' into 'longitude of the perihelion' on page 1 line 16, page 11 line 258 and page 53 of Table 2. We have also checked and corrected the value in Table 2 on page 53.

7. Figure S9. Why is it no 'Overall' values for MRI-CGCM3?

**Response:** Thanks for the reminder, the Figure S9 has been corrected on page 29 in Supplementary Information.

[revised manuscript text omitted]